# Bi-level Hierarchical Neural Contextual Bandits for Online Recommendation

**Yunzhe Qi**[1]**, Yao Zhou**[2]**, Yikun Ban**[1]**, Allan Stewart**[3]**, Chuanwei Ruan**[3,*]**, Jiachuan He**[3]**,
Shishir Kumar Prasad**[3]**, Haixun Wang**[4]**, Jingrui He**[1]
[1] University of Illinois Urbana-Champaign, [2] Google, [3] Instacart, [4] EvenUp
*{yunzheq2, jingrui}@illinois.edu, yaozhoucosmos@google.com*          * Work done while working at Instacart.

**Reviewed on OpenReview:** `https://openreview.net/forum?id=k3XsA75SGv`

## Abstract

Contextual bandit algorithms aim to identify the optimal choice among a set of candidate arms, based on their contextual information. Among others, neural contextual bandit algorithms have demonstrated generally superior performance compared to conventional linear and kernel-based methods. Nevertheless, neural methods can be inherently unsuitable for handling a large number of candidate arms due to their high computational cost when performing principled exploration. Motivated by the widespread availability of arm category information (e.g., movie genres, retailer types), we formulate contextual bandits as a bi-level online recommendation problem, and propose a novel neural bandit framework, named $H_2N$-Bandit, which utilizes a bi-level hierarchical neural architecture to mitigate the substantial computational cost found in conventional neural bandit methods. To demonstrate its theoretical effectiveness, we provide regret analysis under general over-parameterization settings, along with a guarantee for category-level recommendation. To illustrate its effectiveness and efficiency, we conduct extensive experiments on multiple real-world data sets, highlighting that $H_2N$-Bandit can significantly reduce the computational cost over existing strong non-linear baselines, while achieving better or comparable performance under online recommendation settings.

## 1 Introduction

Contextual bandits are a type of multi-armed bandit (MAB) that leverages contextual information to navigate the exploration-exploitation dilemma (Auer et al., 2002), in online settings without state transitions or sequential dependencies. Unlike greedy algorithms that only exploit known information (He et al., 2017), contextual bandits can utilize *principled exploration* strategies to achieve improved long-term outcomes, making them valuable in various real-world applications, such as medical experiment (Durand et al., 2018) and personalized recommendation (Li et al., 2010; Wu et al., 2016). For example, consider an online grocery platform that decides which retailer to recommend. It can either exploit its knowledge of a user's preferred retailer or explore a new one to discover a potentially new match. *Principled exploration* strategies, such as Upper Confidence Bound (UCB) (Li et al., 2010), enables the platform to balance this trade-off, by adaptively exploring new options based on their levels of uncertainty. By intelligent exploration, the system can effectively uncover user preferences and improve long-term satisfaction (Li et al., 2010; 2011).

Among various types of contextual bandit algorithms, neural contextual bandits (Zhou et al., 2020; Zhang et al., 2021) have recently risen to prominence, due to the superior representation and learning capabilities of neural networks. However, a nontrivial challenge arises from their computational cost, particularly when dealing with a large pool of candidate arms (e.g., a large number of candidate retailers). One major reason is that neural bandit methods with *principled exploration* typically require gradient computation for each arm to support effective exploration, making them less viable in real-world applications (Jia et al., 2022). To address this challenge, we leverage widely available arm hierarchical information (e.g., retailer types, movie genres) (Hu et al., 2013; Zhu et al., 2021), for a refined recommendation strategy. Our approach differs from conventional

contextual bandit algorithms that operate in a "flat" action space (Li et al., 2010; Valko et al., 2013; Zhou et al., 2020), by utilizing the arm category hierarchy to efficiently narrow down the candidate pool before the final arm selection. Inspired by the potential of arm hierarchies to improve bandit performance and efficiency (Hong et al., 2022; Zuo et al., 2022; Aouali et al., 2023), our work pioneers the hierarchical neural bandit framework. This formulation is uniquely capable of achieving strong recommendation outcomes in potentially complex settings with unknown, arbitrary arm reward mappings (Zhou et al., 2020). Specifically, we formulate a bi-level hierarchical strategy to improve efficiency: first, the category-level model selects a small subset of high-potential arm categories. Second, with an arm-level model, the arm recommendation is performed by only selecting from the candidates within this pre-filtered set. This approach can substantially reduce computational cost, as the final selection is performed on a dramatically smaller pool of candidates, rather than the entire candidate arm space. Distinct from conventional greedy recommendation methods that utilize the item category information (Cai et al., 2021; He et al., 2016), we tackle the inherent exploitation-exploration dilemma with bandit-based *principled exploration*.

Meanwhile, this bi-level hierarchical strategy can also adeptly navigate the exploitation-exploration dilemma at the *category level*, facilitating more efficient exploration by preventing unnecessary effort on unpromising categories. For example, consider a user with an unknown seafood allergy. Instead of learning this aversion for individual arms (e.g., learning from arm contexts with category information), our hierarchical design can aggregate feedback at a higher level. From negative interactions, the category-level model can perceive that the entire "seafood" category is unpromising, allowing the system to prune seafood-related arms before the computationally expensive arm-level scoring phase, yielding a dual benefit: significant computational savings and more sample-efficient exploration, as the model learns to avoid unpromising options more efficiently.

Motivated by real-world application scenarios and to tackle the aforementioned challenges, we propose a novel contextual bandit framework named Bi-level Hierarchical Neural Bandit ($H_2N$-Bandit), which leverages the arm category information to formulate a highly efficient yet effective recommendation strategy. To manage the potentially vast number of candidate arms, $H_2N$-Bandit formulates contextual bandits into a bi-level hierarchical recommendation problem, where it initially reduces the size of the candidate pool by filtering arm categories, thus streamlining the selection process for the final arm recommendation. Our $H_2N$-Bandit not only reduces the computational demands by utilizing a filtered pool of candidate arms, but also incorporates two types of principled exploration strategies for category-level and arm-level recommendations. This bi-level design improves efficiency while effectively addressing the exploitation-exploration dilemma across both levels. Our main contributions can be summarized as follows:

**Problem Formulation and Proposed Framework**: We formulate and study the problem of hierarchical neural contextual bandits, where a learner is expected to navigate a bi-level exploitation-exploration trade-off across both arm categories and individual arms under unknown, potentially complex reward mappings. Unlike existing hierarchical works that rely on assumed reward priors, $H_2N$-Bandit is the first stochastic neural bandit framework capable of handling arbitrary arm reward functions. By leveraging the arm-category hierarchy, $H_2N$-Bandit can perform principled exploration effectively and efficiently at both levels. This structural design allows it to prune the candidate pool, focusing on promising arms and improving computational efficiency, while maintaining strong recommendation performance.

**Theoretical Analysis**: Under general over-parameterization settings for neural bandit, we establish the first regret analysis for hierarchical stochastic neural bandit. To handle the absence of true category rewards, we develop a comprehensive theoretical framework that jointly optimizes both category and arm-level models. Our framework achieves greater generality by avoiding the restrictive arm separateness and reward-prior assumptions, which are common in existing works. We also provide a theoretical guarantee for category-level recommendation of $H_2N$-Bandit, demonstrating the effectiveness of our bi-level design and instantiation.

**Experiments**: Extensive experiments are performed on public real-world data sets, where we demonstrate that our approach not only delivers better or comparable performance compared to strong existing stochastic bandit baselines, but also offers a substantial advantage in computational efficiency. We further investigate our framework's properties and capabilities through additional experiments, including a parameter study, an analysis of the category-level strategy, and an extension of $H_2N$-Bandit to three-level hierarchy settings.

## 2 Related Works

With assumed linear reward mapping functions, linear contextual bandit algorithms (Chu et al., 2011; Li et al., 2010; Auer et al., 2002; Abbasi-Yadkori et al., 2011) are first proposed to tackle the exploitation-exploration dilemma. In particular, Lin-UCB (Chu et al., 2011; Li et al., 2010) utilizes linear regression, with the UCB for principled exploration. Afterward, to formulate the reward and confidence bound estimation for non-linear settings, kernelized bandits (Valko et al., 2013; Krause & Ong, 2011; Deshmukh et al., 2017) assume the reward mapping function belongs to the Reproducing Kernel Hilbert Space (RKHS). Meanwhile, collaborative algorithms (Gentile et al., 2014; Li et al., 2019; Ban & He, 2021b; Gentile et al., 2017; Cesa-Bianchi et al., 2013; Wu et al., 2016; Ban et al., 2024; Su et al., 2025) try to incorporate similar users based on the estimated or known *user correlations* or perform user-item collaborative filtering (Li et al., 2016). Several works leverage arm hierarchy information (Hong et al., 2022; Zuo et al., 2022; Aouali et al., 2023; Yang et al., 2020a; Sen et al., 2021) but rely on pre-defined reward mapping priors tailored to specific settings. Yang et al. (2020a) addresses resource-constrained linear bandits with budgeted recommendations, Hong et al. (2022) assumes a tree-structured action space with known priors and fixed cardinality under Bayes regret, and Sen et al. (2021) studies top-k bandits with realizability assumptions (e.g., known function class). These settings differ from our stochastic neural bandit, as the pre-defined reward assumptions can fail in real-world applications.

Neural bandit algorithms leverage the representation power of neural networks, to relax structural assumptions on reward mappings. Inspired by Neural Tangent Kernel (NTK) (Jacot et al., 2018), prior works apply fully-connected (FC) neural networks for reward estimation, with NTK-based UCB (Zhou et al., 2020; Ban et al., 2021; Ban & He, 2021a; Qi et al., 2022; Kassraie & Krause, 2022; Gu et al., 2024; Dai et al., 2023; Ban et al., 2024) or Thompson Sampling (Zhang et al., 2021) for exploration. Meanwhile, Ban et al. (2022); Qi et al. (2023) utilizes the second neural network to learn the confidence ellipsoid for adaptive exploration. However, these approaches require searching across the entire candidate arm pool, which introduces additional computational challenges, especially when the number of candidate arms is large in practice. While Osband et al. (2023); Zhu & Van Roy (2023a;b) employ heuristic EpiNet-style uncertainty estimation for improved efficiency, they typically assume a fixed, known action space and lack theoretical guarantees. This limits their applicability in our stochastic neural bandit settings, where rigorous regret analysis is a critical requirement, and the arm space is typically unknown, dynamic, or unbounded (Zhou et al., 2020; Zhang et al., 2021).

## 3 Problem Definition

Given a finite horizon $T \in \mathbb{N}^+$, consider a fixed pool $\mathcal{C}$ of arm categories, and each category $c \in \mathcal{C}$ is associated with an unknown category context distribution. As arm contexts can include the background from *both item and user perspectives*, a setup common in applications such as "*Online Personalized News Recommendation*" (Li et al., 2010), it is natural to reason that arm contexts can capture information from both sides.

In each round $t \in [T]$, we will receive a subset of arm categories $\mathcal{C}_t \subseteq \mathcal{C}$. For instance, retailer categories available can vary across regions (e.g., large wholesale chains are typically absent in rural areas). For each category $c \in \mathcal{C}_t$, the collection of $K_{c,t}$ candidate arms is denoted by $\mathcal{X}_{c,t} = \{\boldsymbol{x}_{c,t}^{(1)}, \ldots, \boldsymbol{x}_{c,t}^{(K_{c,t})}\}$, drawn from the category-specific context distribution, with $|\mathcal{X}_{c,t}| = K_{c,t}$. Consequently, the overall set of candidate arms, irrespective of their categories, is given by $\mathcal{X}_t = \bigcup_{c \in \mathcal{C}_t} \mathcal{X}_{c,t}$, with total size $|\mathcal{X}_t| = K$. We generally have $|\mathcal{X}_t| = K \gg |\mathcal{C}_t|$. Thus, there are $K$ candidate arms for selection in round $t$ in total, following the stochastic contextual bandit setting (Zhou et al., 2020). By definitions, an arm can belong to multiple categories: for some $c, c' \in \mathcal{C}_t$, it is possible that $\mathcal{X}_{c,t} \cap \mathcal{X}_{c',t} \neq \emptyset$. For example, a "Vegan Mexican" restaurant belongs to both the "Vegan" and "Mexican" categories. With $d$-dimensional arm context $\boldsymbol{x}_{c,t}^{(i)} \in \mathcal{X}_t, i \in [K_{c,t}]$, we consider its reward $r_{c,t}^{(i)}$ is given by

$$r_{c,t}^{(i)} = h_A(\boldsymbol{x}_{c,t}^{(i)}) + \epsilon_{c,t}^{(i)} \tag{1}$$

where $h_A(\cdot)$ represents the unknown arm-level reward mapping function, and $\epsilon$ is zero-mean $\alpha$-sub-Gaussian noise, consistent with prior studies (Zhou et al., 2020; Zhang et al., 2021). Considering practical scenarios like online recommendation with normalized ratings, for an arbitrary arm $\boldsymbol{x}$ and its reward $r$, we consider the bounded expected reward $\mathbb{E}_\epsilon[r] = h_A(\boldsymbol{x}) \in [0, 1]$, similar to Gentile et al. (2014; 2017); Ban & He (2021b); Qi et al. (2023). In round $t$, the learner aims to choose one arm $\boldsymbol{x}_t \in \mathcal{X}_t$ with the *maximal expected reward*.

**Learning Objective.** Within the framework of stochastic contextual bandits, the learning goal of the learner is selecting one arm $\boldsymbol{x}_t \in \mathcal{X}_t$ in each round $t \in [T]$, in order to minimize cumulative pseudo-regret:

$$R(T) = \mathbb{E}_\epsilon \big[ \sum\nolimits_{t=1}^{T} (r_t^* - r_t) \big] \tag{2}$$

where $r_t$ is the reward of chosen arm $\boldsymbol{x}_t$; and $r_t^*$ is the reward of optimal arm $\boldsymbol{x}_t^*$, i.e., $\mathbb{E}[r_t^* | \mathcal{X}_t] = h_A(\boldsymbol{x}_t^*) = \max_{\boldsymbol{x}_{c,t}^{(i)} \in \mathcal{X}_t} h_A(\boldsymbol{x}_{c,t}^{(i)})$. Motivated by this learning objective, we formulate our category reward definition below.

**Category Reward Formulation.** In round $t$, we define the expected category reward $\mathbb{E}[r_{c,t} | \mathcal{X}_{c,t}]$ as the maximum expected reward among the arms within category $c \in \mathcal{C}$. Let $\boldsymbol{\nu}_{c,t} \in \mathbb{R}^{d_C}$ be category representation derived from contexts $\mathcal{X}_{c,t}$. Then, with an unknown category-level mapping function $h_C(\cdot)$, we formulate

$$\mathbb{E}[r_{c,t} | \mathcal{X}_{c,t}] = h_C(\boldsymbol{\nu}_{c,t}) = \max_{i \in [K_{c,t}]} \mathbb{E}[r_{c,t}^{(i)}] = h_A(\boldsymbol{x}_{c,t}^*), \tag{3}$$

where $\boldsymbol{x}_{c,t}^* = \arg \max_{\boldsymbol{x}_{c,t}^{(i)} \in \mathcal{X}_{c,t}} h_A(\boldsymbol{x}_{c,t}^{(i)})$ is the optimal arm in category $c$, and the expectation is taken over the random reward noise. Our specific instantiated formulation of category representations $\boldsymbol{\nu}$ and their integration are detailed in Subsec. 4.1.

**Design Motivation for Category-level Reward.** In round $t \in [T]$, the learner aims to choose *one arm* $x_t \in \mathcal{X}_t$ with the highest expected reward. Here, our category reward (Eq. 3) is designed such that selecting the best category $c_t^* \in \mathcal{C}_t$ naturally aligns with selecting the best arm $x_t^*$, i.e., $h_C(\boldsymbol{\nu}_{c_t^*, t}) = \max_{c \in \mathcal{C}_t} h_C(\boldsymbol{\nu}_{c,t}) = \max_{\boldsymbol{x} \in \mathcal{X}_t} h_A(\boldsymbol{x}) = h_A(\boldsymbol{x}_t^*)$. Thus, the target of the category-level model will naturally be choosing $B$ arm categories $\tilde{\mathcal{C}}_t \subset \mathcal{C}_t$, such that the highest-reward arms fall within them. This category-based formulation underpins our *divide-and-conquer* workflow, reducing the computational complexity to $\mathcal{O}(|\mathcal{C}_t| + |\bigcup_{c \in \tilde{\mathcal{C}}_t} \mathcal{X}_{c,t}|)$, rather than $\mathcal{O}(K)$ in conventional neural bandit (e.g., Zhou et al. (2020); Zhang et al. (2021)). It also ensures a close connection between category-level and arm-level models, which is crucial for our theoretical analysis. On the other hand, alternative definitions (e.g., an *average* category reward) can misalign with the goal of stochastic contextual bandits: *choosing **one** optimal arm per round*, as a high average reward does not necessarily indicate the presence of the best arm. Complementary discussions are shown in Appendix D.

**Notation.** We apply vectors $\boldsymbol{x}, \boldsymbol{\nu}$ to denote arm context and category representation, respectively. $\mathcal{C}_t$ denotes available arm categories in round $t \in [T]$, where each $c \in \mathcal{C}_t$ has arm collection $\mathcal{X}_{c,t} \subset \mathbb{R}^d$. We aggregate candidate arms as $\mathcal{X}_t = \bigcup_{c \in \mathcal{C}_t} \mathcal{X}_{c,t}$. For the category level, let $\tilde{\mathcal{C}}_t \subset \mathcal{C}_t$ denote selected categories. For the arm level, we have the chosen arm $\boldsymbol{x}_t \in \tilde{\mathcal{X}}_t$, with $\tilde{\mathcal{X}}_t = \bigcup_{c \in \tilde{\mathcal{C}}_t} \mathcal{X}_{c,t}$ being the filtered candidate arms. For readers' convenience, we have also included a notation table in Appendix A.

## 4 Proposed Framework

**Overview of H$_2$N-Bandit.** Our pseudo-code is provided in Algorithm 1. With the arm-category hierarchy, the category-level model is utilized to filter arm categories before the actual arm recommendation. Inspired by the neural adaptive exploration (Ban et al., 2022; Qi et al., 2023), we implement two neural networks $f_C^{(1)}(\cdot; \boldsymbol{\Theta}_C^{(1)}), f_C^{(2)}(\cdot; \boldsymbol{\Theta}_C^{(2)})$ to perform exploitation and exploration respectively, where $\boldsymbol{\Theta}_C$ refer to their trainable parameters. The estimated category rewards and the exploration scores are applied for category selection (lines 5-7, Algorithm 1), effectively narrowing down the entire pool of candidate arms, and significantly reducing computational costs. Subsequently, the arm-level neural model is utilized to perform the precise arm recommendation (lines 8-11, Algorithm 1). Here, we adopt the arm-level network $f_A(\cdot; \boldsymbol{\Theta}_A)$ for arm reward estimation with trainable parameters $\boldsymbol{\Theta}_A$, and employ UCB-based exploration. After selecting one arm and obtaining its reward (lines 11 and 12, Algorithm 1), we update the neural models with Gradient Descent (GD) (lines 12-16, Algorithm 1). We leverage two exploration strategies to balance performance and computational efficiency. For the category-level model, where the number of choices is relatively small, we use an adaptive exploration strategy known for its strong performance (Ban et al., 2022; Qi et al., 2023). For the more computationally demanding arm-level model, we adopt a relatively lightweight UCB-based approach, allowing us to leverage the performance benefits of bi-level exploration while maintaining overall efficiency.

---

**Algorithm 1** Bi-level Hierarchical Neural Bandit (H$_2$N-Bandit)

---

1: **Input:** Horizon $T$. GD iterations $J$. Learning rates $\eta_C, \eta_A$. Exploration coefficient $\alpha$. Regularization coefficient $\lambda$. Number of selected arm categories $B$.

2: **Initialization:** Initialize parameters $[\boldsymbol{\Theta}_C^{(1)}]_0, [\boldsymbol{\Theta}_C^{(2)}]_0$, and $[\boldsymbol{\Theta}_A]_0$. Initialize covariance matrix: $\boldsymbol{\Sigma}_0 = \lambda \mathbf{I}$.

3: **for** each round $t \in [T]$ **do**

4:      Receive available categories $\mathcal{C}_t$ and their arms $\mathcal{X}_{c,t}, c \in \mathcal{C}_t$.

         ▷ **First-stage (category-level) recommendation**

5:      Derive the representation $\boldsymbol{\nu}_{c,t}$ for each arm category $c \in \mathcal{C}_t$.

6:      With category-level networks, calculate the *overall score* for each category $c \in \mathcal{C}_t$, denoted by $\mathcal{F}_C(\boldsymbol{\nu}_{c,t}; [\boldsymbol{\Theta}_C]_{t-1}) := f_C^{(1)}(\boldsymbol{\nu}_{c,t}; [\boldsymbol{\Theta}_C^{(1)}]_{t-1}) + f_C^{(2)}(\nabla f_C^{(1)}(\boldsymbol{\nu}_{c,t}); [\boldsymbol{\Theta}_C^{(2)}]_{t-1})$.

7:      Choose top-$B$ arm categories $\widetilde{\mathcal{C}}_t = \operatorname{argmax}_{\widetilde{\mathcal{C}}_t \subset \mathcal{C}_t, |\widetilde{\mathcal{C}}_t| = B} \left[ \sum_{c \in \widetilde{\mathcal{C}}_t} \mathcal{F}_C(\boldsymbol{\nu}_{c,t}; [\boldsymbol{\Theta}_C]_{t-1}) \right]$ with *highest overall scores*. Form the filtered candidate arm pool $\widetilde{\mathcal{X}}_t = \bigcup_{c \in \widetilde{\mathcal{C}}_t} \mathcal{X}_{c,t}$.

         ▷ **Second-stage (arm-level) recommendation**

8:      **for** each filtered candidate arm $\boldsymbol{x}_{c,t}^{(i)} \in \widetilde{\mathcal{X}}_t$ **do**

9:          Calculate the reward estimation $f_A(\boldsymbol{x}_{c,t}^{(i)}; [\boldsymbol{\Theta}_A]_{t-1})$, and its UCB value $\mathsf{UCB}_\alpha(\boldsymbol{x}_{c,t}^{(i)})$.

10:      **end for**

11:      Recommend arm $\boldsymbol{x}_t = \arg\max_{\boldsymbol{x}_{c,t}^{(i)} \in \widetilde{\mathcal{X}}_t} \left[ f_A(\boldsymbol{x}_{c,t}^{(i)}; [\boldsymbol{\Theta}_A]_{t-1}) + \mathsf{UCB}_\alpha(\boldsymbol{x}_{c,t}^{(i)}) \right]$.

         ▷ **Updating Neural Bandit Models**

12:      Receive reward $r_t$, and update arm-level model $f_A$ with GD. Update gradient covariance matrix $\boldsymbol{\Sigma}_t = \boldsymbol{\Sigma}_{t-1} + g(\boldsymbol{x}_t; [\boldsymbol{\Theta}_A]_{t-1}) \cdot g(\boldsymbol{x}_t; [\boldsymbol{\Theta}_A]_{t-1})^\mathsf{T} / m$.

13:      **for** each chosen arm category $c \in \widetilde{\mathcal{C}}_t$ **do**

14:          Record $\boldsymbol{\nu}_{c,t}$, and its pseudo reward $\widetilde{r}_{c,t} = \min\left\{ \max_{\boldsymbol{x}_{c,t}^{(i)} \in \mathcal{X}_{c,t}} [f_A(\boldsymbol{x}_{c,t}^{(i)}; [\boldsymbol{\Theta}_A]_{t-1}) + \mathsf{UCB}_\alpha(\boldsymbol{x}_{c,t}^{(i)})], \ 1 \right\}$.

15:      **end for**

16:      Train category-level models $f_C^{(1)}, f_C^{(2)}$ with collected category representations and pseudo rewards.

17: **end for**

---

## 4.1 First Level: Category-level Recommendation

Recall that each category $c \in \mathcal{C}_t, \forall t \in [T]$ corresponds to a representation vector $\boldsymbol{\nu}_{c,t}$ to encode the information regarding its associated arm collection $\mathcal{X}_{c,t}$. In this work, we formulate the category representation as

$$\boldsymbol{\nu}_{c,t} := \left[ \frac{1}{|\mathcal{X}_{c,t}|} \Big( \sum\nolimits_{\boldsymbol{x}_{c,t}^{(i)} \in \mathcal{X}_{c,t}} \boldsymbol{x}_{c,t}^{(i)} \Big); \ \mathsf{Std}(\mathcal{X}_{c,t}) \right] \in \mathbb{R}^{d_C} \tag{4}$$

with $[\cdot; \cdot]$ being the concatenation operation, and naturally $d_C = 2d$. We recall that $d$ is the context dimension. Here, given the arm collection $\mathcal{X}_{c,t}$ of category $c$, we have $\mathsf{Std}(\mathcal{X}_{c,t}) \in \mathbb{R}^d$ being the element-wise standard deviation w.r.t. each element of the arm context vector $\boldsymbol{x}_{c,t}^{(i)} \in \mathcal{X}_{c,t}$.

**Formulation Intuition of Category Representation.** The intuition is that we need to encode the context spread for arm collection $\mathcal{X}_{c,t}$ within the context space. Here, with the first $d$ elements (the average arm context) representing the center context point, the second $d$ elements in $\boldsymbol{\nu}_{c,t}$ (i.e., $\mathsf{Std}(\mathcal{X}_{c,t})$) formulate the interval around that mean point, in terms of each dimension within the context space. With the representation in Eq. 4, we encode the given arm contexts $\mathcal{X}_{c,t}$ as the dense hyper-ellipsoid. In this way, with a region that is proportional to this ellipsoid (e.g., according to the $3\sigma$ rule), we can have each arm from $\mathcal{X}_{c,t}$ falling into this region with a high probability. As a result, motivated by the "optimism-in-the-face-of-uncertainty" principle (Abbasi-Yadkori et al., 2011; Yang et al., 2020b), we utilize this design of the category representation to provide sufficient information.

### 4.1.1 Category Reward and Exploration Score Estimation.

With category representation $\boldsymbol{\nu}_{c,t}, c \in \mathcal{C}_t$, we apply category exploitation network $f_C^{(1)}(\cdot; \boldsymbol{\Theta}_C^{(1)})$ to estimate the category reward (exploitation). It is also known that the confidence interval of reward estimation can be expressed as a function of the reward estimation model gradients (i.e., the gradients of exploitation network $f_C^{(1)}$) (Ban et al., 2022; Zhou et al., 2020; Qi et al., 2023). Therefore, to learn this relationship, we adopt a second category exploration network $f_C^{(2)}(\cdot; \boldsymbol{\Theta}_C^{(2)})$ to calculate the exploration score for category $c \in \mathcal{C}_t$. This measures the "potential gain", i.e., $\mathbb{E}[r_{c,t}|\mathcal{X}_{c,t}] - f_C^{(1)}(\boldsymbol{\nu}_{c,t}; \boldsymbol{\Theta}_C^{(1)})$, in terms of the category reward estimation. In this case, the input of exploration network $f_C^{(2)}$ will be the normalized gradient of the exploitation model: $\nabla f_C^{(1)}(\boldsymbol{\nu}_{c,t}) := \nabla_{\boldsymbol{\Theta}} f_C^{(1)}(\boldsymbol{\nu}_{c,t}; \boldsymbol{\Theta}_C^{(1)})$, where we treat it as a column vector for notation simplicity. The estimated exploration score is denoted by $f_C^{(2)}(\nabla f_C^{(1)}(\boldsymbol{\nu}_{c,t}); \boldsymbol{\Theta}_C^{(2)})$.

**Category Selection.** We select top-$B$ categories (line 7, Algorithm 1) with the highest overall scores $\{\mathcal{F}_C(\boldsymbol{\nu}_{c,t}; \boldsymbol{\Theta}_C)\}_{c \in \mathcal{C}_t} := \left\{ f_C^{(1)}(\boldsymbol{\nu}_{c,t}; \boldsymbol{\Theta}_C^{(1)}) + f_C^{(2)}(\nabla f_C^{(1)}(\boldsymbol{\nu}_{c,t}); \boldsymbol{\Theta}_C^{(2)}) \right\}_{c \in \mathcal{C}_t}$ as the chosen arm categories $\widetilde{\mathcal{C}}_t$. Then, the chosen categories and their associated candidate arms $\widetilde{\mathcal{X}}_t = \bigcup_{c \in \widetilde{\mathcal{C}}_t} \mathcal{X}_{c,t}$ will be consequently applied for the arm-level recommendation in Subsec. 4.2.

**Remark 1** (Reducing Recommendation Complexity)**.** *For conventional stochastic neural bandit algorithms, the computational cost is $\mathcal{O}(K)$, which can be significantly expensive as the learner needs to perform computationally intensive principled exploration for every candidate arm. On the other hand, $H_2N$-Bandit operates with the computational cost of $\mathcal{O}(|\mathcal{C}_t| + |\bigcup_{c \in \widetilde{\mathcal{C}}_t} \mathcal{X}_{c,t}|)$ where $|\bigcup_{c \in \widetilde{\mathcal{C}}_t} \mathcal{X}_{c,t}| < K$, $|\mathcal{C}_t| \ll K$, resulting in a significant reduction of complexity.*

**Category-level Neural Networks.** Recall that we adopt two separate networks $f_C^{(1)}, f_C^{(2)}$ in terms of exploitation and exploration respectively. Given the input category representation $\boldsymbol{\nu}_{c,t} \in \mathbb{R}^{d_C}$, $t \in [T]$, we first define the exploitation network $f_C^{(1)}$ to be a FC network, with depth $L \geq 2$ and width $m \in \mathbb{N}^+$:

$$f(\boldsymbol{\nu}_{c,t}; \boldsymbol{\Theta}_C^{(1)}) := \boldsymbol{\Theta}_L \phi(\boldsymbol{\Theta}_{L-1} \phi(\boldsymbol{\Theta}_{L-2} \dots \phi(\boldsymbol{\Theta}_1 \boldsymbol{\nu}_{c,t}))) \tag{5}$$

where $\phi(\cdot)$ is ReLU activation, $\boldsymbol{\Theta}_1 \in \mathbb{R}^{m \times d_C}$, $\boldsymbol{\Theta}_l \in \mathbb{R}^{m \times m}$, $2 \leq l \leq L-1$, $\boldsymbol{\Theta}^L \in \mathbb{R}^{1 \times m}$. For notation simplicity, we denote its parameters as a column vector: $\boldsymbol{\Theta}_C^{(1)} := [\text{vec}(\boldsymbol{\Theta}_1)^\intercal, \text{vec}(\boldsymbol{\Theta}_2)^\intercal, \dots, \text{vec}(\boldsymbol{\Theta}_L)^\intercal]^\intercal$, and we use $[\boldsymbol{\Theta}_C^{(1)}]_t$ to represent the parameters $\boldsymbol{\Theta}_C^{(1)}$, after being trained by GD in $t$-th round.

Analogously, we also apply an $L$-layer FC network (with width $m$) to be the exploration network $f_C^{(2)}$ that can be similarly represented by Eq. 5, with the trainable parameters $\boldsymbol{\Theta}_C^{(2)}$. Note that the first-layer weight matrices between these two networks will have different shapes, since $f_C^{(2)}$ is taking the network gradients $\nabla f_C^{(1)}(\cdot)$ as the input instead. Similar to existing works (Ban et al., 2022; Qi et al., 2023), we apply *average pooling* to the original gradient vector $\nabla f_C^{(1)}(\cdot)$ before feeding it into $f_C^{(2)}$, in order to reduce both runtime and space complexity. We adopt this method in $H_2N$-Bandit for all experiments (Section 6).

### 4.1.2 Efficient Category-level Training with Pseudo Rewards.

Recall that for category $c \in \widetilde{\mathcal{C}}_t$, its expected category reward is defined as the highest expected arm reward within the category-specific arm pool $\mathcal{X}_{c,t}$ (Eq. 3). Consequently, a key challenge is the *lack of access to true category rewards*: we can only observe the reward $r_t$ for the chosen arm $\boldsymbol{x}_t$, which may not be the best arm within the selected categories.

**Pseudo Category Reward.** To tackle this challenge, we know that the combination of reward point estimation ($f_A(\boldsymbol{x})$) and the corresponding UCB (i.e., $\mathsf{UCB}_\alpha(\boldsymbol{x})$ from Eq. 7) can offer an upper bound for the expected arm reward for an arbitrary arm $\boldsymbol{x}$ with a high probability, based on derivations of our Theorem 5.1. Inspired by the principle of "optimism-in-the-face-of-uncertainty" (Abbasi-Yadkori et al., 2011), we use this upper bound as the *"pseudo reward"* for training category-level models. This enables us to allocate the pseudo-reward to each chosen category $c \in \widetilde{\mathcal{C}}_t$, thereby enriching the training samples for category-level

networks. For each chosen category $c \in \widetilde{\mathcal{C}}_t$ and its candidate pool $\mathcal{X}_{c,t}$, we define its clipped pseudo-reward

$$\widetilde{r}_{c,t} = \min\left\{1, \ \max_{\boldsymbol{x}_{c,t}^{(i)} \in \mathcal{X}_{c,t}} [f_A(\boldsymbol{x}_{c,t}^{(i)}; [\boldsymbol{\Theta}_A]_{t-1}) + \mathsf{UCB}_\alpha(\boldsymbol{x}_{c,t}^{(i)})]\right\}, \tag{6}$$

corresponding to line 14 of Algorithm 1. The effectiveness of this formulation is supported by our theoretical analysis (Section 5) and experiments (Section 6). Note that, since arm reward estimates and UCBs are *already computed* (lines 8–10, Algorithm 1), we can directly reuse them to derive the category-level pseudo-rewards without incurring additional computational costs.

**Model Training.** With collected category representations and pseudo rewards $\{\boldsymbol{\nu}_{c,\tau}, \ \widetilde{r}_{c,\tau}\}_{\tau \in [t], c \in \widetilde{\mathcal{C}}_\tau}$, we define the training loss for the exploitation network $f_C^{(1)}$ as $\mathcal{L}_C^{(1)}(\boldsymbol{\Theta}) = \frac{1}{2}\sum_{\tau \in [t]}\sum_{c \in \widetilde{\mathcal{C}}_\tau} \left|f_C^{(1)}(\boldsymbol{\nu}_{c,\tau}; \boldsymbol{\Theta}) - \widetilde{r}_{c,\tau}\right|^2$, where the label is pseudo reward $\widetilde{r}$. Meanwhile, since the input of the exploration network $f_C^{(2)}$ is the gradients of $f_C^{(1)}$, we can utilize the previous gradients and the potential gain to train the exploration network. Thus, we formulate its loss: $\mathcal{L}_C^{(2)}(\boldsymbol{\Theta}) = \frac{1}{2}\sum_{\tau \in [t]}\sum_{c \in \widetilde{\mathcal{C}}_\tau} \left|f_C^{(2)}(\nabla f_C^{(1)}(\boldsymbol{\nu}_{c,\tau}); \boldsymbol{\Theta}) - \left(\widetilde{r}_{c,\tau} - f_C^{(1)}(\boldsymbol{\nu}_{c,\tau}; [\boldsymbol{\Theta}_C^{(1)}]_{\tau-1})\right)\right|^2$, and the label is "potential gain" $\widetilde{r} - f_C^{(1)}(\boldsymbol{x})$. With the loss functions above, we run Gradient Descent for $J$ iterations to update networks $f_C^{(1)}$ and $f_C^{(2)}$, given the learning rate $\eta_C$ (line 16, Algorithm 1).

## 4.2 Second Level: Arm-level Recommendation

Next, with the chosen categories $\widetilde{\mathcal{C}}_t \subset \mathcal{C}_t$, we need to recommend one arm $\boldsymbol{x}_t \in \widetilde{\mathcal{X}}_t$ from the filtered candidate arms $\widetilde{\mathcal{X}}_t = \bigcup_{c \in \widetilde{\mathcal{C}}_t} \mathcal{X}_{c,t}$ accordingly.

**Arm Reward Estimation and UCB.** Given an arm $\boldsymbol{x}$, we apply the arm-level neural network $f_A(\cdot; \boldsymbol{\Theta}_A)$ to estimate its reward, and the reward estimation will naturally be the network output $f_A(\boldsymbol{x}; \boldsymbol{\Theta}_A)$. To achieve arm-level exploration, we apply UCB-based strategy to reduce the computational cost. Given probability constant $\delta \in (0,1)$, we aim to formulate the UCB, such that $\mathbb{P}\big(|f_A(\boldsymbol{x}; \boldsymbol{\Theta}_A) - h_A(\boldsymbol{x})| > \mathsf{UCB}_\alpha(\boldsymbol{x})\big) \le \delta$, for each arm $\boldsymbol{x} \in \widetilde{\mathcal{X}}_t$. Based on NTK norm $S > 0$ and the confidence ellipsoid derivations from Theorem 5.1, with the probability at least $1 - \delta$, we have

$$\mathsf{UCB}_\alpha(\boldsymbol{x}) = \mathcal{O}\left(\alpha\sqrt{\log\frac{\det(\boldsymbol{\Sigma}_{t-1})}{\det(\lambda\mathbf{I})} - 2\log(\delta)} + \lambda^{1/2}S\right) \cdot \|g(\boldsymbol{x}; [\boldsymbol{\Theta}_A]_{t-1})/\sqrt{m}\|_{\boldsymbol{\Sigma}_{t-1}^{-1}}, \tag{7}$$

where the covariance matrix $\boldsymbol{\Sigma}_{t-1} = \lambda\mathbf{I} + \sum_{\tau \in [t]} g(\boldsymbol{x}_\tau; [\boldsymbol{\Theta}_A]_{\tau-1}) \cdot g(\boldsymbol{x}_\tau; [\boldsymbol{\Theta}_A]_{\tau-1})^\intercal/m$, with the regularization coefficient $\lambda > 0$. Recall that in Eq. 1, $\alpha$ represents the variance proxy. If this value is the unknown prior, we can treat $\alpha \ge 0$ as a tunable coefficient controlling exploration intensity, as in existing works (Zhou et al., 2020; Zhang et al., 2021).

**Arm Selection.** Using reward estimation, along with the UCB scores and exploration coefficient $\alpha$, we recommend arm $\boldsymbol{x}_t$ with the highest overall score from the candidate pool $\widetilde{\mathcal{X}}_t$ (line 11, Algorithm 1):

$$\boldsymbol{x}_t = \arg\max_{\boldsymbol{x}_{c,t}^{(i)} \in \widetilde{\mathcal{X}}_t} \left[f_A(\boldsymbol{x}_{c,t}^{(i)}; \boldsymbol{\Theta}_A) + \mathsf{UCB}_\alpha(\boldsymbol{x}_{c,t}^{(i)})\right].$$

Upon the selection of arm $\boldsymbol{x}_t$, its actual reward $r_t$ will be revealed, following the stochastic bandit settings.

**Arm-level Neural Network and Training.** Similar to category-level models (Eq. 5), we also adopt an $L$-layer FC network for reward estimation: $f_A(\boldsymbol{x}; \boldsymbol{\Theta}_A) := \boldsymbol{\Theta}_L'\phi(\boldsymbol{\Theta}_{L-1}'\phi(\boldsymbol{\Theta}_{L-2}'\ldots\phi(\boldsymbol{\Theta}_1' \cdot \boldsymbol{x})))$, with the hidden dimension $m$. Note that the input of $f_A$ will be the arm contexts with dimensionality $d$. Therefore, the shape of the weight matrix in $f_A$'s first layer would be $\boldsymbol{\Theta}_1' \in \mathbb{R}^{m \times d}$. Analogous to category-level model, $\boldsymbol{\Theta}_A := [\text{vec}(\boldsymbol{\Theta}_1')^\intercal, \text{vec}(\boldsymbol{\Theta}_2')^\intercal, \ldots, \text{vec}(\boldsymbol{\Theta}_L')^\intercal]^\intercal \in \mathbb{R}^{p_A}$ represents the trainable parameters. We also update $f_A$ via Gradient Descent (line 12, Algorithm 1). With the chosen arms and their rewards $\{\boldsymbol{x}_\tau, r_\tau\}_{\tau \in [t]}$ up to round $t$, we define the quadratic loss function with regularization as $\mathcal{L}_A(\boldsymbol{\Theta}) = \frac{1}{2}\sum_{\tau \in [t]} \left|f_A(\boldsymbol{x}_\tau; \boldsymbol{\Theta}) - r_\tau\right|^2 + \frac{m\lambda}{2}\|\boldsymbol{\Theta} - \boldsymbol{\Theta}_0\|_2^2$. Then, we perform the Gradient Descent for $J$ iterations, with learning rate $\eta_A > 0$.

# 5 Theoretical Analysis

**Theoretical Contributions.** *First*, we establish the regret analysis for hierarchical stochastic neural bandit without assuming specific reward-mapping priors (e.g., Hong et al. (2022); Zuo et al. (2022); Aouali et al. (2023); Yang et al. (2020a); Sen et al. (2021)), allowing arbitrary arm reward functions for greater generality. *Second*, we propose a novel pipeline to address the challenge of the absence of true category-level rewards. By leveraging pseudo-rewards derived from the arm-level model, we ensure joint optimization of the bi-level models while maintaining theoretical guarantees. *Third*, we develop a novel theoretical proof framework for our two-level exploration strategy, where category-level neural exploration refines arm candidate pool, before arm-level UCB optimizes action choice, ensuring a balance between efficiency and performance. *Fourth*, unlike prior works with arm separateness assumptions (e.g., Zhou et al. (2020); Zhang et al. (2021); Ban et al. (2021; 2022)), we integrate novel confidence ellipsoids with martingale-based analysis to enable our regret analysis, without relying on such assumptions. *Finally*, with a general regret bound in Theorem 5.1, Theorem 5.2 can further tighten the bound under a mild "category margin" condition, highlighting the effectiveness of our category-level filtering and bi-level formulation.

## 5.1 Preliminaries for Theoretical Analysis

Inspired by Zhou et al. (2020); Allen-Zhu et al. (2019), for initializing category-level $f_C^{(1)}$ and $f_C^{(2)}$, we draw each entry of $\boldsymbol{\Theta}_l$, $\forall l \in [L-1]$, from the Gaussian distribution $\mathcal{N}(0, 2/m)$; each entry of $\boldsymbol{\Theta}_L$ is drawn from $\mathcal{N}(0, 1/m)$. For the arm-level model $f_A$, we sample the entries of its intermediate weight matrices $\boldsymbol{\Theta}'_l, l \in [L-1]$ from $\mathcal{N}(0, 4/m)$, while sampling the entries of $\boldsymbol{\Theta}'_L$ from Gaussian distribution $\mathcal{N}(0, 2/m)$.

Analogous to Zhou et al. (2020); Zhang et al. (2021); Ban et al. (2022); Qi et al. (2023; 2024), we consider unit-length arm contexts $\|\boldsymbol{x}_{c,t}^{(i)}\|_2 = 1, \forall i \in [K_{c,t}], c \in \mathcal{C}_t, t \in [T]$. Without loss of generality, we apply a normalization process inspired by Allen-Zhu et al. (2019); Zhou et al. (2020); Jia et al. (2022); Qi et al. (2024): with unprocessed $\widetilde{\boldsymbol{x}}_{c,t}^{(i)}$, the corresponding normalized arm context will be $\boldsymbol{x}_{c,t}^{(i)} = \left[ \frac{\widetilde{\boldsymbol{x}}_{c,t}^{(i)}}{2 \cdot \|\widetilde{\boldsymbol{x}}_{c,t}^{(i)}\|_2}, \frac{1}{2}, \frac{\widetilde{\boldsymbol{x}}_{c,t}^{(i)}}{2 \cdot \|\widetilde{\boldsymbol{x}}_{c,t}^{(i)}\|_2}, \frac{1}{2} \right]$.

We consequently have: (i) $\|\boldsymbol{x}_{c,t}^{(i)}\|_2 = 1$; (ii) no two normalized arm contexts will be in opposite directions; and (iii) $f_A(\boldsymbol{x}_{c,t}^{(i)}; [\boldsymbol{\Theta}_A]_0) = 0$ with randomly initialized parameters $[\boldsymbol{\Theta}_A]_0$.

**NTK Gram Matrices.** Let $\widetilde{\boldsymbol{x}}_t^* = \arg\max_{\boldsymbol{x} \in (\bigcup_{c \in \widetilde{\mathcal{C}}_t} \mathcal{X}_{c,t})} h(\boldsymbol{x})$ be the "best possible chosen" arm from chosen categories $\widetilde{\mathcal{C}}_t, t \in [T]$, along with the chosen arms $\{\boldsymbol{x}_t\}_{t \in [T]}$ and optimal arms $\{\boldsymbol{x}_t^*\}_{t \in [T]}$, where $\boldsymbol{x}_t^* = \arg\max_{\boldsymbol{x}_{c,t}^{(i)} \in \mathcal{X}_t} h_A(\boldsymbol{x}_{c,t}^{(i)})$. Then, we simply merge these three arm collections as $\mathcal{A}_T$ with cardinality $|\mathcal{A}_T| = 3T$, which can possibly contain duplicate arms. Meanwhile, we define $\bar{\mathcal{A}}_T := (\{\boldsymbol{x}_t\}_{t \in [T]} \cup \{\widetilde{\boldsymbol{x}}_t^*\}_{t \in [T]} \cup \{\boldsymbol{x}_t^*\}_{t \in [T]})$ as the union set; and consequently, $\bar{\mathcal{A}}_T$ will only contain unique arms from $\mathcal{A}_T$, s.t. $|\bar{\mathcal{A}}_T| \leq |\mathcal{A}_T| = 3T$.

**Definition 1** (NTK Gram Matrix with Possibly Duplicate Arms (Jacot et al., 2018; Zhou et al., 2020)). *Let $\mathcal{N}$ denote the Gaussian distribution. With ReLU activation $\phi(\cdot)$ and its derivative $\phi'(\cdot)$, define following recursive process. For $l \in [L]$ and $i, j \in \{1, \ldots, |\mathcal{A}_T|\}$, with $\mathbf{M}_{i,j}^0 = \boldsymbol{\Psi}_{i,j}^0 = \langle \boldsymbol{x}_i, \boldsymbol{x}_j \rangle$, let*

$$\mathbf{N}_{i,j}^l = \begin{pmatrix} \boldsymbol{\Psi}_{i,i}^l & \boldsymbol{\Psi}_{i,j}^l \\ \boldsymbol{\Psi}_{j,i}^l & \boldsymbol{\Psi}_{j,j}^l \end{pmatrix}, \quad \boldsymbol{\Psi}_{i,j}^l = 2\mathbb{E}_{a,b \sim \mathcal{N}(\mathbf{0}, \mathbf{N}_{i,j}^{l-1})}[\phi(a)\phi(b)],$$

$$\mathbf{M}_{i,j}^l = 2\mathbf{M}_{i,j}^{l-1}\mathbb{E}_{a,b \sim \mathcal{N}(\mathbf{0}, \mathbf{N}_{i,j}^{l-1})}[\phi'(a)\phi'(b)] + \boldsymbol{\Psi}_{i,j}^l.$$

*Given arm collection $\mathcal{A}_T$ ($|\mathcal{A}_T| = 3T$) with possibly duplicate arms, define its NTK Gram Matrix as $\mathbf{M} = (\mathbf{M}^L + \boldsymbol{\Psi}^L)/2 \in \mathbb{R}^{3T \times 3T}$.*

**Definition 2** (NTK Gram Matrix with *Unique* Arms). *Follow the recursive process in Def. 1. For arm set $\bar{\mathcal{A}}_T$ with unique arms, define its NTK matrix $\bar{\mathbf{M}} = (\bar{\mathbf{M}}^L + \bar{\boldsymbol{\Psi}}^L)/2 \in \mathbb{R}^{|\bar{\mathcal{A}}_T| \times |\bar{\mathcal{A}}_T|}$, where $|\bar{\mathcal{A}}_T| \leq |\mathcal{A}_T|$.*

**Remark 2.** *(Removing Arm Separateness Assumption) Most neural bandit works generally adopt arm separateness assumptions, e.g., Zhou et al. (2020); Zhang et al. (2021); Ban et al. (2021); Dai et al. (2023); Xu et al. (2020) assume minimum NTK matrix eigenvalue $\lambda_0 > 0$; Ban et al. (2022); Qi et al. (2023) assume the Euclidean separateness: $\|\boldsymbol{x} - \boldsymbol{x}'\|_2 > 0$ for any two observed candidate arms $\boldsymbol{x}, \boldsymbol{x}'$. They can be easily*

*violated when we have duplicate arm contexts. Inspired by Qi et al. (2024), we integrate refined confidence ellipsoids with our novel martingale-based analysis to derive regret bounds without separateness assumptions, enabling joint optimization of our bi-level models with two different exploration mechanisms.*

**Definition 3** (NTK Matrix Effective Dimension (Valko et al., 2013; Zhou et al., 2020; Zhang et al., 2021; Qi et al., 2024)). *Given horizon $T$, regularization coefficient $\lambda$ (Eq. 7), and NTK matrix $\mathbf{M}$, define effective dimension $\widetilde{d}$ of matrix $\mathbf{M}$ as $\widetilde{d} = \frac{\log \det(\mathbf{I}+\mathbf{M}/\lambda)}{\log(1+TK/\lambda)}$, with respect to arm collection $\mathcal{A}_T$ with $|\mathcal{A}_T| = 3T$.*

## 5.2 Regret Analysis

We also would like to mention that our theoretical analysis results can also be readily generalized to other types of neural architectures, such as Convolutional Neural Networks (CNNs) and ResNet with over-parameterization (Allen-Zhu et al., 2019; Cao & Gu, 2019). Next, we present the main regret analysis result in Theorem 5.1, considering both category-level and arm-level error.

**Theorem 5.1.** *Given the finite horizon $T$, for any $\delta \in (0,1)$ and given $\lambda, S, R, \xi_R > 0$, suppose width $m \geq \Omega(poly(T, L, R, \bar{\lambda}_0^{-1}, S^{-1}) \cdot K\xi_R \log(1/\delta))$, $\eta_C = \frac{R^2}{\sqrt{m}}$, $\eta_A = \mathcal{O}((TmL + m\lambda)^{-1})$, $J \geq \widetilde{\mathcal{O}}(TL/\lambda)$. Let $f_C^{(1)}, f_C^{(2)}$, and $f_A$ be the L-layer FC networks with width $m$. Then, $H_2N$-Bandit achieves the regret bound*

$$R(T) \leq \mathcal{O}\Big(\sqrt{T\widetilde{d}\log(1+TK/\lambda)}\big(\alpha\sqrt{\widetilde{d}\log(1+TK/\lambda) - 2\log(\delta)} + \sqrt{\lambda}S\big)\Big)$$
$$+ \mathcal{O}\big(\sqrt{T\log(1/\delta)} + \sqrt{T\xi_R}\big) + \mathcal{O}(1),$$

*with probability at least $1 - \delta$ over the random initialization.*

The proof is presented in Appendix E. Despite the category-level recommendation we applied to reduce the computational cost, our regret bound is comparable with state-of-the-art results in neural bandit works (Zhou et al., 2020; Zhang et al., 2021; Ban et al., 2021; Jia et al., 2022; Qi et al., 2022) with an overall regret of $\widetilde{\mathcal{O}}(\widetilde{d}\sqrt{T})$. Our NTK Gram matrices (Defs. 1, 2) depend on at most $3T$ arms, different from prior works (e.g., Zhou et al. (2020); Zhang et al. (2021)) that rely on all $TK$ candidate arms $\mathcal{X}_t, t \in [T]$, leading to a compact definition of effective dimension $\widetilde{d}$. In addition, our parameter norm $S$ can be tighter than in prior works (e.g., Zhang et al. (2021); Zhou et al. (2020)), as our confidence ellipsoid around $\mathbf{\Theta}_0$ only needs to hold for arms in collection $\bar{\mathcal{A}}_T$ ($|\bar{\mathcal{A}}_T| \leq 3T$), rather than all the $TK$ observed candidate arms.

## 5.3 Category-level Recommendation Analysis

Different from the Subsection 5.2 above, for the next Theorem 5.2, we instead consider settings analogous to those in existing works (Qi et al., 2022; Ban et al., 2022), where candidate arms $\mathcal{X}_t$ are sampled i.i.d. from a context distribution. Meanwhile, we also suppose that the expected category rewards satisfy a mild $\zeta$-margin condition below, to ensure their separability and facilitate stronger theoretical results. That is, with the optimal category $c_t^* = \arg\max_{c \in \mathcal{C}_t} h_C(\boldsymbol{\nu}_{c,t})$ and second-optimal category $c_t^\circ = \arg\max_{c \in (\mathcal{C}_t \setminus c_t^*)} h_C(\boldsymbol{\nu}_{c,t})$, we suppose there exist a constant $\zeta > 0$, s.t. $\left|\mathbb{E}[r_{c_t^*,t}|\mathcal{X}_{c_t^*,t}] - \mathbb{E}[r_{c_t^\circ,t}|\mathcal{X}_{c_t^\circ,t}]\right| \geq \zeta$, with notation from Eq. 3. Note that there can be multiple optimal or second-optimal categories with identical expected rewards respectively. As long as there is a gap $\zeta$ between the rewards of the optimal and second-optimal category(-ies), i.e., not all arm categories have the same expected reward, this mild condition will consequently hold.

**Theorem 5.2.** *Suppose $f_C^{(1)}, f_C^{(2)}$ are L-layer FC networks with width $m$. Let $m, L, J, \eta_C$ satisfy the conditions in Theorem 5.1. Suppose $H_2N$-Bandit chooses $B$ categories in each round $t$. Then, given the category margin $\zeta > 0$, with the notation from Theorem 5.1 and Algorithm 1, when round $t$ satisfies*

$$t \geq \widetilde{T} := \left(\frac{\widetilde{\mathcal{O}}\big(\alpha\sqrt{\widetilde{d}^2 - 2\widetilde{d}\log(\delta)} + S\sqrt{\lambda\widetilde{d}}\big) + \mathcal{O}\big(\sqrt{\log(\delta^{-1})} + \sqrt{\xi_R}\big)}{B \cdot \zeta}\right)^2,$$

*the optimal arm category $c_t^* = \arg\max_{c \in \mathcal{C}_t} h_C(\boldsymbol{\nu}_{c,t})$ is assured to be selected by the category-level model (i.e., $c_t^* \in \widetilde{\mathcal{C}}_t$) with probability at least $1 - \delta$.*

The proof is presented in Appendix F. According to Theorem 5.2, when $t \geq \widetilde{T}$, $H_2N$-Bandit is guaranteed to choose the optimal arm category with a high probability, where we will consequently have zero "category-level"

error. Thus, for the second Big-O term on the RHS of Theorem 5.1 that corresponds to the category-level error, we can replace the finite horizon $T$ term with the $T_{\min} = \min\{T, \widetilde{T}\}$ based on Theorem 5.2, and it can lead to the alternative regret bound of $R(T) \leq \widetilde{\mathcal{O}}(\widetilde{d}\sqrt{T}) + \mathcal{O}\big(\sqrt{T_{\min}\log(\delta^{-1})} + \sqrt{T_{\min}\xi_R}\big)$. Here, $\widetilde{T}$ is data-dependent, decreasing with smaller effective dimension $\widetilde{d}$ and NTK norm $S$. More categories chosen (i.e., larger $B$ values) can also reduce $\widetilde{T}$ by increasing the likelihood of selecting the optimal category.

# 6 Experiments

We conduct experiments on three real-world data sets with varying specifications. Meanwhile, our *nine* baseline algorithms are: (1) **Linear algorithms**: Lin-UCB (Li et al., 2010), Lin-UCB-Category (Li et al., 2010); (2) **Kernelized algorithms**: Kernel-UCB (Valko et al., 2013), KMTL-UCB (Deshmukh et al., 2017), Kernel-UCB-Ind (Deshmukh et al., 2017); and (3) **Neural algorithms** with principled exploration: Neural-UCB (Zhou et al., 2020), Neural-UCB-Category (Zhou et al., 2020), Neural-TS (Zhang et al., 2021), EE-Net (Ban et al., 2022). Here, Lin-UCB-Category and Neural-UCB-Category extend Linear-UCB (Chu et al., 2011) and Neural-UCB (Zhou et al., 2020) respectively, by incorporating category-aware embeddings (Qi et al., 2022), *allowing them to also leverage **arm category information***. Kernel-UCB-Ind refers to Kernel-UCB (Valko et al., 2013) under the *disjoint setting* (Li et al., 2010), where individual estimators are learned for each arm category. Detailed descriptions and source code link are provided in Appendix B.1.

## 6.1 Experiment Results on Three Real Data Sets

**MovieLens and Yelp Data Sets.** Our first data set is *MovieLens-20M* (Harper & Konstan, 2015). With genome-scores of user-specified tags, we choose the top 20 tags with the highest genome-score variance to create the movie features $\boldsymbol{p} \in \mathbb{R}^{d'}$ ($d' = 20$). The movies are classified into 19 categories based on their genres: $|\mathcal{C}| = 19$. We then obtain user features $\boldsymbol{u} \in \mathbb{R}^{d'}$ following existing works (Li et al., 2019; Ban et al., 2022; Qi et al., 2022; 2023), by applying singular value decomposition (SVD) to the user-movie interaction matrix. Then, 5,000 movies and 10,000 users with the most reviews are applied following existing works (e.g., Li et al. (2019)). With a user $u_t$ sampled from the 10,000 users at each time step $t$, we follow the idea of Generalized Matrix Factorization (GMF) (He et al., 2017; Zhou et al., 2021) to encode user information and $i$-th item into the arm context through concatenation $\widetilde{\boldsymbol{x}}_{c,t}^{(i)} = [\boldsymbol{u}_{u_t}; \boldsymbol{p}_i] \in \mathbb{R}^{2d'}, c \in \mathcal{C}_t, i \in [K_{c,t}]$, and let sampled arm pool size $K = |\mathcal{X}_t| = 1,000$. We then append a constant of 0.01 to $\widetilde{\boldsymbol{x}}_{c,t}^{(i)}$ to obtain $\boldsymbol{x}_{c,t}^{(i)} \in \mathbb{R}^d$, before applying $L_2$ normalization. Such a procedure leads to context dimensionality $d = 41$. Arm rewards $r_{c,t}^{(i)}$ are user ratings normalized into range $[0,1]$. For *Yelp data set* (https://www.yelp.com/dataset), SVD is analogously applied to extract user and item features with a hidden dimension of $d' = 20$. Then, we apply 10,000 users and 4,000 restaurants from 20 categories ($|\mathcal{C}| = 20$) with the highest number of reviews for candidate selection. We then apply the above GMF procedure to obtain arm contexts $\boldsymbol{x}_{c,t}^{(i)} \in \mathcal{X}_t \subset \mathbb{R}^d$ ($d = 41$, $|\mathcal{X}_t| = 1,000$), incorporating user and item information. Arm rewards are user ratings normalized to $[0,1]$.

**Amazon Data Set.** For Amazon recommendation data set (He & McAuley, 2016), we select the items from $|\mathcal{C}| = 10$ categories, and each user-item pair (arm) is associated with a review and a corresponding rating. To derive the arm context, we transform the review text into vector representations, by following the same text processing procedure outlined in the "Sentires" package (Zhang et al., 2014; Li et al., 2020). The obtained arm context maintains a dimensionality of $d = 41$, with the $L_2$ normalization applied on arm contexts. Similar to the Yelp and MovieLens data sets, we let $|\mathcal{X}_t| = 1,000$ in each round, by sampling from transformed user-item pairs (each treated as an arm); and we define the unknown arm reward as the normalized user rating scaled to range $[0,1]$. Complementary details are presented in Appendix B.1.

**Main Results.** On the left side of Fig. 1, $H_2$N-Bandit consistently outperforms all these strong baselines. The standard deviation of $H_2$N-Bandit is also plotted as the red shaded area. Among these baselines, the neural algorithms (Neural-UCB, EE-Net, Neural-TS) tend to outperform linear and kernel algorithms, due to the representation power of neural networks. In addition, by leveraging the category information, Neural-UCB-Category (Neu-Cat.) achieves better performance compared with the original Neural-UCB, on the MovieLens and Yelp data sets. In contrast, Linear-UCB-Category (Lin-UCB-Cat.) generally performs worse than the original Lin-UCB. This performance discrepancy is observed because the discrimination power of linear models is limited, and they can possibly fail to learn the underlying mapping as well as effectively

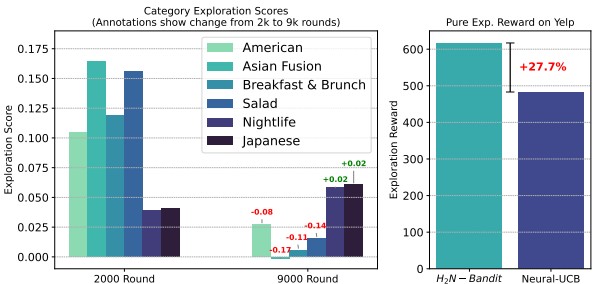

Figure 1: [Left Figure] Regret results on three recommendation data sets. [Right Figure] Regret vs. Running Time (Linear baselines symbols: ○; Kernel baselines: △; Neural baselines: □; H₂N-Bandit: ⋆).

leverage the category information encoded. Compared with Neural-UCB-Category, this observation further supports the notion that neural methods are better-suited solutions for utilizing available category information. Meanwhile, Kernel-UCB-Ind (Ker-Ind), which learns individual kernelized models for each category, suffers from the data sparsity issue and exhibits the worst performance among all the baselines. Our H₂N-Bandit outperforms the baselines by effectively balancing the category-level and arm-level exploitation-exploration. Meanwhile, H₂N-Bandit can take as little as 1/3 of the running time of other neural algorithms, which need to enumerate all the candidate arms for arm-level exploration. In addition, in Subsec. 6.4, we empirically demonstrate that H₂N-Bandit can also be extended to *three-level arm hierarchy* settings, which results in enhanced computational efficiency with only moderate performance degradation. This demonstrates the compatibility and scalability of our H₂N-Bandit framework, enabling practitioners to determine the specific arm hierarchy based on their own application scenarios and data set characteristics, in order to strike a good balance between computational costs and model performance.

**Performance vs. Running Time.** On the right side of Fig. 1, we compare with baselines regarding the model performance and running time. Each colored dot represents one method, where the methods (dots) in the lower left region are considered better in terms of both the running time and model performance. From the results, H₂N-Bandit achieves superior performance when compared to other baselines. Notably, H₂N-Bandit achieves competitive performance in terms of cumulative regret while requiring only one-third of the running time of Neural-UCB. This aligns with our claim that category-level recommendation can strike a good balance between the model performance and the computational cost. Furthermore, the kernelized methods also suffer from the high computational cost issue, because the kernel evaluation has to be performed between all the candidate arms and the past chosen arms, which can be extremely time-consuming.

## 6.2 Category-level Recommendation Analysis

Recall H₂N-Bandit applies networks $f_C^{(1)}(\cdot), f_C^{(2)}(\cdot)$ to balance exploitation and exploration in category-level recommendations. We further analyze its category-level exploration behavior on the Yelp data set.

In the left figure of Fig. 2, we compare the category exploration scores (i.e., outputs of $f_C^{(2)}$) for six specific arm categories. At round 2,000, H₂N-Bandit actively explores major categories like "American" restaurants, which have more arms and user interactions than minor categories such as "Japanese." Early in the recommendation process, this helps H₂N-Bandit build a strong global understanding for high-quality recommendations. By round 9,000, exploration scores shift toward minor categories, as major categories have already been sufficiently explored. Notably, the "Asian"

Figure 2: Exploration results on Yelp Data set. [Left Figure] The category-level exploration *score change* from 2000 round to 9000 round. [Right Figure] Exploration *cumulative reward* comparison showcasing the benefit of our category-level filtration.

category receives a negative score, since $f_C^{(2)}$ can produce negative values. This indicates H₂N-Bandit is compensating for overly high estimated rewards in that category, encouraging exploration of underrepresented categories for complementary information. Additionally, the average exploration score decreases over time, as the system gathers more interactions from the environment, gradually favoring exploitation. On the

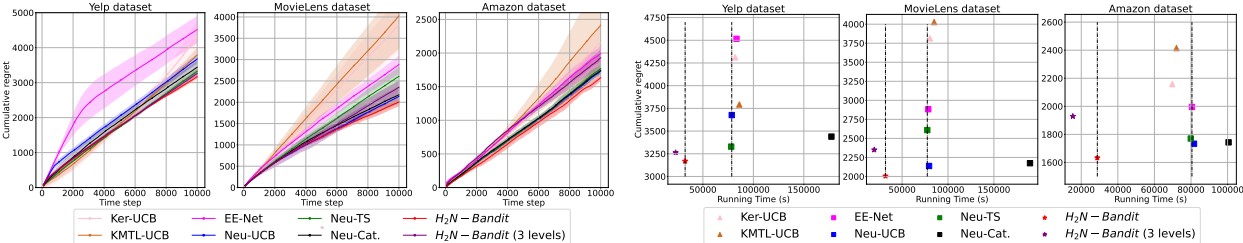

Figure 3: Regret (left three figures) and running time comparison (right three figures) for three-level H$_2$N-Bandit. [Kernel baselines: △; Neural baselines: □; H$_2$N-Bandit and H$_2$N-Bandit (3 levels): ⋆].

| Data set | $B = 2$ | $B = 4$ | $B = 6$ | $B = 8$ |
|---|---|---|---|---|
| Yelp | $3354_{\pm 26}$ | $3077_{\pm 41}$ | $3159_{\pm 50}$ | $2961_{\pm 33}$ |
| Amazon | $2062_{\pm 40}$ | $1623_{\pm 42}$ | $1608_{\pm 32}$ | $1753_{\pm 27}$ |

Table 1: Regret w.r.t. chosen category quantity $B$

| Data set | $\alpha = 0$ | $\alpha = 0.1$ | $\alpha = 0.3$ | $\alpha = 0.5$ | No Cat. Exp. |
|---|---|---|---|---|---|
| Yelp | $3212_{\pm 22}$ | $3128_{\pm 19}$ | $3077_{\pm 41}$ | $3253_{\pm 27}$ | $3345_{\pm 23}$ |
| Amazon | $1809_{\pm 36}$ | $1623_{\pm 42}$ | $1658_{\pm 23}$ | $1653_{\pm 47}$ | $1761_{\pm 27}$ |

Table 2: Regret w.r.t exploration coefficients $\alpha$

right side of Fig. 2, we compare with Neural-UCB in terms of exploration quality. Different from line 11 in Algorithm 1, we alternatively choose arm $\boldsymbol{x}_t, t \in [T]$ with "pure exploration", by only using the UCB: $\boldsymbol{x}_t = \arg\max_{\boldsymbol{x}_{c,t}^{(i)} \in \widetilde{\mathcal{X}}_t} \left[ \mathsf{UCB}_\alpha(\boldsymbol{x}_{c,t}^{(i)}) \right]$, where $\widetilde{\mathcal{X}}_t = \bigcup_{c \in \widetilde{\mathcal{C}}_t} \mathcal{X}_{c,t}$. Similarly, Neural-UCB will select arms solely based on UCB values. As H$_2$N-Bandit achieves a higher *cumulative reward*, its category-level recommendation contributes to improved exploration quality.

### 6.3 Parameter Study

Next, we present a parameter study for the number of selected categories, $B$ in Table 1, where the effect of $B$ is task-dependent. For the Yelp data set (20 total categories), a larger $B$ can improve performance by increasing the chance of selecting the optimal category, but we find $B = 4$ offers a good trade-off with computational cost. In contrast, for the Amazon data set (10 categories), a larger $B$ does not necessarily improve and can degrade performance by reducing filtering effectiveness, leading to unnecessary arm-level exploration. Thus, practitioners can tune $B$ starting from smaller values to balance performance and efficiency.

Empirical results with different exploration coefficient $\alpha$ are in Table 2, where setting $\alpha \in [0.1, 0.3]$ generally achieves satisfactory performance. It is worth noting that both no exploration ($\alpha = 0$) and excessively high exploration can negatively impact model performance. H$_2$N-Bandit, equipped with category-level exploration and a modest level of arm-level exploration, can effectively support arm recommendation. In addition, we include results for the setting with *no category-level exploration* by disabling $f_C^{(2)}$, demonstrating that equipping H$_2$N-Bandit with category-level exploration is essential for achieving optimal performance.

### 6.4 Extension: Adapting H$_2$N-Bandit to a Three-Level Arm Hierarchy

We also extend H$_2$N-Bandit to a three-level hierarchy (hyper-category, category, and arm), by grouping existing categories into *three hyper-categories* using $K$-means clustering on their mean arm contexts. To handle this structure, we introduce an additional hyper-category model analogous to our original category-level model; it uses a similar neural architecture, derives representations as in Eq. 4, and is trained on pseudo-labels generated by the category-level networks. The recommendation becomes a three-stage filtering process from hyper-category down to the final arm. We let $|\mathcal{X}_t| = 1,000, t \in [T]$, and compare with kernelized and neural baselines from Subsec. 6.1. Complementary details of the experimental settings are in Appendix B.2.

Based on the three line charts at the left of Fig. 3, incorporating a third-level (hyper-category-level) model generally results in a slight performance degradation in terms of cumulative regret across the three real data sets. For the Yelp data set, H$_2$N-Bandit (3 levels) is able to outperform kernel-based baselines and maintain comparable or close performance relative to strong neural baselines (e.g., Neural-UCB and Neural-TS). However, for the remaining two data sets, MovieLens and Amazon, its performance tends to be slightly inferior to these neural baselines. Regarding the running time results, H$_2$N-Bandit (3 levels) improves the efficiency of the two-level H$_2$N-Bandit by significantly reducing the number of candidate arms for arm-level selection through the hyper-category-level filtering. This demonstrates a trade-off between model performance and computational efficiency in terms of arm hierarchy levels. While H$_2$N-Bandit (3 levels) achieves the

shortest running time, our proposed $H_2N$-Bandit (2 levels, Algorithm 1) can help strike a better balance between model performance and computational cost.

## 7 Conclusion

We propose a novel neural bandit framework, $H_2N$-Bandit, to leverage the available arm category information in recommendation. It utilizes the bi-level hierarchical neural structure to tackle online recommendation tasks, along with principled exploration strategies to address the exploitation-exploration dilemma. By leveraging category-level recommendation, $H_2N$-Bandit can significantly reduce the computational cost by filtering the arms based on their categories, while simultaneously improving the arm-level recommendation. Under standard neural bandit assumptions and over-parameterized neural network settings, we perform regret analysis and provide the performance guarantee for the category-level recommendation. Extensive experiments are conducted to show the effectiveness of $H_2N$-Bandit against strong baselines, and the behaviors of $H_2N$-Bandit in terms of running time and category-level recommendation.

### Acknowledgments

This work is supported by National Science Foundation under Award No. IIS-2117902. The views and conclusions are those of the authors and should not be interpreted as representing the official policies of the funding agencies or the government.

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

**Broader Impact Statement**

This paper introduces $H_2N$-Bandit, a novel hierarchical framework that advances neural contextual bandits by making principled exploration computationally efficient for large-scale online recommendation. This enhancement improves the utility and performance of such systems by intelligently leveraging category-level information. While our work has broad applications, we do not foresee significant negative impacts. Potential concerns, such as the amplification of biases present in the data, are general to the field of recommender systems and not uniquely introduced by our method.

## A  Notation Table

In this section, to enhance the readability of readers, we include a table for the notation along with the corresponding descriptions.

| Notation - Description Table | | | |
|---|---|---|---|
| Notation | Description | Notation | Description |
| $T$ | Finite horizon | $K$ | Total number of candidate arms in each round |
| $\boldsymbol{x}$ | Arm context | $B$ | Number of selected arm categories |
| $\boldsymbol{\nu}$ | Category representation vector | $R(T)$ | Cumulative pseudo-regret for $T$ rounds |
| $r$ | Reward | $L$ | Network depth / number of layers |
| $d$ | Arm context dimension | $m$ | Network width / number of hidden units |
| $d_C$ | Category representation dimension | $\mathbf{M}$ | NTK Gram matrix with possibly duplicate arms |
| $\mathcal{A}_T, \breve{\mathcal{A}}_T$ | Arm collections / set for optimal and chosen arms | $\breve{\mathbf{M}}$ | NTK Gram matrix with non-duplicate arms |
| $\mathcal{C}$ | Collection of all categories | $\lambda$ | Regularization coefficient |
| $\mathcal{C}_t$ | Received categories collection (in round $t$) | $\epsilon$ | Reward random noise |
| $\widetilde{\mathcal{C}}_t$ | Selected categories collection (in round $t$) | $\alpha$ | Exploration coefficient |
| $\mathcal{X}_t$ | Collection of all candidate arms (in round $t$) | $J$ | Number of iterations for GD |
| $\mathcal{X}_{c,t}$ | Candidate arms from category $c$ (in round $t$) | $\breve{\lambda}_0$ | Minimum eigenvalue of the NTK matrix $\mathbf{M}$ |
| $\widetilde{\mathcal{X}}_t$ | Candidate arms from selected categories $\widetilde{\mathcal{C}}_t$ (in round $t$) | $R$ | Learning radius for category-level model |
| $h_A(\cdot)$ | Mapping function for arm reward | $\xi_R$ | Regression difficulty with the radius $R$ |
| $h_C(\cdot)$ | Mapping function for category reward | $S$ | NTK norm parameter |
| $f_C^{(1)}(\cdot)$ | Category-level exploitation network | $\delta$ | Probability parameter |
| $f_C^{(2)}(\cdot)$ | Category-level exploration network | $\eta_C$ | Category-level learning rate |
| $\mathcal{F}_C(\cdot)$ | Shorthand for category overall score $f_C^{(1)}(\cdot) + f_C^{(2)}(\cdot)$ | $\eta_A$ | Arm-level learning rate |
| $f_A(\cdot)$ | Arm-level reward estimation network | $\boldsymbol{\Sigma}$ | Gradient covariance matrix |
| $\boldsymbol{\Theta}$ | Trainable network parameters | $g(\cdot; \boldsymbol{\Theta})$ | Vector of network gradients w.r.t. parameters $\boldsymbol{\Theta}$ |
| $[\boldsymbol{\Theta}]_t$ | Network parameters trained by $t$-th round GD | $\mathcal{L}(\cdot)$ | Loss function |

Table 3: Notation Table

## B  Experiment Details and Complementary Experiments

### B.1  Experiment Details

Here, for all the UCB-based baselines, we choose their exploration parameter from the range $\{0.01, 0.1, 1\}$ with grid search. Following experiment settings from the original papers of our baselines (Zhou et al., 2020; Zhang et al., 2021; Ban et al., 2022), we set the number of layers $L = 2$ for all the deep learning models including our proposed $H_2N$-Bandit. We also set the network width with grid search $m \in \{100, 200, 300\}$. The learning rate of all neural algorithms are selected by grid search from the range $\{0.0001, 0.001, 0.01\}$. For all the algorithms, we choose the regularization parameter $\lambda$ from the range $\{0.0001, 0.001, 0.01\}$. Then, we choose the exploration coefficient $\alpha$ from the value range $\{0.2, 0.5, 1\}$ for $H_2N$-Bandit in terms of scaling. For kernelized baselines, we apply the radial basis function (RBF) kernel, and choose the kernel bandwidth from $\{0.01, 0.1, 1\}$. All experiments are performed on a server with Intel Xeon CPU and NVIDIA V100 GPUs. The results are based on the experiments across 3 runs. [1]

Here, we would like to add the URLs for the data sets and the data processing package applied in the experiments:

---

[1] Our code implementation is available: `https://drive.google.com/drive/folders/1z_OJuFYOzKBkamUVAgEChMVS6_NpVUhO?usp=sharing`

- MovieLens data set: `https://grouplens.org/datasets/movielens/20m/`

- Yelp Review data set: `https://www.yelp.com/dataset`

- Amazon data set: `https://jmcauley.ucsd.edu/data/amazon/index_2014.html`

- Sentires package for processing the Amazon data set: `https://github.com/lileipisces/Sentires-Guide`

In addition, we also would like to include more detailed descriptions for our nine baseline algorithms:

- Lin-UCB represents Linear-UCB under the *"pooling setting"* (Li et al., 2010) where it applies a single estimator for all arm categories.

- Lin-UCB-Category is Linear-UCB (Chu et al., 2011; Li et al., 2010) equipped with category-aware embedding (Qi et al., 2022), where the category-aware embedding can encode the available arm category information into the arm contexts.

- Kernel-UCB represents Kernel-UCB (Valko et al., 2013) under the *"pooling setting"* where it applies a single estimator for all arm categories.

- KMTL-UCB (Deshmukh et al., 2017) estimates the "task similarities" with received contextual information, by considering each arm category as a task. The estimations are based on a variant of kernel ridge regression.

- Kernel-UCB-Ind is Kernel-UCB (Valko et al., 2013) under the *"disjoint setting"* (Li et al., 2010) where it learns individual estimators for each arm category.

- Neural-UCB stands for Neural-UCB (Zhou et al., 2020) with a single neural network to evaluate the reward, and calculate the upper confidence bounds with the network gradients.

- Neural-UCB-Category refers to Neural-UCB (Zhou et al., 2020) with Neural-UCB with category-aware embedding (Qi et al., 2022) for utilizing the arm category information.

- Neural-TS (Zhang et al., 2021) utilizes a FC network to estimate the arm reward, along with a Thompson sampling (Agrawal & Goyal, 2013) exploration strategy.

- EE-Net (Ban et al., 2022) utilizes two FC networks for exploitation as well as exploration respectively, where the gradients of the first network is deemed as the input of the exploration (second) network.

### B.2 Three-level Arm Hierarchy

In Subsec. 6.4, we further extend the bi-level $H_2N$-Bandit to three-level hierarchy settings: (1) arm hyper-category level; (2) arm-category level; and (3) arm-level recommendation. Here, for each of the three data sets, we group the existing arm categories into three hyper-categories using K-means clustering, where the category feature for clustering is determined by averaging the arm contexts associated with that category. We also let $|\mathcal{X}_t| = 1,000, t \in [T]$, and compare with kernelized and neural baselines from Subsec. 6.1.

To extend $H_2N$-Bandit to the three-level hierarchy setting, we introduce an additional hyper-category-level model. Similar to the original category-level model $f_C$, the hyper-category-level model incorporates an exploitation neural network for hyper-category reward estimation and an exploration network for computing potential gains. Here, hyper-category representation is derived in a manner similar to Eq. 4, encoding both the mean arm context information and variance information. In each round, the process consists of: (1) hyper-category filtering, followed by (2) category-level selection (lines 7-9, Algorithm 1), and finally (3) arm-level recommendation (lines 11-14, Algorithm 1). Since the true hyper-category reward is not accessible, the hyper-category-level model is trained on pseudo-labels generated by the category-level networks, similar to the training process of the category-level models from Subsec. 4.1.

## C    Complementary Experiments

In this subsection, we present complementary experiments to further analysis H$_2$N-Bandit's behaviors and properties.

### C.1    Empirical Validation of Effective Dimension Reduction

By definition, we can first let $M_{TK}$ be the kernel Gram matrix formed from all $TK$ arms observed up to round $T$, and let $M_{3T}$ be the Gram matrix formed from three arms per round (chosen, oracle-best, best-within-chosen-categories).

After re-indexing, $M_{3T}$ is a principal sub-matrix of $M_{TK}$; hence, by interlacing, $\mu_i(M_{3T}) \leq \mu_i(M_{TK})$ for $i \leq 3T$. For any PSD matrix $M$ with eigenvalues $\{\mu_i(M)\}$ in non-increasing order and $g(u) = \log(1 + u/\lambda)$, we have $\log \det(I + M/\lambda) = \sum_i g(\mu_i(M))$. Applying this to $M_{3T}$ and $M_{TK}$ yields $\log \det(I + M_{3T}/\lambda) = \sum_{i=1}^{3T} g(\mu_i(M_{3T})) \leq \sum_{i=1}^{3T} g(\mu_i(M_{TK})) \leq \sum_{i=1}^{TK} g(\mu_i(M_{TK})) = \log \det(I + M_{TK}/\lambda)$, where the first inequality follows from $\mu_i(M_{3T}) \leq \mu_i(M_{TK})$ (since $M_{3T}$ is a principal sub-matrix) and the monotonicity of $g$, and the second inequality simply adds the nonnegative tail terms for $i > 3T$.

Afterwards, using the same denominator $\log(1 + TK/\lambda)$ for the normalized effective dimension then gives $\tilde{d}(M_{3T}, \lambda) \leq \tilde{d}(M_{TK}, \lambda)$, with equality only when the omitted arms add no new kernel directions under the kernel feature map.

On the other hand, Fig. 4 empirically validates this behavior on three datasets: the $3T$ curves are uniformly lower than the $TK$ curves for all $T$, grow more slowly, and typically saturate earlier. This indicates that adding only three representative arms per round introduces fewer novel directions in the kernel feature space than including all $K$ candidates, thereby reducing the used capacity without harming learning stability.

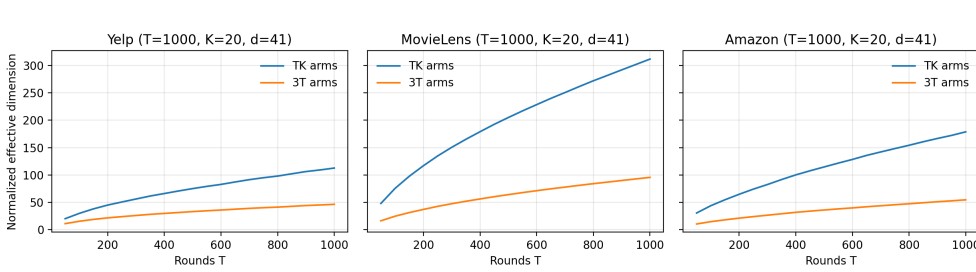

Figure 4: Normalized effective dimension vs. rounds $T$. Each subplot compares using all $TK$ arms per round (blue) with three arms per round ($3T$, orange). The $3T$ curves are uniformly lower and saturate earlier, corroborating $\tilde{d}(M_{3T}, \lambda) \leq \tilde{d}(M_{TK}, \lambda)$.

### C.2    Computational Complexity with Different Gradient Input Pooling Scales

To keep the exploration network efficient, we pool the gradients from the category scorer $f_C^{(1)}$ into a lower-dimensional vector before feeding them to $f_C^{(2)}$. This gives a simple handle, namely the pooling size, to control the input dimensionality of $f_C^{(2)}$ without changing the rest of the pipeline. We investigate a wide range of pooling sizes on Yelp and Amazon and report both runtime and regret as ratios to a fixed reference.

Fig. 5 shows that runtime stays generally flat across pooling sizes, while regret remains close to parity with the reference. In practice, this means $f_C^{(2)}$ is not the main computational bottleneck, and the dominant cost lies at the arm-level gradient-based exploration. Pooling therefore offers a safe and convenient way to cap the exploration network's input size and memory footprint, with negligible impact on effectiveness. A moderate pooling size is sufficient to retain useful gradient signal while keeping the system lightweight.

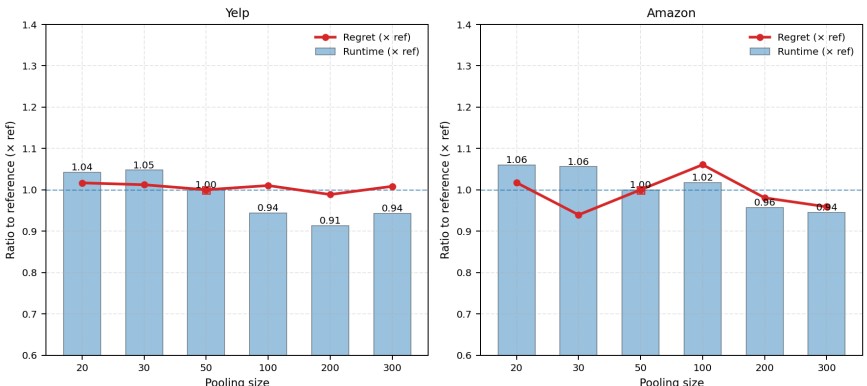

Figure 5: Pooling-size sensitivity on Yelp and Amazon. Runtime (bars) and regret (line) are shown as ratios to a fixed reference baseline (pooling size equals to 50). Varying the pooled gradient dimension fed to $f_C^{(2)}$ leaves runtime nearly unchanged and regret near parity, indicating pooling reduces input dimensionality without significant overhead or accuracy loss.

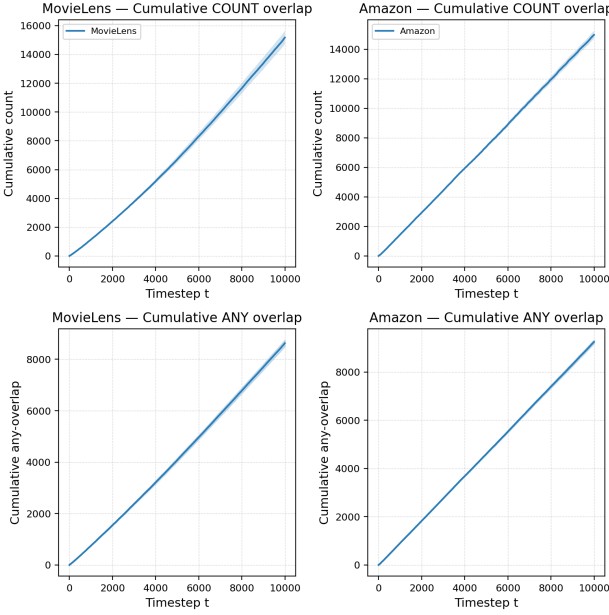

Figure 6: Category-level overlap trends. Top: cumulative count of overlapping categories with the oracle; Bottom: cumulative any-overlap (indicator of any match). Both measures increase over time, indicating rising hit rate and coverage in category selection.

## C.3 Category-level Selection Trends

Next, to assess how well the category stage focuses the arm search over time, we track two complementary signals: (i) the cumulative count of overlaps between the selected categories and the oracle-optimal categories (summing the number of matches at each step), and (ii) the cumulative any-overlap (adding 1 whenever at least one selected category matches the oracle at a step). Fig. 6 shows consistent upward trends for both measures on MovieLens and Amazon. The any-overlap curves grow consistently, indicating that the selector routinely includes at least one optimal category as $t$ increases. The count curves rise faster than the any-overlap curves, reflecting growing multi-category agreement with the oracle and thus broader coverage among the top choices. Together, these trends suggest that the category-level model improves its hit rate and coverage over time, providing a reliable, compact filter that guides arm-level exploration.

## C.4 Conservative Pseudo-labels for Non-selected Classes

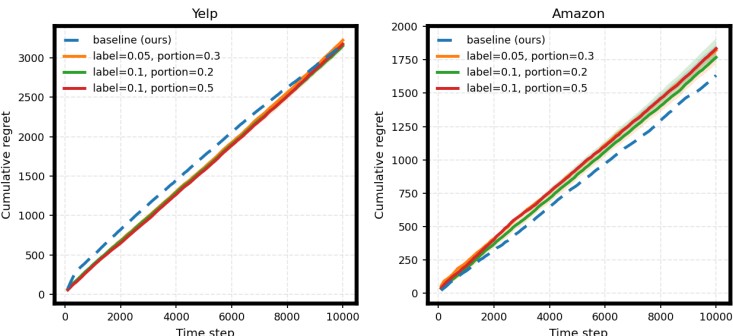

Figure 7: Conservative pseudo-labels on non-selected categories. Baseline (no conservative labels) vs. injecting small labels into a random subset of non-selected categories at each step (varying label magnitude and sampling portion).

We also evaluate a variant that injects conservative pseudo-labels into a random subset of non-selected categories at each time step. Concretely, after selecting categories, we sample a proportion of the remaining categories and assign them a small target value (e.g., label = 0.05 or 0.1) in place of the usual pseudo-reward, then update the category model with these targets. This aims to regularize exploration by softly encouraging coverage of unselected categories.

Fig. 7 summarizes the cumulative regret. On Amazon, all tested settings underperform the baseline and the gap widens over time, suggesting that conservative labels can potentially dilute informative signals and divert exploration toward randomly chosen categories. On the other hand, for Yelp, conservative labels provide a slight early gain, but converge to the baseline or worse, indicating no substantial benefit.

## C.5 Effects of Perturbed Category Representation

We evaluate robustness of the category stage by injecting zero-mean Gaussian noise into the arm contexts *before* category aggregation, thereby perturbing only the category feature (mean + std) while leaving the arm module and supervision unchanged. Concretely, for each round $t$ and category $c$, we add random noise drawn from $\mathcal{N}(0, \sigma^2 I)$ to the normalized arm contexts and vary the scale $\sigma \in \{0.05, 0.1, 0.2\}$.

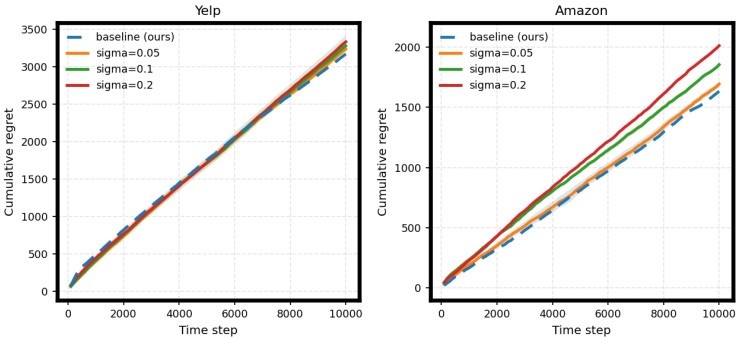

Figure 8: Robustness to category-representation noise: cumulative regret under random Gaussian perturbations applied to the inputs used for category aggregation, for $\sigma \in \{0.05, 0.1, 0.2\}$.

Fig. 8 shows that our bi-level architecture is generally stable under mild noise, with small gaps at $\sigma=0.05$. As $\sigma$ increases, performance can vary, and the sensitivity is dataset dependent: Yelp remains relatively resilient even at $\sigma=0.2$, whereas Amazon exhibits a rise in cumulative regret when injected strong perturbations. These results indicate that robustness to category-level noise can vary across domains and be application-specific.

Thus, developing dedicated denoised aggregation mechanisms to handle category-injected noise is a promising future direction of our work, to further strengthen the stability under adversarial settings.

### C.6 Augmenting Category Representation with Quantile and Top-k Features

We explore two tail-sensitive enrichment to the category representation.

**Quantile Category Feature.** To help capture rare but potentially high-value information within a category, we augment the category representation with a per-dimension quantile of the arm contexts. Let $\mathcal{X}_{c,t} = \{\mathbf{x}_1, \ldots, \mathbf{x}_{n_c}\} \subset \mathbb{R}^d$ denote the set of normalized arm contexts available for category $c$ at round $t$. In addition to the mean and standard deviation vectors, $\boldsymbol{\mu}_{c,t} = \frac{1}{n_c} \sum_{i=1}^{n_c} \mathbf{x}_i$, $\boldsymbol{\sigma}_{c,t} = \sqrt{\frac{1}{n_c} \sum_{i=1}^{n_c} (\mathbf{x}_i - \boldsymbol{\mu}_{c,t})^{\odot 2}}$, we compute a coordinate-wise $q$-quantile $\mathbf{q}_{c,t} \in \mathbb{R}^d$ with $q \in (0,1)$, where $[\mathbf{q}_{c,t}]_j$ is the $q$-quantile of $\{[\mathbf{x}_i]_j\}_{i=1}^{n_c}$. The resulting feature is the concatenation $\nu_{c,t}^{\text{quant}} = \text{norm}\left([\boldsymbol{\mu}_{c,t}; \boldsymbol{\sigma}_{c,t}; \mathbf{q}_{c,t}]\right)$, where $\text{norm}(\cdot)$ denotes $L_2$ normalization. Choosing a high quantile (we set $q = 0.7$) helps emphasize the upper tail of each coordinate distribution, making the representation more sensitive to rare but strong signals.

**Top-$k$ Category Feature.** We can also augment the representation with a pooled statistic over the top-$k$ arms according to a simple, permutation-invariant score. Given a scoring function $s(\cdot)$ (we apply coordinate variance), select the indices of the $k$ highest-scoring arms, $I_k = \arg\text{top-k}_{i \in [n_c]} s(\mathbf{x}_i)$, and compute the top-$k$ mean $\bar{\mathbf{x}}_{c,t}^{(k)} = \frac{1}{k} \sum_{i \in I_k} \mathbf{x}_i$. Concatenating this vector with the base statistics yields $\nu_{c,t}^{\text{topk}} = \text{norm}\left([\boldsymbol{\mu}_{c,t}; \boldsymbol{\sigma}_{c,t}; \bar{\mathbf{x}}_{c,t}^{(k)}]\right)$. This heuristic can highlight rare information-rich arms that might be washed out by the global mean, and we set $k = 5$.

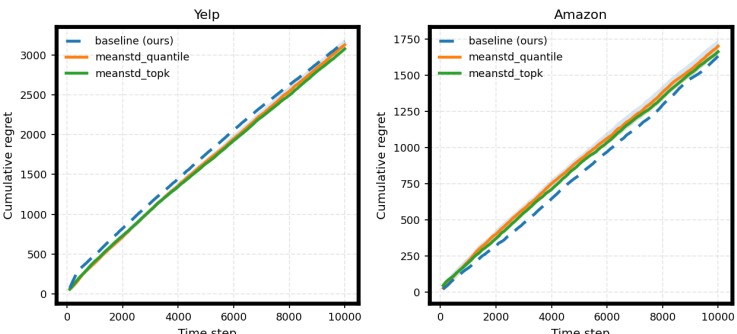

Figure 9: Quantile and top-$k$ category features vs. baseline, reported as cumulative regret. We compare the baseline (mean+std) with two augmentations that concatenate either a per-dimension high quantile or a top-$k$ pooled vector.

Fig. 9 compares cumulative regret against the baseline. On Yelp, both enrichments provide small, consistent gains; on Amazon, they slightly underperform the baseline. This suggests tail-aware features help when rare, high-magnitude arms carry actionable signal, but can add variance when tails are noisy. Given the dataset-dependent benefits, we retain our basic mean+std representation as our default, while view quantile/top-$k$ features as optional and data-dependent enhancements that the practitioner can choose based on their specific application scenarios.

## D Complementary Explanations for Category Reward Modeling and Definition

We first recall that in Remark 1, the computational complexity is $\mathcal{O}(|\mathcal{C}_t| + \sum_{c \in \tilde{\mathcal{C}}_t} |\mathcal{X}_{c,t}|)$, where $\tilde{\mathcal{C}}_t$ refers to the chosen categories with the cardinality $|\tilde{\mathcal{C}}_t| = B$. In this work, our efficiency improvement comes from the divide-and-conquer workflow based on the category information, which requires our category reward formulation (Eq. 3). Here, we will need our category reward definition to achieve the above time complexity, instead of the $\mathcal{O}(K)$ complexity in conventional neural bandit algorithms (Zhou et al., 2020; Zhang et al., 2021). This is to make sure the category-level model can adequately learn the relationship between category

representations and corresponding category rewards. Meanwhile, we would like to mention that it is common to introduce specific formulations in order to leverage the arm category information. For instance, (Hong et al., 2022) assumes a tree structure with known distribution priors, and (Deshmukh et al., 2017) assumes the reward mapping function is in the Reproducing Kernel Hilbert Space (RKHS) induced by the designated kernel. The effectiveness of our formulation for leveraging the arm category information is also supported by our experiment results, where we also include several baselines that are capable of leveraging the arm category information (e.g., Linear-UCB-Category, Neural-UCB-Category, KMTL-UCB).

In addition, we would like to provide additional explanations for our formulation of the category reward. Recall that under the stochastic contextual bandit settings (Zhou et al., 2020; Chu et al., 2011; Valko et al., 2013), in each round $t$, the learner aims to choose one arm $x_t \in \mathcal{X}_t$ with the maximum reward out of candidates $\mathcal{X}_t$ as its recommendation. Here, our definition of the category reward aims to comply with the learner's objective in each round $t \in [T]$. In this case, since the learner is only required to choose one arm in round $t$ with the maximum reward, the target of the category-level model will naturally be choosing $B$ arm categories $\tilde{\mathcal{C}}_t \subset \mathcal{C}_t$, so that these chosen categories contain the arms with highest rewards among all candidate arms. Therefore, our formulation helps align the category reward definition with our overall objective as well as the target of the arm-level model: recommending one arm with the highest reward out of the filtered candidate arms $\tilde{\mathcal{X}}_t$. In this case, we formulate the reward of the category as the maximum arm reward it contains. Meanwhile, from the theoretical analysis perspective, our category reward formulation also enables us to bridge the category-level model and the arm-level model. Otherwise, it will be implausible to make our regret analysis pipeline feasible.

On the other hand, if we consider some other formulations, the category reward can fail to comply with our objective and thus fail to capture the arm with the highest reward. For instance, let us alternatively consider the category reward to be the *average* reward of all the arms in this category. In this case, the category reward can intuitively fail to align with our overall learning objective: choosing one arm with the highest reward in each round. This is because selecting arm categories with the highest category rewards (e.g., average arm reward) does not necessarily guarantee the optimal arm is in the chosen categories, which can lead to sub-optimal performances.

As an example, suppose we are working on the online movie recommendation task, under the stochastic contextual bandit settings (Li et al., 2010; Valko et al., 2013; Zhou et al., 2020). Naturally, our objective is to select one movie $x_t \in \mathcal{X}_t$ with the maximum reward as our recommendation to the target user in each round. Here, suppose the candidate arms $\mathcal{X}_t$ are made up with two kinds of movies: action movies and comedy movies. Suppose the target user generally considers all the action movies to be average-to-good (arm rewards in this category: mean=0.6, variance=0.03). For the comedy movies, there are a few pieces that are target user's favorites, but the target user also dislikes some of the comedy movies, which makes arm rewards in "comedy" category: mean=0.5, variance=0.2. In this case, if we model the category reward to be the *average* reward of all the arms in this category, we will be more likely to choose action movies. This can consequently lead to sub-optimal results, even with a powerful arm-level model that can accurately estimate arm rewards. This is because the arms (movies) with the highest rewards, i.e., comedy movie pieces that the target user favorites, are located in the "comedy" category.

As a result, it helps justify our formulation that the goal of the category-level model should be choosing category that contains the arm with maximum reward, based on category reward definition (Eq. 3).

### D.1 Categorical distribution and model performances.

Here, our theoretical analysis, which involves the category-level error and the arm-level error and considers the worst-case scenario. In this case, different category distributions will not impair our theoretical results.

Recall that our objective is to choose the arm with *maximum reward* in each round. In this case, let us consider two cases: (1) a heavy-tailed category with one highest reward arm, while the other arms are of low rewards; (2) a category of arms with nearly averaged rewards. For case (1), it is obvious that the *reward margin* between this highest-reward arm and the other sub-optimal arms will be large. Thus, it would be easier for the arm-level model to distinguish this optimal arm for the arm-level recommendation. On the other hand, for case (2), the reward discrepancy among arms within this category will be small, which makes

it harder for the arm-level model to distinguish the best arm within this category. In this case, although the arm-level model may pull a sub-optimal arm, the single-round regret will be relatively smaller, because the reward difference between the category-level optimal arm and the chosen arm will be small, due to the nature of arm reward distribution within this category of case (2).

Therefore, there naturally exists a balance between (1) the arm-level reward estimation error and the single-round regret, as well as (2) the arm reward discrepancy within the same category. Meanwhile, as we have discussed, the category-level model naturally aims to choose $B$ arm categories that contain arms with *highest rewards* for arm-level recommendation. Thus, we formulate the category reward as the maximum arm reward it contains, to reflect this property. As the category-level and arm-level models will be co-optimized along with more rounds, they will become increasingly more accurate, in terms of both the category selection and arm selection.

We would like to mention that under the stochastic contextual bandit settings, the learner will only receive the arm reward *for the chosen arm* from the environment. In this case, the true arm category reward will not necessarily be available, no matter what the category reward definition is. Since the arm reward distribution within categories is unknown, this is also the exact reason why we train the category-model with the *pseudo-labels* (lines 17-19, Algorithm 1 pseudo-code). Therefore, the category reward definition in Eq. 3 can be considered as our way of leveraging the category information. Meanwhile, since we will only receive the reward for the chosen arm in each round, it can be impractical for the learner to gain a comprehensive view regarding the category distribution. In this case, we utilize our category reward definition and the bi-level neural architecture to strike a good balance between model performances and computational costs.

### D.2 The trade-off between the category-level recommendation and the arm-level recommendation.

There are also natural trade-offs between the category-level recommendation and the arm-level recommendation, in terms of (1) the performance trade-off; (2) the computational complexity trade-off. And they directly relate to the $B$ value (line 1, Algorithm 1 pseudo-code), which is the number of categories chosen in each round.

#### D.2.1 The performance trade-off between category-level recommendation and arm-level recommendation

In Table 1, we conduct experiments with different $B$ values, showing setting $B$ to high values does not necessarily lead to better performances. Here, although increasing the $B$ values can intuitively increase the possibility of choosing optimal categories, it can impose more burden for the arm-level model, which can lead to sub-optimal performances as for Amazon data set in Table 1. Similarly, setting $B$ to overly small values can also lead to unsatisfactory performances, because it limits the power of the arm-level model. Therefore, practitioners may need to choose $B$ based on their specific scenarios, to balance the computational cost and model performance.

#### D.2.2 Computational complexity and category-level identification trade-off

Recall that for Remark 1, our computational complexity is $\mathcal{O}(|\mathcal{C}_t| + \sum_{c \in \tilde{\mathcal{C}}_t} |\mathcal{X}_{c,t}|)$, where $\tilde{\mathcal{C}}_t$ refers to the chosen categories with the cardinality $|\tilde{\mathcal{C}}_t| = B$. Thus, with more chosen categories (higher $B$ values), it will lead to more candidate arms for arm-level recommendation ($\sum_{c \in \tilde{\mathcal{C}}_t} |\mathcal{X}_{c,t}|$ value), which consequently increases the computational complexity. On the other hand, as in Theorem 5.2, the optimal category is guaranteed to be chosen by the category-level model with a high probability, when we have $t \geq \tilde{T}$. Based on the definition of $\tilde{T}$, we see that $\tilde{T}$ will decrease along with increasingly higher $B$ values, which can lead to higher chances of including the optimal arm category for arm-level recommendation.

## E  Proof of Cumulative Regret Bound (Proof of Theorem 5.1)

Here, we denote $\mathcal{F}_C(\cdot) = f_C^{(1)}(\cdot) + f_C^{(2)}(\cdot)$ as well as $\mathcal{F}_A(\cdot) = f_A(\cdot) + \mathsf{UCB}_\alpha(\cdot)$, for the sake of notation simplicity. Recall that the learner is expected to choose one arm $\boldsymbol{x}_t \in \mathcal{X}_t$ in each round $t \in [T]$, in order to minimize the

cumulative pseudo-regret, denoted by

$$R(T) = \mathbb{E}[\sum_{t=1}^{T}(r_t^* - r_t)]$$

where we have $r_t$ referring to the reward of the chosen arm $\boldsymbol{x}_t \in \mathcal{X}_t$, and $r_t^*$ being the reward for the optimal arm, such that $\mathbb{E}[r_t^*|\mathcal{X}_t] = \max_{\boldsymbol{x}_{c,t}^{(i)} \in \mathcal{X}_t} h_A(\boldsymbol{x}_{c,t}^{(i)})$. Under the over-parameterized neural networks settings (Allen-Zhu et al., 2019; Cao & Gu, 2019), we can bound the approximation error of neural networks in terms of the reward mapping functions $h_A(\cdot), h_C(\cdot)$. Here, we aim to minimize the pseudo-regret for $T$ rounds, with $\boldsymbol{x}_t, \boldsymbol{x}_t^* \in \mathcal{X}_t$ being the chosen arm and the optimal arm in round $t \in [T]$, the regret bound $R(T)$ is

$$R(T) = \sum_{t=1}^{T} R_t = \sum_{t=1}^{T} \mathbb{E}_\epsilon \left[ r_t^* - r_t \right] = \sum_{t=1}^{T} \left[ h_A(\boldsymbol{x}_t^*) - h_A(\boldsymbol{x}_t) \right]$$

$$= \sum_{t=1}^{T} \left[ \min\{h_A(\boldsymbol{x}_t^*) - h_A(\boldsymbol{x}_t),\ 1\} \right]$$

where $R_t$ is the regret for a single round $t$, and the last equality is because the expected reward is bounded, $h_A(\boldsymbol{x}) \in [0, 1]$, in our problem definition. Then, defining $\widetilde{\boldsymbol{x}}_t^* = \arg\max_{\boldsymbol{x} \in \bigcup_{c \in \widetilde{\mathcal{C}}_t} \mathcal{X}_{c,t}} h(\boldsymbol{x})$ as the "best possibly chosen" arm conditioned on the first-stage selected categories $\widetilde{\mathcal{C}}_t$, we have

$$R_t = \min\{h_A(\boldsymbol{x}_t^*) - h_A(\boldsymbol{x}_t),\ 1\}$$
$$= \min\{h_A(\boldsymbol{x}_t^*) - h_A(\widetilde{\boldsymbol{x}}_t^*) + h_A(\widetilde{\boldsymbol{x}}_t^*) - h_A(\boldsymbol{x}_t),\ 1\}$$
$$= \min\{\underbrace{h_C(\boldsymbol{\nu}_{c_t^*,t}) - h_C(\boldsymbol{\nu}_{c_t,t})}_{\text{category-level error}} + \underbrace{h_A(\widetilde{\boldsymbol{x}}_t^*) - h_A(\boldsymbol{x}_t)}_{\text{arm-level error}},\ 1\}$$

where $c_t^*$ is the optimal arm category, $c_t$ is the "chosen optimal" category $c_t = \arg\max_{c \in \widetilde{\mathcal{C}}_t} h_C(\boldsymbol{\nu}_{c,t})$. Here, we see that the pseudo regret for each round $t \in [T]$ can be decomposed into two sub-components, namely the category-level error and the arm-level error.

Next, we first proceed to bound the category-level error with the following steps. For the sake of notation simplicity, for the derivation steps below, we hide the min-operation $\min\{\cdot,\ 1\}$ by default, which will lead to

$$\min\{h_C(\boldsymbol{\nu}_{c_t^*,t}) - h_C(\boldsymbol{\nu}_{c_t,t}),\ 1\}$$
$$= h_C(\boldsymbol{\nu}_{c_t^*,t}) - \mathcal{F}_C(\boldsymbol{\nu}_{c_t,t}) + \mathcal{F}_C(\boldsymbol{\nu}_{c_t,t}) - h_C(\boldsymbol{\nu}_{c_t,t})$$
$$\leq h_C(\boldsymbol{\nu}_{c_t^*,t}) - \mathcal{F}_C(\boldsymbol{\nu}_{c_t^*,t}) + \mathcal{F}_C(\boldsymbol{\nu}_{c_t,t}) - h_C(\boldsymbol{\nu}_{c_t,t})$$
$$\leq h_C(\boldsymbol{\nu}_{c_t^*,t}) - \mathcal{F}_A^*(\boldsymbol{x}_t^*) + \mathcal{F}_A^*(\boldsymbol{x}_t^*) - \mathcal{F}_C(\boldsymbol{\nu}_{c_t^*,t}) + \mathcal{F}_C(\boldsymbol{\nu}_{c_t,t}) - \mathcal{F}_A(\boldsymbol{x}_t) + \mathcal{F}_A(\boldsymbol{x}_t) - h_C(\boldsymbol{\nu}_{c_t,t})$$
$$\leq \left| h_C(\boldsymbol{\nu}_{c_t^*,t}) - \mathcal{F}_A^*(\boldsymbol{x}_t^*) \right| + \left| \mathcal{F}_A^*(\boldsymbol{x}_t^*) - \mathcal{F}_C(\boldsymbol{\nu}_{c_t^*,t}) \right| + \left| \mathcal{F}_C(\boldsymbol{\nu}_{c_t,t}) - \mathcal{F}_A(\boldsymbol{x}_t) \right| + \left| \mathcal{F}_A(\boldsymbol{x}_t) - h_C(\boldsymbol{\nu}_{c_t,t}) \right|$$
$$= \left| h_A(\boldsymbol{x}_t^*) - \mathcal{F}_A^*(\boldsymbol{x}_t^*) \right| + \left| \mathcal{F}_A^*(\boldsymbol{x}_t^*) - \mathcal{F}_C(\boldsymbol{\nu}_{c_t^*,t}) \right| + \left| \mathcal{F}_C(\boldsymbol{\nu}_{c_t,t}) - \mathcal{F}_A(\boldsymbol{x}_t) \right| + \left| \mathcal{F}_A(\boldsymbol{x}_t) - h_C(\boldsymbol{\nu}_{c_t,t}) \right|$$
$$\leq \left| h_A(\boldsymbol{x}_t^*) - \mathcal{F}_A^*(\boldsymbol{x}_t^*) \right| + \left| \mathcal{F}_A^*(\boldsymbol{x}_t^*) - \mathcal{F}_C(\boldsymbol{\nu}_{c_t^*,t}) \right| + \left| \mathcal{F}_C(\boldsymbol{\nu}_{c_t,t}) - \mathcal{F}_A(\boldsymbol{x}_t) \right| + \left| \mathcal{F}_A(\boldsymbol{x}_t) - h_A(\boldsymbol{x}_t) \right| + \left| h_A(\boldsymbol{x}_t) - h_C(\boldsymbol{\nu}_{c_t,t}) \right|$$
$$= \underbrace{\left| h_A(\boldsymbol{x}_t^*) - \mathcal{F}_A^*(\boldsymbol{x}_t^*) \right|}_{I_1} + \underbrace{\left| \mathcal{F}_A^*(\boldsymbol{x}_t^*) - \mathcal{F}_C(\boldsymbol{\nu}_{c_t^*,t}) \right|}_{I_2} + \underbrace{\left| \mathcal{F}_C(\boldsymbol{\nu}_{c_t,t}) - \mathcal{F}_A(\boldsymbol{x}_t) \right|}_{I_3} + \underbrace{\left| \mathcal{F}_A(\boldsymbol{x}_t) - h_A(\boldsymbol{x}_t) \right|}_{I_4} + \underbrace{\left| h_A(\boldsymbol{x}_t) - h_A(\widetilde{\boldsymbol{x}}_t^*) \right|}_{I_5}$$

where $\mathcal{F}_A^*$ refer to the imaginary arm-level model trained with optimal arms and their rewards $\{\boldsymbol{x}_\tau^*, r_\tau^*\}_{\tau \in [T]}$, and we also have $\widetilde{\boldsymbol{x}}_t^* = \arg\max_{\boldsymbol{x} \in \bigcup_{c \in \widetilde{\mathcal{C}}_t} \mathcal{X}_{c,t}} h_A(\boldsymbol{x})$ being the best arm within the filtered arm pool $\bigcup_{c \in \widetilde{\mathcal{C}}_t} \mathcal{X}_{c,t}$, i.e., the best arm within chosen arm categories. Here, the first inequality is due to the category selection mechanism, and the second inequality is due to the arm pulling mechanism $\mathcal{F}_A(\widetilde{\boldsymbol{x}}_t) \leq \mathcal{F}_A(\boldsymbol{x}_t)$, as well as $h_A(\boldsymbol{x}_t) \leq h_A(\widetilde{\boldsymbol{x}}_t) = h_C(\boldsymbol{\nu}_{c_t,t})$. As we have noticed, there are a total of five terms on the RHS that we need to bound one by one. Here, terms $I_1, I_4$ refer to the estimation error of our processed arm-level model and the imaginary arm-level model, which can be bounded by Lemma I.1 and Corollary I.2 respectively. Then,

terms $I_2, I_3$ refer to the approximation error of the category-level models, which measures how much the category-level model output deviates from the pseudo label. These two terms can be bounded by Lemma G.1, and Lemma G.2 respectively. With Lemma I.3, we can bound term $I_5$ being the reward discrepancy between the chosen arm $\boldsymbol{x}_t$ and the optimal arm within the chosen categories $\widetilde{\boldsymbol{x}}_t$.

Afterwards, for arm-level error $h_A(\widetilde{\boldsymbol{x}}_t^*) - h_A(\boldsymbol{x}_t)$, it is obvious that this error can be upper bounded by term $I_5$. Therefore, we can apply Lemma I.3 again to obtain the bound. Finally, by assembling all the results, we will have

$$
\begin{aligned}
R(T) = \sum_{t=1}^T R_t = \sum_{t=1}^T & \left[ \min\{h_A(\boldsymbol{x}_t^*) - h_A(\boldsymbol{x}_t),\ 1\} \right] \\
\leq \sum_{t=1}^T & \left[ \min \big\{ \underbrace{\left| h_A(\boldsymbol{x}_t^*) - \mathcal{F}_A^*(\boldsymbol{x}_t^*) \right|}_{I_1} + \underbrace{\left| \mathcal{F}_A^*(\boldsymbol{x}_t^*) - \mathcal{F}_C(\boldsymbol{\nu}_{c_t^*,t}) \right|}_{I_2} + \underbrace{\left| \mathcal{F}_C(\boldsymbol{\nu}_{c_t,t}) - \mathcal{F}_A(\boldsymbol{x}_t) \right|}_{I_3} \right. \\
& \left. + \underbrace{\left| \mathcal{F}_A(\boldsymbol{x}_t) - h_A(\boldsymbol{x}_t) \right|}_{I_4} + 2 \cdot \underbrace{\left| h_A(\boldsymbol{x}_t) - h_A(\widetilde{\boldsymbol{x}}_t^*) \right|}_{I_5},\ 1 \big\} \right]
\end{aligned}
$$

Then, substitute these terms with their upper bounds with the above lemmas, we will have

$$
\begin{aligned}
R(T) \leq \sum_{t=1}^T & \left[ \gamma_{t-1}(\alpha) \cdot \min\{\|g(\boldsymbol{x}_t; [\boldsymbol{\Theta}_A]_{t-1})/\sqrt{m}\|_{\boldsymbol{\Sigma}_{t-1}^{-1}},\ 1\} + \mathcal{O}(m^{-1/6}\sqrt{\log(m)}t^{2/3}\lambda^{-2/3}L^3) \right. \\
& + \mathcal{O}(Sm^{-1/6}\sqrt{\log(m)}t^{1/6}\lambda^{-1/6}L^{2/7}) \\
& + \gamma_{t-1}^*(\alpha) \cdot \min\{\|g(\boldsymbol{x}_t^*; [\boldsymbol{\Theta}_A^*]_{t-1})/\sqrt{m}\|_{(\boldsymbol{\Sigma}_{t-1}^*)^{-1}},\ 1\} + \mathcal{O}(m^{-1/6}\sqrt{\log(m)}t^{2/3}\lambda^{-2/3}L^3) \\
& + \mathcal{O}(Sm^{-1/6}\sqrt{\log(m)}t^{1/6}\lambda^{-1/6}L^{2/7}) \\
& + \gamma_{t-1}(\alpha) \cdot \min\{\|g(\boldsymbol{x}_t; [\boldsymbol{\Theta}_A]_{t-1})/\sqrt{m}\|_{\boldsymbol{\Sigma}_{t-1}^{-1}},\ 1\} + \mathcal{O}(m^{-1/6}\sqrt{\log(m)}t^{2/3}\lambda^{-2/3}L^3) \\
& \left. + \mathcal{O}(Sm^{-1/6}\sqrt{\log(m)}t^{1/6}\lambda^{-1/6}L^{2/7}) \right] \\
& + \mathcal{O}\big(\sqrt{T\log(\tfrac{1}{\delta})}\big) + \sqrt{T}\xi_R + \mathcal{O}\big(\frac{T^2LR^2}{\sqrt{m}} + \frac{T^2R^{4/3}L^2\sqrt{\log m}}{m^{1/3}}\big) \\
& + \mathcal{O}\big(\sqrt{T\log(\tfrac{1}{\delta})}\big) + \sqrt{T}\xi_R + \mathcal{O}\big(\frac{T^2LR^2}{\sqrt{m}} + \frac{T^2R^{4/3}L^2\sqrt{\log m}}{m^{1/3}}\big) + \mathcal{O}(TR^{4/3}m^{-1/3}L^2\sqrt{\log m}) \\
& + \mathcal{O}(T\xi_C R^{1/3}m^{-1/12}L^3\sqrt{\log m})
\end{aligned}
$$

Combining the terms, can further lead to

$$
\begin{aligned}
R(T) \leq \sum_{t=1}^T & \left[ 2\gamma_{t-1}(\alpha) \cdot \min\{\|g(\boldsymbol{x}_t; [\boldsymbol{\Theta}_A]_{t-1})/\sqrt{m}\|_{\boldsymbol{\Sigma}_{t-1}^{-1}},\ 1\} + \mathcal{O}(m^{-1/6}\sqrt{\log(m)}t^{2/3}\lambda^{-2/3}L^3) \right. \\
& + \mathcal{O}(Sm^{-1/6}\sqrt{\log(m)}t^{1/6}\lambda^{-1/6}L^{2/7}) \\
& + \gamma_{t-1}^*(\alpha) \cdot \min\{\|g(\boldsymbol{x}_t^*; [\boldsymbol{\Theta}_A^*]_{t-1})/\sqrt{m}\|_{(\boldsymbol{\Sigma}_{t-1}^*)^{-1}},\ 1\} + \mathcal{O}(m^{-1/6}\sqrt{\log(m)}t^{2/3}\lambda^{-2/3}L^3) \\
& \left. + \mathcal{O}(Sm^{-1/6}\sqrt{\log(m)}t^{1/6}\lambda^{-1/6}L^{2/7}) \right] \\
& + \mathcal{O}\big(\sqrt{T\log(\tfrac{1}{\delta})}\big) + 2\sqrt{T}\xi_R + \mathcal{O}\big(\frac{T^2LR^2}{\sqrt{m}} + \frac{T^2R^{4/3}L^2\sqrt{\log m}}{m^{1/3}}\big) + \mathcal{O}(TR^{4/3}m^{-1/3}L^2\sqrt{\log m}) \\
& + \mathcal{O}(T\xi_C R^{1/3}m^{-1/12}L^3\sqrt{\log m})
\end{aligned}
$$

Afterwards, by applying Lemma J.9, Lemma J.10 and Lemma J.2, it leads to

$$
R(T) \leq \mathcal{O}\Big( \sqrt{T\widetilde{d}\log(1+TK/\lambda)} \cdot \big(\alpha\sqrt{\widetilde{d}\log(1+TK/\lambda) - 2\log(\delta)} + \lambda^{1/2}S\big) \Big)
$$

$$
+ \mathcal{O}\big(\sqrt{T\log(\tfrac{1}{\delta})} + \sqrt{T\xi_R}\big) + \mathcal{O}\big(\frac{T^2LR^2}{\sqrt{m}} + \frac{T^2R^{4/3}L^2\sqrt{\log m}}{m^{1/3}}\big) + \mathcal{O}(TR^{4/3}m^{-1/3}L^2\sqrt{\log m})
$$

$$
+ \mathcal{O}(T\xi_C R^{1/3}m^{-1/12}L^3\sqrt{\log m})
$$

$$
+ \mathcal{O}(Sm^{-1/6}\sqrt{\log(m)}T^{1/6}\lambda^{-1/6}L^{2/7}) + \mathcal{O}(m^{-1/6}\sqrt{\log(m)}T^{2/3}\lambda^{-2/3}L^3)
$$

$$
\leq \mathcal{O}\Big( \sqrt{T\widetilde{d}\log(1+TK/\lambda)} \cdot \big(\alpha\sqrt{\widetilde{d}\log(1+TK/\lambda) - 2\log(\delta)} + \lambda^{1/2}S\big) \Big) + \mathcal{O}\big(\sqrt{T\log(\tfrac{1}{\delta})} + \sqrt{T\xi_R}\big) + \mathcal{O}(1)
$$

where the last inequality is because we have sufficiently large network width $m$, as mentioned in Theorem 5.1, which completes the proof.

## F  Guarantee for Choosing the Optimal Arm Category (Proof of theorem 5.2)

As mentioned in the main body, for the sake of theoretical analysis in this section, we suppose the candidate arms $\mathcal{X}_t$ in each round $t \in [T]$ are randomly sampled from the context distribution $\mathcal{D}$. Meanwhile, to derive the performance guarantee for the category-level model in each round, we consider the preference scores to satisfy the definition of the $\zeta$-margin among arm categories. Here, we note that this definition is necessary for defining the "optimal arm category" and deriving the performance guarantee for the category-level recommendation. Otherwise, consider a scenario where we receive $|\mathcal{C}_t| = 10$ categories all with the same category reward. In this case, there will be no need to perform category-level recommendation, since choosing arbitrary categories can lead to the highest category reward.

Recall that in each round, we will choose $|\widetilde{\mathcal{C}}_t| = B$ categories for the first-stage recommendation. With the definition on the category margin, if the optimal category falls into the chosen ones, i.e., $c_t^* \in \widetilde{\mathcal{C}}_t$, the discrimination power of the category-level model must be enough. If the category-level can definitely discriminate the optimal category $c_t^* \in \widetilde{\mathcal{C}}_t$, we can consider that for all the other arm categories, we need to make sure there is no overlap in terms of the confidence bounds with the optimal category. With a minimum category reward interval $\zeta$, denoting $\text{error}(c)$ to be the estimation error in terms of category $c$'s reward, we will have

$$
\mathbb{E}_{\mathcal{X}_t \sim \mathcal{D}}\Big[\text{error}(c_t^*) + \text{error}(c_t)\Big] \leq \zeta \implies \mathbb{E}\Big[\big|(h_C(\boldsymbol{\nu}_{c_t^*,t}) - \mathcal{F}_C(\boldsymbol{\nu}_{c_t^*,t}))\big| + \big|(h_C(\boldsymbol{\nu}_{c_t,t}) - \mathcal{F}_C(\boldsymbol{\nu}_{c_t,t}))\big|\Big] \leq \zeta
$$

$$
\implies \mathbb{E}\Big[\big|(h_A(\boldsymbol{x}_t^*) - \mathcal{F}_C(\boldsymbol{\nu}_{c_t^*,t})) + (h_A(\widetilde{\boldsymbol{x}}_t^*) - \mathcal{F}_C(\boldsymbol{\nu}_{c_t,t}))\big|\Big] \leq \zeta
$$

$$
\implies \mathbb{E}\Big[\big|(h_A(\boldsymbol{x}_t^*) - \mathcal{F}_C(\boldsymbol{\nu}_{c_t^*,t})) + (h_A(\widetilde{\boldsymbol{x}}_t^*) - \mathcal{F}_C(\boldsymbol{\nu}_{c_t,t}))\big|\Big] \leq \mathbb{E}\Big[\big|h_A(\boldsymbol{x}_t^*) - \mathcal{F}_C(\boldsymbol{\nu}_{c_t^*,t})\big| + \big|h_A(\widetilde{\boldsymbol{x}}_t^*) - \mathcal{F}_C(\boldsymbol{\nu}_{c_t,t})\big|\Big]
$$

$$
\leq \zeta
$$

By only considering the relationship between the chosen category and the optimal category, while omitting the expectation $\mathbb{E}[\cdot]$ notation for simplicity, we can satisfy the condition above by enabling

$$
\zeta \geq \big|h_A(\boldsymbol{x}_t^*) - \mathcal{F}_A^*(\boldsymbol{x}_t^*)\big| + \big|\mathcal{F}_A^*(\boldsymbol{x}_t^*) - \mathcal{F}_C(\boldsymbol{\nu}_{c_t^*,t})\big| + \big|h_A(\widetilde{\boldsymbol{x}}_t^*) - h_A(\widetilde{\boldsymbol{x}}_t)\big| + \big|h_A(\widetilde{\boldsymbol{x}}_t) - \mathcal{F}_A(\boldsymbol{x}_t)\big| + \big|\mathcal{F}_A(\boldsymbol{x}_t) - \mathcal{F}_C(\boldsymbol{\nu}_{c_t,t})\big|
$$

$$
\geq \big|h_A(\boldsymbol{x}_t^*) - \mathcal{F}_A^*(\boldsymbol{x}_t^*) + \mathcal{F}_A^*(\boldsymbol{x}_t^*) - \mathcal{F}_C(\boldsymbol{\nu}_{c_t^*,t})\big| + \big|h_A(\widetilde{\boldsymbol{x}}_t^*) - h_A(\widetilde{\boldsymbol{x}}_t) + h_A(\widetilde{\boldsymbol{x}}_t) - \mathcal{F}_A(\boldsymbol{x}_t) + \mathcal{F}_A(\boldsymbol{x}_t) - \mathcal{F}_C(\boldsymbol{\nu}_{c_t,t})\big|
$$

$$
= \big|h_A(\boldsymbol{x}_t^*) - \mathcal{F}_A^*(\boldsymbol{x}_t^*) + \mathcal{F}_A^*(\boldsymbol{x}_t^*) - \mathcal{F}_C(\boldsymbol{\nu}_{c_t^*,t})\big| + \big|h_A(\widetilde{\boldsymbol{x}}_t^*) - \mathcal{F}_A(\boldsymbol{x}_t) + \mathcal{F}_A(\boldsymbol{x}_t) - \mathcal{F}_C(\boldsymbol{\nu}_{c_t,t})\big|
$$

$$
\geq \big|h_A(\boldsymbol{x}_t^*) - \mathcal{F}_C(\boldsymbol{\nu}_{c_t^*,t})\big| + \big|h_A(\widetilde{\boldsymbol{x}}_t^*) - \mathcal{F}_C(\boldsymbol{\nu}_{c_t,t})\big|.
$$

Furthermore, with $B = |\widetilde{\mathcal{C}}_t|$ tasks chosen in round $t$ as well as the notation from Section E, for $c_t \in \widetilde{\mathcal{C}}_t$, and we can further relax our objective, because as long as our overall estimation error is smaller than $B \cdot \zeta$, we

will be able to include $c_t^*$ to $\widetilde{\mathcal{C}}_t$. This consequently leads to

$$\mathbb{E}\bigg[\underbrace{\big|h_A(\boldsymbol{x}_t^*) - \mathcal{F}_A^*(\boldsymbol{x}_t^*)\big|}_{I_1} + \underbrace{\big|\mathcal{F}_A^*(\boldsymbol{x}_t^*) - \mathcal{F}_C(\boldsymbol{\nu}_{c_t^*,t})\big|}_{I_2} + \underbrace{\big|\mathcal{F}_C(\boldsymbol{\nu}_{c_t,t}) - \mathcal{F}_A(\boldsymbol{x}_t)\big|}_{I_3} + \underbrace{\big|\mathcal{F}_A(\boldsymbol{x}_t) - h_A(\boldsymbol{x}_t)\big|}_{I_4} + \underbrace{\big|h_A(\boldsymbol{x}_t) - h_A(\widetilde{\boldsymbol{x}}_t^*)\big|}_{I_5}\bigg]$$
$$\leq B \cdot \zeta$$

where there are a total of five terms on the LHS that we need to bound one by one. Here, terms $I_1, I_4$ refer to the estimation error of our processed arm-level model and the imaginary arm-level model, which can be bounded by Lemma I.1 and Corollary I.2 respectively. Then, terms $I_2, I_3$ refer to the approximation error of the category-level models, which measures how much the category-level model output deviates from the pseudo label. These two terms can be bounded by Corollary G.3, Corollary G.4 respectively. With Lemma I.3, we can bound term $I_5$ being the reward discrepancy between the chosen arm $\boldsymbol{x}_t$ and the optimal arm within the chosen categories $\widetilde{\boldsymbol{x}}_t$. Afterwards, apply the upper bound for these terms, we will have

$$\begin{aligned}
B \cdot \zeta \geq\ & 2\gamma_T(\alpha) \cdot \min\{\|g(\boldsymbol{x}_t;[\boldsymbol{\Theta}_A]_{t-1})/\sqrt{m}\|_{\boldsymbol{\Sigma}_{t-1}^{-1}},\ 1\} + \gamma_T^*(\alpha) \cdot \min\{\|g(\boldsymbol{x}_t^*;[\boldsymbol{\Theta}_A^*]_{t-1})/\sqrt{m}\|_{(\boldsymbol{\Sigma}_{t-1}^*)^{-1}},\ 1\} \\
& + \mathcal{O}(Sm^{-1/6}\sqrt{\log(m)}t^{1/6}\lambda^{-1/6}L^{2/7}) + \mathcal{O}(m^{-1/6}\sqrt{\log(m)}t^{2/3}\lambda^{-2/3}L^3) \\
& + \mathcal{O}\Big(\sqrt{\tfrac{1}{t}\log(\tfrac{1}{\delta})}\Big) + \sqrt{\tfrac{\xi_R}{t}} + \mathcal{O}\Big(\tfrac{tLR^2}{\sqrt{m}} + \tfrac{tR^{4/3}L^2\sqrt{\log m}}{m^{1/3}}\Big) \\
& + \mathcal{O}(tR^{4/3}m^{-1/3}L^2\sqrt{\log m}) + \mathcal{O}(t\xi_C R^{1/3}m^{-1/12}L^3\sqrt{\log m}) \\
\geq\ & 2\gamma_{t-1}(\alpha) \cdot \min\{\|g(\boldsymbol{x}_t;[\boldsymbol{\Theta}_A]_{t-1})/\sqrt{m}\|_{\boldsymbol{\Sigma}_{t-1}^{-1}},\ 1\} + \gamma_{t-1}^*(\alpha) \cdot \min\{\|g(\boldsymbol{x}_t^*;[\boldsymbol{\Theta}_A^*]_{t-1})/\sqrt{m}\|_{(\boldsymbol{\Sigma}_{t-1}^*)^{-1}},\ 1\} \\
& + \mathcal{O}(Sm^{-1/6}\sqrt{\log(m)}t^{1/6}\lambda^{-1/6}L^{2/7}) + \mathcal{O}(m^{-1/6}\sqrt{\log(m)}t^{2/3}\lambda^{-2/3}L^3) \\
& + \mathcal{O}\Big(\sqrt{\tfrac{1}{t}\log(\tfrac{1}{\delta})}\Big) + \sqrt{\tfrac{\xi_R}{t}} + \mathcal{O}\Big(\tfrac{tLR^2}{\sqrt{m}} + \tfrac{tR^{4/3}L^2\sqrt{\log m}}{m^{1/3}}\Big) \\
& + \mathcal{O}(tR^{4/3}m^{-1/3}L^2\sqrt{\log m}) + \mathcal{O}(t\xi_C R^{1/3}m^{-1/12}L^3\sqrt{\log m}).
\end{aligned}$$

Then, for the $\min\{\|g(\boldsymbol{x}_t;[\boldsymbol{\Theta}_A]_{t-1})/\sqrt{m}\|_{\boldsymbol{\Sigma}_{t-1}^{-1}},\ 1\}$, $\min\{\|g(\boldsymbol{x}_t^*;[\boldsymbol{\Theta}_A^*]_{t-1})/\sqrt{m}\|_{(\boldsymbol{\Sigma}_{t-1}^*)^{-1}},\ 1\}$ terms, since we have the candidate arms $\mathcal{X}_t$ drawn in an i.i.d. manner, we can follow an analogous approach as in the proof of Corollary G.3. By constructing the martingale difference sequence and applying the Azuma-Hoeffding inequality, we will have

$$\begin{aligned}
& \mathbb{E}\big[\min\{\|g(\boldsymbol{x}_t;[\boldsymbol{\Theta}_A]_{t-1})/\sqrt{m}\|_{\boldsymbol{\Sigma}_{t-1}^{-1}},\ 1\}\big] = \frac{1}{t}\sum_{\tau\in[t]}\mathbb{E}\big[\min\{\|g(\boldsymbol{x}_\tau;[\boldsymbol{\Theta}_A]_{t-1})/\sqrt{m}\|_{\boldsymbol{\Sigma}_{t-1}^{-1}},\ 1\}\big] \\
& \leq \frac{1}{t}\sum_{\tau\in[t]}\mathbb{E}\big[\min\{\|g(\boldsymbol{x}_\tau;[\boldsymbol{\Theta}_A]_{t-1})/\sqrt{m}\|_{\boldsymbol{\Sigma}_{\tau-1}^{-1}},\ 1\}\big] \\
& \leq \frac{1}{t}\sum_{\tau\in[t]}\mathbb{E}\big[\min\{\|g(\boldsymbol{x}_\tau;[\boldsymbol{\Theta}_A]_{\tau-1})/\sqrt{m}\|_{\boldsymbol{\Sigma}_{\tau-1}^{-1}},\ 1\}\big] + \mathcal{O}(m^{-1/6}\sqrt{\log(m)}t^{2/3}\lambda^{-2/3}L^3) \\
& \leq \frac{1}{t}\sum_{\tau\in[t]}\big|\min\{\|g(\boldsymbol{x}_\tau;[\boldsymbol{\Theta}_A]_{\tau-1})/\sqrt{m}\|_{\boldsymbol{\Sigma}_{\tau-1}^{-1}},\ 1\}\big| + \mathcal{O}\Big(\sqrt{\tfrac{2}{t}\log(\tfrac{1}{\delta})}\Big) + \mathcal{O}(m^{-1/6}\sqrt{\log(m)}t^{2/3}\lambda^{-2/3}L^3) \\
& \leq \mathcal{O}\sqrt{\tfrac{\widetilde{d}}{t}\log(1+tK/\lambda)} + \mathcal{O}\Big(\sqrt{\tfrac{2}{t}\log(\tfrac{1}{\delta})}\Big) + \mathcal{O}(m^{-1/6}\sqrt{\log(m)}t^{2/3}\lambda^{-2/3}L^3)
\end{aligned}$$

where the first inequality is because of Corollary 7.7.4. (a) from (Meyer, 2000), second inequality is because of Lemma H.4, and the last inequality is based on Lemma J.9, Lemma J.10 and Lemma J.2. Similar bound can also be obtained for term $\min\{\|g(\boldsymbol{x}_t^*;[\boldsymbol{\Theta}_A^*]_{t-1})/\sqrt{m}\|_{(\boldsymbol{\Sigma}_{t-1}^*)^{-1}},\ 1\}$.

Finally, by assembling all the results above, we can have

$$
B \cdot \zeta \geq \mathcal{O}\left(\sqrt{\frac{\widetilde{d}}{t}} \log(1 + tK/\lambda) \cdot \left(\alpha\sqrt{\widetilde{d}\log(1 + tK/\lambda) - 2\log(\delta)} + \lambda^{1/2}S\right)\right) + \mathcal{O}(Sm^{-1/6}\sqrt{\log(m)}t^{1/6}\lambda^{-1/6}L^{2/7})
$$

$$
+ \mathcal{O}(m^{-1/6}\sqrt{\log(m)}t^{2/3}\lambda^{-2/3}L^3)
$$

$$
+ \mathcal{O}\left(\sqrt{\frac{1}{t}\log(\frac{1}{\delta})} + \sqrt{\frac{\xi_R}{t}}\right) + \mathcal{O}\left(\frac{tLR^2}{\sqrt{m}} + \frac{tR^{4/3}L^2\sqrt{\log m}}{m^{1/3}}\right)
$$

$$
+ \mathcal{O}(tR^{4/3}m^{-1/3}L^2\sqrt{\log m}) + \mathcal{O}(t\xi_C R^{1/3}m^{-1/12}L^3\sqrt{\log m}).
$$

Then, with sufficiently large network width $m$ indicated in Theorem 5.1, we can finally have

$$
t \geq \left(\frac{\mathcal{O}\left(\sqrt{\widetilde{d}\log(1 + tK/\lambda)} \cdot \left(\alpha\sqrt{\widetilde{d}\log(1 + tK/\lambda) - 2\log(\delta)} + \lambda^{1/2}S\right)\right) + \mathcal{O}\left(\sqrt{\log(\delta^{-1})} + \sqrt{\xi_R}\right) + \mathcal{O}(1)}{B \cdot \zeta}\right)^2.
$$

# G   Bounding Estimation Error for the Category-level Model

Recall that the category-level models $f_C^{(1)}, f_C^{(2)}$ are trained with the output of the arm-level model $f_A$. As shown in the main body, this will lead to the error term $\min\{|\mathcal{F}_C(\boldsymbol{\nu}_{c_t,t}) - \mathcal{F}_A(\boldsymbol{x}_t)|, 1\}$, which bound the output difference between the category-level model and the arm-level model. In order to derive the $T$-round regret $R(T)$, after accumulating this error term for $T$ rounds, we need to bound the following two terms

$$
\sum_{t\in[T]} \mathbb{E}\left[\min\{|\mathcal{F}_C(\boldsymbol{\nu}_{c_t,t}) - \mathcal{F}_A(\boldsymbol{x}_t)|, 1\} \,\Big|\, \mathcal{X}_t\right], \quad \sum_{t\in[T]} \mathbb{E}\left[\min\{|\mathcal{F}_A^*(\boldsymbol{x}_t^*) - \mathcal{F}_C(\boldsymbol{\nu}_{c_t^*,t})|, 1\} \,\Big|\, \mathcal{X}_t\right]
$$

where we recall $\mathcal{F}_C(\boldsymbol{\nu}_{c_t,t}) = f_C^{(1)}(\boldsymbol{\nu}_{c_t,t}; [\boldsymbol{\Theta}_C^{(1)}]_{t-1}) + f_C^{(2)}(\nabla f_C^{(1)}(\boldsymbol{\nu}_{c_t,t}); [\boldsymbol{\Theta}_C^{(2)}]_{t-1})$ represents the estimated score for the chosen category $c_t$, while on the other hand, we have $\mathcal{F}_A(\boldsymbol{x}_t) = f_A(\boldsymbol{x}_t; [\boldsymbol{\Theta}_A]_{t-1}) + \mathsf{UCB}_\alpha(\boldsymbol{x}_t)$ being the score from the arm-level model. Here, the first objective measures the approximation error of the category-level model $\mathcal{F}_C$, which the second term measures the error of the imaginary category-level model that has been trained on optimal arm categories. Then, we will proceed to introduce the following results.

**Lemma G.1.** *Given past collected records, we suppose $m, \eta_C, J$ satisfy the conditions in Theorem 5.1, and randomly draw the parameter $[\boldsymbol{\Theta}_C^{(1)}]_t, [\boldsymbol{\Theta}_C^{(2)}]_t \sim \{[\widehat{\boldsymbol{\Theta}}_C^{(1)}]_\tau, [\widehat{\boldsymbol{\Theta}}_C^{(2)}]_\tau\}_{\tau\in[t-1]}$. Consider the past records $\mathcal{P}_t = \{\boldsymbol{\nu}_{c_\tau,\tau}, \widetilde{r}_{c_\tau,\tau}\}_{\tau\in[t-1]}$ up to round $t$ are generated by a policy. Then, with $\xi_R > 0$ and the probability at least $1 - \delta$, given a pair of arm category and its pseudo reward $\{\boldsymbol{\nu}_{c_t,t}, \widetilde{r}_{c_t,t}\}$, we have*

$$
\sum_{\tau\in[t]} \mathbb{E}\left[\min\left\{|f_C^{(1)}(\boldsymbol{\nu}_{c_\tau,\tau}; [\boldsymbol{\Theta}_C^{(1)}]_{\tau-1}) + f_C^{(2)}(\nabla f_C^{(1)}(\boldsymbol{\nu}_{c_\tau,\tau}); [\boldsymbol{\Theta}_C^{(2)}]_{\tau-1}) - \widetilde{r}_{c_\tau,\tau}|, 1\right\} \,\Big|\, \mathcal{X}_\tau\right]
$$

$$
\leq \mathcal{O}\left(\sqrt{t\log(\frac{1}{\delta})}\right) + \sqrt{t\xi_R} + \mathcal{O}\left(\frac{t^2LR^2}{\sqrt{m}} + \frac{t^2R^{4/3}L^2\sqrt{\log m}}{m^{1/3}}\right)
$$

*where $\widetilde{r}_{c_\tau,\tau}$ is the pseudo reward generated by the arm-level model $\mathcal{F}_A(\cdot)$, as defined in Algorithm 1.*

**Proof.** We prove this Lemma following a similar approach as in Lemma C.1 from (Ban et al., 2022). First, for the LHS, we have

$$
\left|f_C^{(1)}(\boldsymbol{\nu}_{c_t,t}; [\boldsymbol{\Theta}_C^{(1)}]_{t-1}) + f_C^{(2)}(\nabla f_C^{(1)}(\boldsymbol{\nu}_{c_t,t}); [\boldsymbol{\Theta}_C^{(2)}]_{t-1}) - \widetilde{r}_{c_t,t}\right|
$$

$$
\leq \left|f_C^{(1)}(\boldsymbol{\nu}_{c_t,t}; [\boldsymbol{\Theta}_C^{(1)}]_{t-1})\right| + \left|f_C^{(2)}(\nabla f_C^{(1)}(\boldsymbol{\nu}_{c_t,t}); [\boldsymbol{\Theta}_C^{(2)}]_{t-1})\right| + \left|\widetilde{r}_{c_t,t}\right| \leq \mathcal{O}(1)
$$

based on the conclusion from Lemma H.1, as well as the value range for pseudo reward $\widetilde{r}$. Then, we define the following martingale difference sequence with regard to the previous records up to round $\tau \in [t]$, as

$$
V_\tau = \mathbb{E}\left[|f_C^{(1)}(\boldsymbol{\nu}_{c_\tau,\tau}; [\boldsymbol{\Theta}_C^{(1)}]_{\tau-1}) + f_C^{(2)}(\nabla f_C^{(1)}(\boldsymbol{\nu}_{c_\tau,\tau}); [\boldsymbol{\Theta}_C^{(2)}]_{\tau-1}) - \widetilde{r}_{c_\tau,\tau}|\right]
$$

$$
- |f_C^{(1)}(\boldsymbol{\nu}_{c_\tau,\tau}; [\boldsymbol{\Theta}_C^{(1)}]_{\tau-1}) + f_C^{(2)}(\nabla f_C^{(1)}(\boldsymbol{\nu}_{c_\tau,\tau}); [\boldsymbol{\Theta}_C^{(2)}]_{\tau-1}) - \widetilde{r}_{c_\tau,\tau}|.
$$

Since the records in set $\mathcal{P}_\tau$ are sharing the same reward mapping function, we have the expectation

$$\mathbb{E}[V_\tau|F_\tau] = \mathbb{E}\big[|f_C^{(1)}(\boldsymbol{\nu}_{c_\tau,\tau};[\boldsymbol{\Theta}_C^{(1)}]_{\tau-1}) + f_C^{(2)}(\nabla f_C^{(1)}(\boldsymbol{\nu}_{c_\tau,\tau});[\boldsymbol{\Theta}_C^{(2)}]_{\tau-1}) - \widetilde{r}_{c_\tau,\tau}|\big]$$
$$-\mathbb{E}\big[|f_C^{(1)}(\boldsymbol{\nu}_{c_\tau,\tau};[\boldsymbol{\Theta}_C^{(1)}]_{\tau-1}) + f_C^{(2)}(\nabla f_C^{(1)}(\boldsymbol{\nu}_{c_\tau,\tau});[\boldsymbol{\Theta}_C^{(2)}]_{\tau-1}) - \widetilde{r}_{c_\tau,\tau}|\big|F_\tau\big] = 0.$$

where $F_\tau$ denotes the filtration given the past records. Then, we can have the mean value of $V_\tau$ across different time steps being

$$\frac{1}{t}\sum_{\tau\in[t]}[V_\tau] = \frac{1}{t}\sum_{\tau\in[t]}\mathbb{E}\big[|f_C^{(1)}(\boldsymbol{\nu}_{c_\tau,\tau};[\boldsymbol{\Theta}_C^{(1)}]_{\tau-1}) + f_C^{(2)}(\nabla f_C^{(1)}(\boldsymbol{\nu}_{c_\tau,\tau});[\boldsymbol{\Theta}_C^{(2)}]_{\tau-1}) - \widetilde{r}_{c_\tau,\tau}|\big]$$
$$-\frac{1}{t}\sum_{\tau\in[t]}|f_C^{(1)}(\boldsymbol{\nu}_{c_\tau,\tau};[\boldsymbol{\Theta}_C^{(1)}]_{\tau-1}) + f_C^{(2)}(\nabla f_C^{(1)}(\boldsymbol{\nu}_{c_\tau,\tau});[\boldsymbol{\Theta}_C^{(2)}]_{\tau-1}) - \widetilde{r}_{c_\tau,\tau}|.$$

with the expectation of zero. Then, we proceed to bound the expected estimation error of the exploitation model with the estimation error from existing samples. Afterwards, by applying the Azuma-Hoeffding inequality, with a constant $\delta \in (0,1)$, we have

$$\mathbb{P}\Big[\frac{1}{t}\sum_{\tau\in[t]}\mathbb{E}\big[|f_C^{(1)}(\boldsymbol{\nu}_{c_\tau,\tau};[\boldsymbol{\Theta}_C^{(1)}]_{\tau-1}) + f_C^{(2)}(\nabla f_C^{(1)}(\boldsymbol{\nu}_{c_\tau,\tau});[\boldsymbol{\Theta}_C^{(2)}]_{\tau-1}) - \widetilde{r}_{c_\tau,\tau}|\big]$$
$$-\frac{1}{t}\sum_{\tau\in[t]}|f_C^{(1)}(\boldsymbol{\nu}_{c_\tau,\tau};[\boldsymbol{\Theta}_C^{(1)}]_{\tau-1}) + f_C^{(2)}(\nabla f_C^{(1)}(\boldsymbol{\nu}_{c_\tau,\tau});[\boldsymbol{\Theta}_C^{(2)}]_{\tau-1}) - \widetilde{r}_{c_\tau,\tau}| \geq \mathcal{O}(1)\cdot\sqrt{\frac{2}{t}\log(\frac{1}{\delta})}\Big] \leq \delta.$$

As we have the parameter $[\boldsymbol{\Theta}_C^{(1)}]_t, [\boldsymbol{\Theta}_C^{(2)}]_t \sim \{[\widehat{\boldsymbol{\Theta}}_C^{(1)}]_\tau, [\widehat{\boldsymbol{\Theta}}_C^{(2)}]_\tau\}_{\tau\in[t]}$, with the probability at least $1-\delta$, the expected loss on $[\boldsymbol{\Theta}_C^{(1)}]_t$ could be bounded as

$$\frac{1}{t}\sum_{\tau\in[t]}\mathbb{E}\big[|f_C^{(1)}(\boldsymbol{\nu}_{c_\tau,\tau};[\boldsymbol{\Theta}_C^{(1)}]_{\tau-1}) + f_C^{(2)}(\nabla f_C^{(1)}(\boldsymbol{\nu}_{c_\tau,\tau});[\boldsymbol{\Theta}_C^{(2)}]_{\tau-1}) - \widetilde{r}_{c_\tau,\tau}|\big]$$
$$\leq \frac{1}{t}\sum_{\tau\in[t]}|f_C^{(1)}(\boldsymbol{\nu}_{c_\tau,\tau};[\boldsymbol{\Theta}_C^{(1)}]_{\tau-1}) + f_C^{(2)}(\nabla f_C^{(1)}(\boldsymbol{\nu}_{c_\tau,\tau});[\boldsymbol{\Theta}_C^{(2)}]_{\tau-1}) - \widetilde{r}_{c_\tau,\tau}| + \mathcal{O}(1)\cdot\sqrt{\frac{2}{t}\log(\frac{1}{\delta})}$$

where for the second term on the RHS, we have

$$\frac{1}{t}\sum_{\tau\in[t]}|f_C^{(1)}(\boldsymbol{\nu}_{c_\tau,\tau};[\boldsymbol{\Theta}_C^{(1)}]_{\tau-1}) + f_C^{(2)}(\nabla f_C^{(1)}(\boldsymbol{\nu}_{c_\tau,\tau});[\boldsymbol{\Theta}_C^{(2)}]_{\tau-1}) - \widetilde{r}_{c_\tau,\tau}|$$
$$\leq \frac{1}{\sqrt{t}}\sqrt{\sum_{\tau\in[t]}|f_C^{(1)}(\boldsymbol{\nu}_{c_\tau,\tau};[\boldsymbol{\Theta}_C^{(1)}]_{\tau-1}) + f_C^{(2)}(\nabla f_C^{(1)}(\boldsymbol{\nu}_{c_\tau,\tau});[\boldsymbol{\Theta}_C^{(2)}]_{\tau-1}) - \widetilde{r}_{c_\tau,\tau}|^2}$$
$$+ \mathcal{O}(1) + \frac{tLR^2}{\sqrt{m}} + \mathcal{O}(\frac{tR^{4/3}L^2\sqrt{\log m}}{m^{1/3}})$$
$$\leq \sqrt{\frac{\xi_R}{t}} + \mathcal{O}(\frac{tLR^2}{\sqrt{m}} + \frac{tR^{4/3}L^2\sqrt{\log m}}{m^{1/3}})$$

where with the notation in Lemma H.7 and Lemma H.6, and we ca further have the definition for the first term being $\xi_R = \inf_{\boldsymbol{\Theta}_C\in\mathcal{B}([\boldsymbol{\Theta}_C]_0,\ Rm^{-\frac{1}{4}})}\big[\frac{1}{2}\sum_{\tau\in[t]}|f_C^{(1)}(\boldsymbol{\nu}_{c_\tau,\tau};\boldsymbol{\Theta}_C^{(1)}) + f_C^{(2)}(\nabla f_C^{(1)}(\boldsymbol{\nu}_{c_\tau,\tau});\boldsymbol{\Theta}_C^{(2)}) - \widetilde{r}_{c_\tau,\tau}|^2\big] > 0$, with $[\boldsymbol{\Theta}_C]_0$ be the initial parameters from either of the two category-level networks. Here, the inequality is the application of Lemma H.7. Summing up all the components would complete the proof.

$\square$

Meanwhile, apart from the sequence of chosen categories $\{\boldsymbol{\nu}_{c_t,t}\}_{t\in[T]}$, we can also have comparable results for other sequences of categories. Next, we will introduce the lemma in terms of the optimal categories in order to prove the bound for our second objective.

**Lemma G.2.** *Given past collected records, we suppose $m, \eta_C, J$ satisfy the conditions in Theorem 5.1, and randomly draw the parameter $[\mathbf{\Theta}_C^{(1)}]_t, [\mathbf{\Theta}_C^{(2)}]_t \sim \{[\widehat{\mathbf{\Theta}}_C^{(1)}]_\tau, [\widehat{\mathbf{\Theta}}_C^{(2)}]_\tau\}_{\tau \in [t-1]}$. Consider the past records $\mathcal{P}_t = \{\boldsymbol{\nu}_{c_\tau^*,\tau}, \widetilde{r}_{c_\tau^*,\tau}^*\}_{\tau \in [t-1]}$ up to round $t$ are generated by a policy. Then, with probability at least $1 - \delta$, given a pair of arm category and its pseudo reward $\{\boldsymbol{\nu}_{c_t,t}, \widetilde{r}_{c_t,t}\}$, we have*

$$\sum_{\tau \in [t]} \mathbb{E}\left[ \min\left\{ |f_C^{(1)}(\boldsymbol{\nu}_{c_\tau^*,\tau}; [\mathbf{\Theta}_C^{(1)}]_{\tau-1}) + f_C^{(2)}(\nabla f_C^{(1)}(\boldsymbol{\nu}_{c_\tau^*,\tau}); [\mathbf{\Theta}_C^{(2)}]_{\tau-1}) - \widetilde{r}_{c_\tau^*,\tau}^*|, \ 1 \right\} \ \bigg| \ \mathcal{X}_\tau \right]$$

$$\leq \mathcal{O}\left(\sqrt{t \log(\tfrac{1}{\delta})}\right) + \sqrt{t \xi_R} + \mathcal{O}\left(\frac{t^2 L R^2}{\sqrt{m}} + \frac{t^2 R^{4/3} L^2 \sqrt{\log m}}{m^{1/3}}\right)$$

$$+ \mathcal{O}(t R^{4/3} m^{-1/3} L^2 \sqrt{\log m}) + \mathcal{O}(t \xi_C R^{1/3} m^{-1/12} L^3 \sqrt{\log m})$$

*where $\widetilde{r}_{c_\tau^*,\tau}^*$ is the pseudo reward generated by the imaginary arm-level model $\mathcal{F}_A^*(\cdot)$, which is trained by optimal arms $\{\boldsymbol{x}_\tau^*, r_\tau^*\}_{\tau \in [t]}$.*

**Proof.** The proof of this lemma follows an analogous approach as in Lemma G.1. Recall that our processed category networks $f_C^{(1)}(\cdot; [\mathbf{\Theta}_C^{(1)}]), f_C^{(2)}(\cdot; [\mathbf{\Theta}_C^{(2)}])$ are trained by the records related to the chosen arms and their rewards $\{\boldsymbol{x}_\tau, r_\tau\}_{\tau \in [t]}$. Here, we first consider the alternative imaginary category-level models $f_C^{(1),*}(\cdot; [\mathbf{\Theta}_C^{(1),*}]), f_C^{(2),*}(\cdot; [\mathbf{\Theta}_C^{(2),*}])$, and arm-level model $f_A^*(\cdot; [\mathbf{\Theta}_A^*])$. These imaginary models are trained by the records optimal arms as well as optimal arm categories.

Similar to Lemma G.1, based on Lemma H.1, we have the LHS of the objective bounded by $\mathcal{O}(1)$. Then, we define the following martingale difference sequence with regard to the previous records up to round $\tau \in [t]$ in terms of the optimal arms and categories, such that

$$V_\tau^* = \mathbb{E}\big[ \big| f_C^{(1),*}(\boldsymbol{\nu}_{c_\tau,\tau}; [\mathbf{\Theta}_C^{(1),*}]_{\tau-1}) + f_C^{(2),*}(\nabla f_C^{(1),*}(\boldsymbol{\nu}_{c_\tau,\tau}); [\mathbf{\Theta}_C^{(2),*}]_{\tau-1}) - \widetilde{r}_{c_\tau^*,\tau}^* \big| \big]$$

$$- \big| f_C^{(1),*}(\boldsymbol{\nu}_{c_\tau,\tau}; [\mathbf{\Theta}_C^{(1),*}]_{\tau-1}) + f_C^{(2),*}(\nabla f_C^{(1),*}(\boldsymbol{\nu}_{c_\tau,\tau}); [\mathbf{\Theta}_C^{(2),*}]_{\tau-1}) - \widetilde{r}_{c_\tau^*,\tau}^* \big|.$$

where we also have its expectation w.r.t. the filtration $F_\tau^*$, $\mathbb{E}[V_\tau^* | F_\tau^*] = 0$. Next, by applying the Azume-Hoeffing inequality, with $1 > \delta > 0$, we can also have

$$\mathbb{P}\bigg[ \frac{1}{t} \sum_{\tau \in [t]} \mathbb{E}\big[ \big| f_C^{(1),*}(\boldsymbol{\nu}_{c_\tau,\tau}; [\mathbf{\Theta}_C^{(1),*}]_{\tau-1}) + f_C^{(2),*}(\nabla f_C^{(1),*}(\boldsymbol{\nu}_{c_\tau,\tau}); [\mathbf{\Theta}_C^{(2),*}]_{\tau-1}) - \widetilde{r}_{c_\tau^*,\tau}^* \big| \big]$$

$$- \frac{1}{t} \sum_{\tau \in [t]} \big| f_C^{(1),*}(\boldsymbol{\nu}_{c_\tau,\tau}; [\mathbf{\Theta}_C^{(1),*}]_{\tau-1}) + f_C^{(2),*}(\nabla f_C^{(1),*}(\boldsymbol{\nu}_{c_\tau,\tau}); [\mathbf{\Theta}_C^{(2),*}]_{\tau-1}) - \widetilde{r}_{c_\tau^*,\tau}^* \big| \geq \mathcal{O}(1) \cdot \sqrt{\frac{2}{t} \log(\tfrac{1}{\delta})} \bigg] \leq \delta.$$

After applying Lemma H.6 and Lemma H.7, we will end up with

$$\frac{1}{t} \sum_{\tau \in [t]} \big| f_C^{(1),*}(\boldsymbol{\nu}_{c_\tau^*,\tau}; [\mathbf{\Theta}_C^{(1),*}]_{\tau-1}) + f_C^{(2),*}(\nabla f_C^{(1),*}(\boldsymbol{\nu}_{c_\tau^*,\tau}); [\mathbf{\Theta}_C^{(2),*}]_{\tau-1}) - \widetilde{r}_{c_\tau^*,\tau}^* \big|$$

$$\leq \frac{1}{\sqrt{t}} \sqrt{\sum_{\tau \in [t]} \big| f_C^{(1),*}(\boldsymbol{\nu}_{c_\tau^*,\tau}; [\mathbf{\Theta}_C^{(1),*}]_{\tau-1}) + f_C^{(2),*}(\nabla f_C^{(1),*}(\boldsymbol{\nu}_{c_\tau^*,\tau}); [\mathbf{\Theta}_C^{(2),*}]_{\tau-1}) - \widetilde{r}_{c_\tau^*,\tau}^* \big|^2}$$

$$+ \mathcal{O}(1) + \frac{t L R^2}{\sqrt{m}} + \mathcal{O}\left(\frac{t R^{4/3} L^2 \sqrt{\log m}}{m^{1/3}}\right)$$

$$\leq \sqrt{\frac{\xi_R}{t}} + \mathcal{O}(1) + \frac{t L R^2}{\sqrt{m}} + \mathcal{O}\left(\frac{t R^{4/3} L^2 \sqrt{\log m}}{m^{1/3}}\right).$$

where the last inequality is because the parameter sphere $\mathcal{B}([\mathbf{\Theta}_C]_0, \ R m^{-1/4})$ also includes the imaginary model parameters. Noticing that the current bound is built upon the imaginary category-models, we then need to further extend this bound to our processed category-level models.

With Lemma H.6, we know that the processed parameters $[\boldsymbol{\Theta}_C^{(1)}]_{\tau-1}, [\boldsymbol{\Theta}_C^{(2)}]_{\tau-1}$ as well as the imaginary ones both fall into $\mathcal{B}([\boldsymbol{\Theta}_C]_0, \ Rm^{-1/4})$. As a result, we can have

$$\left| f_C^{(1),*}(\boldsymbol{\nu}_{c_\tau^*,\tau}; [\boldsymbol{\Theta}_C^{(1),*}]_{\tau-1}) + f_C^{(2),*}(\nabla f_C^{(1),*}(\boldsymbol{\nu}_{c_\tau^*,\tau}); [\boldsymbol{\Theta}_C^{(2),*}]_{\tau-1}) - f_C^{(1)}(\boldsymbol{\nu}_{c_\tau^*,\tau}; [\boldsymbol{\Theta}_C^{(1)}]_{\tau-1}) + f_C^{(2)}(\nabla f_C^{(1)}(\boldsymbol{\nu}_{c_\tau^*,\tau}); [\boldsymbol{\Theta}_C^{(2)}]_{\tau-1}) \right|$$

$$\leq \left| f_C^{(1),*}(\boldsymbol{\nu}_{c_\tau^*,\tau}; [\boldsymbol{\Theta}_C^{(1),*}]_{\tau-1}) - f_C^{(1)}(\boldsymbol{\nu}_{c_\tau^*,\tau}; [\boldsymbol{\Theta}_C^{(1)}]_{\tau-1}) \right|$$

$$+ \left| f_C^{(2),*}(\nabla f_C^{(1),*}(\boldsymbol{\nu}_{c_\tau^*,\tau}); [\boldsymbol{\Theta}_C^{(2),*}]_{\tau-1}) - f_C^{(2)}(\nabla f_C^{(1)}(\boldsymbol{\nu}_{c_\tau^*,\tau}); [\boldsymbol{\Theta}_C^{(2)}]_{\tau-1}) \right|$$

$$\leq \mathcal{O}(Rm^{-1/4}\sqrt{L}) + \mathcal{O}(R^{4/3}m^{-1/3}L^2\sqrt{\log m})$$

$$+ \left| f_C^{(2),*}(\nabla f_C^{(1),*}(\boldsymbol{\nu}_{c_\tau^*,\tau}); [\boldsymbol{\Theta}_C^{(2),*}]_{\tau-1}) - f_C^{(2)}(\nabla f_C^{(1)}(\boldsymbol{\nu}_{c_\tau^*,\tau}); [\boldsymbol{\Theta}_C^{(2)}]_{\tau-1}) \right|$$

where the inequality is by applying Lemma H.3. Then, for the last term on the RHS, supposing the input gradients are normalized by a coefficient $\frac{1}{\mathcal{O}(mL)}$, we will have

$$\left| f_C^{(2),*}(\nabla f_C^{(1),*}(\boldsymbol{\nu}_{c_\tau^*,\tau}); [\boldsymbol{\Theta}_C^{(2),*}]_{\tau-1}) - f_C^{(2)}(\nabla f_C^{(1)}(\boldsymbol{\nu}_{c_\tau^*,\tau}); [\boldsymbol{\Theta}_C^{(2)}]_{\tau-1}) \right|$$

$$\leq \left| f_C^{(2),*}(\nabla f_C^{(1),*}(\boldsymbol{\nu}_{c_\tau^*,\tau}); [\boldsymbol{\Theta}_C^{(2),*}]_{\tau-1}) - f_C^{(2)}(\nabla f_C^{(1),*}(\boldsymbol{\nu}_{c_\tau^*,\tau}); [\boldsymbol{\Theta}_C^{(2)}]_{\tau-1}) \right|$$

$$+ \left| f_C^{(2)}(\nabla f_C^{(1),*}(\boldsymbol{\nu}_{c_\tau^*,\tau}); [\boldsymbol{\Theta}_C^{(2)}]_{\tau-1}) - f_C^{(2)}(\nabla f_C^{(1)}(\boldsymbol{\nu}_{c_\tau^*,\tau}); [\boldsymbol{\Theta}_C^{(2)}]_{\tau-1}) \right|$$

$$\leq \mathcal{O}(Rm^{-1/4}\sqrt{L}) + \mathcal{O}(R^{4/3}m^{-1/3}L^2\sqrt{\log m})$$

$$+ \left| f_C^{(2)}(\nabla f_C^{(1),*}(\boldsymbol{\nu}_{c_\tau^*,\tau}); [\boldsymbol{\Theta}_C^{(2)}]_{\tau-1}) - f_C^{(2)}(\nabla f_C^{(1)}(\boldsymbol{\nu}_{c_\tau^*,\tau}); [\boldsymbol{\Theta}_C^{(2)}]_{\tau-1}) \right|$$

$$\leq \mathcal{O}(Rm^{-1/4}\sqrt{L}) + \mathcal{O}(R^{4/3}m^{-1/3}L^2\sqrt{\log m}) + \mathcal{O}(\xi_C R^{1/3}m^{-1/12}L^3\sqrt{\log m})$$

where the first inequality is by the triangular inequality, the second inequality is because of Lemma H.3, and the final inequality is because of the fact that over-parameterized FC networks are Lipshitz continuous w.r.t. some constant $\xi_C > 0$ (Allen-Zhu et al., 2019) as well as Lemma H.4.

$$\square$$

In addition, apart from the performance guarantee for the $T$ rounds altogether, we can further have the performance guarantee in terms of each $t \in [T]$ round, if each candidate arm is drawn from a distribution in an i.i.d. manner. This can be done by extending the conclusion as well as the proof workflow from Lemma G.1 and Lemma G.2.

**Corollary G.3.** *Given past collected records, we suppose $m, \eta_C, J$ satisfy the conditions in Theorem 5.1, and randomly draw the parameter $[\boldsymbol{\Theta}_C^{(1)}]_t, [\boldsymbol{\Theta}_C^{(2)}]_t \sim \{[\widehat{\boldsymbol{\Theta}}_C^{(1)}]_\tau, [\widehat{\boldsymbol{\Theta}}_C^{(2)}]_\tau\}_{\tau \in [t-1]}$. Consider the arms within candidate arm pools $\{\mathcal{X}_\tau\}_{\tau \in [t]}$ are drawn from a distribution $\mathcal{D}$ in the i.i.d. manner, and the past records $\mathcal{P}_t = \{\boldsymbol{\nu}_{c_\tau,\tau}, \widetilde{r}_{c_\tau,\tau}\}_{\tau \in [t-1]}$ up to round $t$ are generated by a policy. Then, with probability at least $1 - \delta$, given a pair of arm category and its pseudo reward $\{\boldsymbol{\nu}_{c_t,t}, \widetilde{r}_{c_t,t}\}$, we have*

$$\mathbb{E}\left[ \min\left\{ |f_C^{(1)}(\boldsymbol{\nu}_{c_t,t}; [\boldsymbol{\Theta}_C^{(1)}]_{t-1}) + f_C^{(2)}(\nabla f_C^{(1)}(\boldsymbol{\nu}_{c_t,t}); [\boldsymbol{\Theta}_C^{(2)}]_{t-1}) - \widetilde{r}_{c_t,t}|, \ 1 \right\} \ \middle| \ \mathcal{X}_t \right]$$

$$\leq \mathcal{O}\left(\sqrt{\frac{1}{t}\log(\frac{1}{\delta})}\right) + \sqrt{\frac{\xi_R}{t}} + \mathcal{O}\left(\frac{tLR^2}{\sqrt{m}} + \frac{tR^{4/3}L^2\sqrt{\log m}}{m^{1/3}}\right)$$

*where $\widetilde{r}_{c_t,t}$ is the pseudo reward generated by the arm-level model $\mathcal{F}_A(\cdot)$, as defined in Algorithm 1.*

**Proof.** The proof of this corollary follows a comparable approach as in Lemma G.1. Similarly, based on the conclusion from Lemma H.1 and the value range for pseudo reward $\widetilde{r}$, we define the following martingale difference sequence with regard to the previous records up to round $\tau \in [t]$, as

$$V_\tau = \mathbb{E}\left[ |f_C^{(1)}(\boldsymbol{\nu}_{c_\tau,\tau}; [\boldsymbol{\Theta}_C^{(1)}]_{\tau-1}) + f_C^{(2)}(\nabla f_C^{(1)}(\boldsymbol{\nu}_{c_\tau,\tau}); [\boldsymbol{\Theta}_C^{(2)}]_{\tau-1}) - \widetilde{r}_{c_\tau,\tau}| \right]$$

$$- |f_C^{(1)}(\boldsymbol{\nu}_{c_\tau,\tau}; [\boldsymbol{\Theta}_C^{(1)}]_{\tau-1}) + f_C^{(2)}(\nabla f_C^{(1)}(\boldsymbol{\nu}_{c_\tau,\tau}); [\boldsymbol{\Theta}_C^{(2)}]_{\tau-1}) - \widetilde{r}_{c_\tau,\tau}|.$$

Since the records in set $\mathcal{P}_\tau$ are sharing the same reward mapping function, we have the expectation

$$\mathbb{E}[V_\tau | F_\tau] = \mathbb{E}\left[ |f_C^{(1)}(\boldsymbol{\nu}_{c_\tau,\tau}; [\boldsymbol{\Theta}_C^{(1)}]_{\tau-1}) + f_C^{(2)}(\nabla f_C^{(1)}(\boldsymbol{\nu}_{c_\tau,\tau}); [\boldsymbol{\Theta}_C^{(2)}]_{\tau-1}) - \widetilde{r}_{c_\tau,\tau}| \right]$$

$$- \mathbb{E}\left[ |f_C^{(1)}(\boldsymbol{\nu}_{c_\tau,\tau}; [\boldsymbol{\Theta}_C^{(1)}]_{\tau-1}) + f_C^{(2)}(\nabla f_C^{(1)}(\boldsymbol{\nu}_{c_\tau,\tau}); [\boldsymbol{\Theta}_C^{(2)}]_{\tau-1}) - \widetilde{r}_{c_\tau,\tau}| \big| F_\tau \right] = 0.$$

where $F_\tau$ denotes the filtration given the past records. Then, we can have the mean value of $V_\tau$ across different time steps being

$$\frac{1}{t}\sum_{\tau\in[t]}[V_\tau] = \frac{1}{t}\sum_{\tau\in[t]}\mathbb{E}\big[\big|f_C^{(1)}(\boldsymbol{\nu}_{c_\tau,\tau};[\boldsymbol{\Theta}_C^{(1)}]_{\tau-1}) + f_C^{(2)}(\nabla f_C^{(1)}(\boldsymbol{\nu}_{c_\tau,\tau});[\boldsymbol{\Theta}_C^{(2)}]_{\tau-1}) - \widetilde{r}_{c_\tau,\tau}\big|\big]$$

$$-\frac{1}{t}\sum_{\tau\in[t]}\big|f_C^{(1)}(\boldsymbol{\nu}_{c_\tau,\tau};[\boldsymbol{\Theta}_C^{(1)}]_{\tau-1}) + f_C^{(2)}(\nabla f_C^{(1)}(\boldsymbol{\nu}_{c_\tau,\tau});[\boldsymbol{\Theta}_C^{(2)}]_{\tau-1}) - \widetilde{r}_{c_\tau,\tau}\big|.$$

with the expectation of zero. Then, we proceed to bound the expected estimation error of the exploitation model with the estimation error from existing samples. Afterwards, by applying the Azuma-Hoeffding inequality, with the probability at least $1-\delta$, the expected loss on $[\boldsymbol{\Theta}_C^{(1)}]_t$ could be bounded as

$$\frac{1}{t}\sum_{\tau\in[t]}\mathbb{E}\big[\big|f_C^{(1)}(\boldsymbol{\nu}_{c_\tau,\tau};[\boldsymbol{\Theta}_C^{(1)}]_{\tau-1}) + f_C^{(2)}(\nabla f_C^{(1)}(\boldsymbol{\nu}_{c_\tau,\tau});[\boldsymbol{\Theta}_C^{(2)}]_{\tau-1}) - \widetilde{r}_{c_\tau,\tau}\big|\big]$$

$$\leq \frac{1}{t}\sum_{\tau\in[t]}\big|f_C^{(1)}(\boldsymbol{\nu}_{c_\tau,\tau};[\boldsymbol{\Theta}_C^{(1)}]_{\tau-1}) + f_C^{(2)}(\nabla f_C^{(1)}(\boldsymbol{\nu}_{c_\tau,\tau});[\boldsymbol{\Theta}_C^{(2)}]_{\tau-1}) - \widetilde{r}_{c_\tau,\tau}\big| + \mathcal{O}(1)\cdot\sqrt{\frac{2}{t}\log(\frac{1}{\delta})}$$

where for the second term on the RHS, since we have $\mathcal{X}_\tau \sim \mathcal{D}, \tau \in [t]$ as well as $[\boldsymbol{\Theta}_C^{(1)}]_t, [\boldsymbol{\Theta}_C^{(2)}]_t \sim \{[\widehat{\boldsymbol{\Theta}}_C^{(1)}]_\tau, [\widehat{\boldsymbol{\Theta}}_C^{(2)}]_\tau\}_{\tau\in[t-1]}$, it will lead to

$$\mathbb{E}_{\boldsymbol{\nu}_{c_t,t}}\big|f_C^{(1)}(\boldsymbol{\nu}_{c_t,t};[\boldsymbol{\Theta}_C^{(1)}]_{t-1}) + f_C^{(2)}(\nabla f_C^{(1)}(\boldsymbol{\nu}_{c_t,t});[\boldsymbol{\Theta}_C^{(2)}]_{t-1}) - \widetilde{r}_{c_t,t}\big|$$

$$= \frac{1}{t}\sum_{\tau\in[t]}\big|f_C^{(1)}(\boldsymbol{\nu}_{c_\tau,\tau};[\boldsymbol{\Theta}_C^{(1)}]_{\tau-1}) + f_C^{(2)}(\nabla f_C^{(1)}(\boldsymbol{\nu}_{c_\tau,\tau});[\boldsymbol{\Theta}_C^{(2)}]_{\tau-1}) - \widetilde{r}_{c_\tau,\tau}\big|$$

$$\leq \frac{1}{\sqrt{t}}\sqrt{\sum_{\tau\in[t]}\big|f_C^{(1)}(\boldsymbol{\nu}_{c_\tau,\tau};[\boldsymbol{\Theta}_C^{(1)}]_{\tau-1}) + f_C^{(2)}(\nabla f_C^{(1)}(\boldsymbol{\nu}_{c_\tau,\tau});[\boldsymbol{\Theta}_C^{(2)}]_{\tau-1}) - \widetilde{r}_{c_\tau,\tau}\big|^2}$$

$$+ \mathcal{O}(1) + \frac{tLR^2}{\sqrt{m}} + \mathcal{O}(\frac{tR^{4/3}L^2\sqrt{\log m}}{m^{1/3}})$$

$$\leq \sqrt{\frac{\xi_R}{t}} + \mathcal{O}(1) + \frac{tLR^2}{\sqrt{m}} + \mathcal{O}(\frac{tR^{4/3}L^2\sqrt{\log m}}{m^{1/3}})$$

where with the notation from Lemma H.7 and Lemma H.6, we will also have the first term measuring the regression difficulty being $\xi_R = \inf_{\boldsymbol{\Theta}_C\in\mathcal{B}([\boldsymbol{\Theta}_C]_0,\ Rm^{-\frac{1}{4}})}\big[\frac{1}{2}\sum_{\tau\in[t]}\big|f_C^{(1)}(\boldsymbol{\nu}_{c_\tau,\tau};\boldsymbol{\Theta}_C^{(1)}) + f_C^{(2)}(\nabla f_C^{(1)}(\boldsymbol{\nu}_{c_\tau,\tau});\boldsymbol{\Theta}_C^{(2)}) - \widetilde{r}_{c_\tau,\tau}\big|^2\big]$, with $[\boldsymbol{\Theta}_C]_0$ be the initial parameters from either of the two category-level networks. The inequality is the application of Lemma H.7. Summing up all the components would complete the proof.

$\square$

**Corollary G.4.** *Given past collected records, we suppose $m, \eta_C, J$ satisfy the conditions in Theorem 5.1, and randomly draw the parameter $[\boldsymbol{\Theta}_C^{(1)}]_t, [\boldsymbol{\Theta}_C^{(2)}]_t \sim \{[\widehat{\boldsymbol{\Theta}}_C^{(1)}]_\tau, [\widehat{\boldsymbol{\Theta}}_C^{(2)}]_\tau\}_{\tau\in[t-1]}$. Consider the arms within candidate arm pools $\{\mathcal{X}_\tau\}_{\tau\in[t]}$ are drawn from a distribution $\mathcal{D}$ in the i.i.d. manner, and the past records $\mathcal{P}_t = \{\boldsymbol{\nu}_{c_\tau^*,\tau}, \widetilde{r}_{c_\tau^*,\tau}\}_{\tau\in[t-1]}$ up to round $t$ are generated by a policy. Then, with probability at least $1-\delta$, given a pair of arm category and its pseudo reward $\{\boldsymbol{\nu}_{c_t,t}, \widetilde{r}_{c_t,t}\}$, we have*

$$\mathbb{E}\left[\min\big\{|f_C^{(1)}(\boldsymbol{\nu}_{c_t^*,t};[\boldsymbol{\Theta}_C^{(1)}]_{t-1}) + f_C^{(2)}(\nabla f_C^{(1)}(\boldsymbol{\nu}_{c_t^*,t});[\boldsymbol{\Theta}_C^{(2)}]_{t-1}) - \widetilde{r}_{c_t^*,t}|,\ 1\big\}\ \Big|\mathcal{X}_t\right]$$

$$\leq \mathcal{O}\big(\sqrt{\frac{1}{t}\log(\frac{1}{\delta})}\big) + \sqrt{\frac{\xi_R}{t}} + \mathcal{O}(\frac{tLR^2}{\sqrt{m}} + \frac{tR^{4/3}L^2\sqrt{\log m}}{m^{1/3}}) + \mathcal{O}(R^{4/3}m^{-1/3}L^2\sqrt{\log m})$$

$$+ \mathcal{O}(\xi_C R^{1/3}m^{-1/12}L^3\sqrt{\log m})$$

*where $\widetilde{r}_{c_t^*,t}$ is the pseudo reward generated by the imaginary arm-level model $\mathcal{F}_A^*(\cdot)$, which is trained by optimal arms $\{\boldsymbol{x}_\tau^*, r_\tau^*\}_{\tau\in[t]}$.*

**Proof.** The proof of this corollary follows an analogous approach as in Lemma G.2 and Corollary G.3. Similarly, we first consider the alternative imaginary category-level models $f_C^{(1),*}(\cdot; [\boldsymbol{\Theta}_C^{(1),*}])$, $f_C^{(2),*}(\cdot; [\boldsymbol{\Theta}_C^{(2),*}])$, and arm-level model $f_A^*(\cdot; [\boldsymbol{\Theta}_A^*])$. These imaginary models are trained by the records optimal arms as well as optimal arm categories. Analogous to Lemma G.1, based on Lemma H.1, we have the LHS of the objective bounded by $\mathcal{O}(1)$. Then, we define the following martingale difference sequence with regard to the previous records up to round $\tau \in [t]$ in terms of the optimal arms and categories, such that

$$V_\tau^* = \mathbb{E}\big[\big|f_C^{(1),*}(\boldsymbol{\nu}_{c_\tau,\tau}; [\boldsymbol{\Theta}_C^{(1),*}]_{\tau-1}) + f_C^{(2),*}(\nabla f_C^{(1),*}(\boldsymbol{\nu}_{c_\tau,\tau}); [\boldsymbol{\Theta}_C^{(2),*}]_{\tau-1}) - \widetilde{r}_{c_\tau^*,\tau}^*\big|\big]$$
$$- \big|f_C^{(1),*}(\boldsymbol{\nu}_{c_\tau,\tau}; [\boldsymbol{\Theta}_C^{(1),*}]_{\tau-1}) + f_C^{(2),*}(\nabla f_C^{(1),*}(\boldsymbol{\nu}_{c_\tau,\tau}); [\boldsymbol{\Theta}_C^{(2),*}]_{\tau-1}) - \widetilde{r}_{c_\tau^*,\tau}^*\big|.$$

where we also have its expectation w.r.t. the filtration $F_\tau^*$, $\mathbb{E}[V_\tau^* | F_\tau^*] = 0$. Next, by applying the Azume-Hoeffing inequality, with $1 > \delta > 0$, we can also have

$$\mathbb{P}\Bigg[\frac{1}{t}\sum_{\tau \in [t]} \mathbb{E}\big[\big|f_C^{(1),*}(\boldsymbol{\nu}_{c_\tau,\tau}; [\boldsymbol{\Theta}_C^{(1),*}]_{\tau-1}) + f_C^{(2),*}(\nabla f_C^{(1),*}(\boldsymbol{\nu}_{c_\tau,\tau}); [\boldsymbol{\Theta}_C^{(2),*}]_{\tau-1}) - \widetilde{r}_{c_\tau^*,\tau}^*\big|\big]$$
$$- \frac{1}{t}\sum_{\tau \in [t]} \big|f_C^{(1),*}(\boldsymbol{\nu}_{c_\tau,\tau}; [\boldsymbol{\Theta}_C^{(1),*}]_{\tau-1}) + f_C^{(2),*}(\nabla f_C^{(1),*}(\boldsymbol{\nu}_{c_\tau,\tau}); [\boldsymbol{\Theta}_C^{(2),*}]_{\tau-1}) - \widetilde{r}_{c_\tau^*,\tau}^*\big| \geq \mathcal{O}(1) \cdot \sqrt{\frac{2}{t}\log(\frac{1}{\delta})}\Bigg] \leq \delta.$$

After applying Lemma H.6 and Lemma H.7, we will end up with

$$\mathbb{E}_{\boldsymbol{\nu}_{c_t^*,t}}\big|f_C^{(1),*}(\boldsymbol{\nu}_{c_t^*,t}; [\boldsymbol{\Theta}_C^{(1),*}]_{t-1}) + f_C^{(2),*}(\nabla f_C^{(1),*}(\boldsymbol{\nu}_{c_t^*,t}); [\boldsymbol{\Theta}_C^{(2),*}]_{t-1}) - \widetilde{r}_{c_t^*,t}^*\big|$$
$$= \frac{1}{t}\sum_{\tau \in [t]} \big|f_C^{(1),*}(\boldsymbol{\nu}_{c_\tau^*,\tau}; [\boldsymbol{\Theta}_C^{(1),*}]_{\tau-1}) + f_C^{(2),*}(\nabla f_C^{(1),*}(\boldsymbol{\nu}_{c_\tau^*,\tau}); [\boldsymbol{\Theta}_C^{(2),*}]_{\tau-1}) - \widetilde{r}_{c_\tau^*,\tau}^*\big|$$
$$\leq \frac{1}{\sqrt{t}}\sqrt{\sum_{\tau \in [t]} \big|f_C^{(1),*}(\boldsymbol{\nu}_{c_\tau^*,\tau}; [\boldsymbol{\Theta}_C^{(1),*}]_{\tau-1}) + f_C^{(2),*}(\nabla f_C^{(1),*}(\boldsymbol{\nu}_{c_\tau^*,\tau}); [\boldsymbol{\Theta}_C^{(2),*}]_{\tau-1}) - \widetilde{r}_{c_\tau^*,\tau}^*\big|^2}$$
$$+ \mathcal{O}(1) + \frac{tLR^2}{\sqrt{m}} + \mathcal{O}(\frac{tR^{4/3}L^2\sqrt{\log m}}{m^{1/3}})$$
$$\leq \sqrt{\frac{\xi_R}{t}} + \mathcal{O}(1) + \frac{tLR^2}{\sqrt{m}} + \mathcal{O}(\frac{tR^{4/3}L^2\sqrt{\log m}}{m^{1/3}}).$$

where the last inequality is because the parameter sphere $\mathcal{B}([\boldsymbol{\Theta}_C]_0, \; Rm^{-1/4})$ also includes the imaginary model parameters. Noticing that the current bound is built upon the imaginary category-models, we then need to further extend this bound to our processed category-level models.

With Lemma H.6, we know that the processed parameters $[\boldsymbol{\Theta}_C^{(1)}]_{\tau-1}, [\boldsymbol{\Theta}_C^{(2)}]_{\tau-1}$ as well as the imaginary ones both fall into $\mathcal{B}([\boldsymbol{\Theta}_C]_0, \; Rm^{-1/4})$. As a result,

$$\big|f_C^{(1),*}(\boldsymbol{\nu}_{c_\tau^*,\tau}; [\boldsymbol{\Theta}_C^{(1),*}]_{\tau-1}) + f_C^{(2),*}(\nabla f_C^{(1),*}(\boldsymbol{\nu}_{c_\tau^*,\tau}); [\boldsymbol{\Theta}_C^{(2),*}]_{\tau-1})$$
$$- f_C^{(1)}(\boldsymbol{\nu}_{c_\tau^*,\tau}; [\boldsymbol{\Theta}_C^{(1)}]_{\tau-1}) + f_C^{(2)}(\nabla f_C^{(1)}(\boldsymbol{\nu}_{c_\tau^*,\tau}); [\boldsymbol{\Theta}_C^{(2)}]_{\tau-1})\big|$$
$$\leq \big|f_C^{(1),*}(\boldsymbol{\nu}_{c_\tau^*,\tau}; [\boldsymbol{\Theta}_C^{(1),*}]_{\tau-1}) - f_C^{(1)}(\boldsymbol{\nu}_{c_\tau^*,\tau}; [\boldsymbol{\Theta}_C^{(1)}]_{\tau-1})\big|$$
$$+ \big|f_C^{(2),*}(\nabla f_C^{(1),*}(\boldsymbol{\nu}_{c_\tau^*,\tau}); [\boldsymbol{\Theta}_C^{(2),*}]_{\tau-1}) - f_C^{(2)}(\nabla f_C^{(1)}(\boldsymbol{\nu}_{c_\tau^*,\tau}); [\boldsymbol{\Theta}_C^{(2)}]_{\tau-1})\big|$$
$$\leq \mathcal{O}(Rm^{-1/4}\sqrt{L}) + \mathcal{O}(R^{4/3}m^{-1/3}L^2\sqrt{\log m})$$
$$+ \big|f_C^{(2),*}(\nabla f_C^{(1),*}(\boldsymbol{\nu}_{c_\tau^*,\tau}); [\boldsymbol{\Theta}_C^{(2),*}]_{\tau-1}) - f_C^{(2)}(\nabla f_C^{(1)}(\boldsymbol{\nu}_{c_\tau^*,\tau}); [\boldsymbol{\Theta}_C^{(2)}]_{\tau-1})\big|$$

where the inequality is by applying Lemma H.3. Then, for the last term on the RHS, supposing the input gradients are normalized by a coefficient $\frac{1}{\mathcal{O}(mL)}$, we will have

$$
\begin{aligned}
\big| f_C^{(2),*}&(\nabla f_C^{(1),*}(\boldsymbol{\nu}_{c_\tau^*,\tau}); [\boldsymbol{\Theta}_C^{(2),*}]_{\tau-1}) - f_C^{(2)}(\nabla f_C^{(1)}(\boldsymbol{\nu}_{c_\tau^*,\tau}); [\boldsymbol{\Theta}_C^{(2)}]_{\tau-1}) \big| \\
&\leq \big| f_C^{(2),*}(\nabla f_C^{(1),*}(\boldsymbol{\nu}_{c_\tau^*,\tau}); [\boldsymbol{\Theta}_C^{(2),*}]_{\tau-1}) - f_C^{(2)}(\nabla f_C^{(1),*}(\boldsymbol{\nu}_{c_\tau^*,\tau}); [\boldsymbol{\Theta}_C^{(2)}]_{\tau-1}) \big| \\
&\quad + \big| f_C^{(2)}(\nabla f_C^{(1),*}(\boldsymbol{\nu}_{c_\tau^*,\tau}); [\boldsymbol{\Theta}_C^{(2)}]_{\tau-1}) - f_C^{(2)}(\nabla f_C^{(1)}(\boldsymbol{\nu}_{c_\tau^*,\tau}); [\boldsymbol{\Theta}_C^{(2)}]_{\tau-1}) \big| \\
&\leq \mathcal{O}(R m^{-1/4}\sqrt{L}) + \mathcal{O}(R^{4/3} m^{-1/3} L^2 \sqrt{\log m}) \\
&\quad + \big| f_C^{(2)}(\nabla f_C^{(1),*}(\boldsymbol{\nu}_{c_\tau^*,\tau}); [\boldsymbol{\Theta}_C^{(2)}]_{\tau-1}) - f_C^{(2)}(\nabla f_C^{(1)}(\boldsymbol{\nu}_{c_\tau^*,\tau}); [\boldsymbol{\Theta}_C^{(2)}]_{\tau-1}) \big| \\
&\leq \mathcal{O}(R m^{-1/4}\sqrt{L}) + \mathcal{O}(R^{4/3} m^{-1/3} L^2 \sqrt{\log m}) + \mathcal{O}(\xi_C R^{1/3} m^{-1/12} L^3 \sqrt{\log m})
\end{aligned}
$$

where the first inequality is by the triangular inequality, the second inequality is because of Lemma H.3, and the final inequality is because of the fact that over-parameterized FC networks are Lipshitz continuous w.r.t. some constant $\xi_C > 0$ (Allen-Zhu et al., 2019) as well as Lemma H.4.

$\square$

# H Lemmas For Category-level Neural Networks

Here, without the loss of generality, we use $f_C$ to denote both $f_C^{(1)}$ and $f_C^{(2)}$, since these two models will follow very similar patterns. To begin with, given the input feature vector $\boldsymbol{x} \in \mathbb{R}^d$, we denote the $L$-layer fully-connected (FC) neural network with network width $m$ by

$$
f_C(\boldsymbol{x}; \boldsymbol{\Theta}) = \boldsymbol{\Theta}_L \big( \prod_{l=1}^{L-1} \mathbf{D}_l \boldsymbol{\Theta}_l \big) \cdot \boldsymbol{x}, \tag{8}
$$

where with $\phi$ being the ReLU activation, we define the intermediate hidden representations $\boldsymbol{h}_l, l \in \{0, \ldots, L-1\}$ as

$$
\boldsymbol{h}_0 = \boldsymbol{x}, \quad \boldsymbol{h}_l = \phi(\boldsymbol{\Theta}_l \boldsymbol{h}_{l-1}), l \in [L-1].
$$

and we also have the binary diagonal matrix functioning as the ReLU activation being

$$
\mathbf{D}_l = \mathrm{diag}(\mathbb{I}\{(\boldsymbol{\Theta}_l \boldsymbol{h}_{l-1})_1\}, \ldots, \mathbb{I}\{(\boldsymbol{\Theta}_l \boldsymbol{h}_{l-1})_m\}), l \in [L-1].
$$

where $\mathbb{I}(\cdot)$ is the indicator function. Afterwards, the corresponding gradients will naturally become

$$
\nabla_{\boldsymbol{\Theta}_l} f_C(\boldsymbol{x}; \boldsymbol{\Theta}) = \begin{cases} [\boldsymbol{h}_{l-1} \boldsymbol{\Theta}_L (\prod_{\tau=l+1}^{L-1} \mathbf{D}_\tau \boldsymbol{\Theta}_\tau)]^\intercal, l \in [L-1] \\ \boldsymbol{h}_{L-1}^\intercal, l = L. \end{cases} \tag{9}
$$

Recall that for the category-level models, for $l \in [L-1]$, each entry of $\boldsymbol{\Theta}_l$ is drawn from the Gaussian distribution $\mathcal{N}(0, 2/m)$; Each entry of $\boldsymbol{\Theta}_L$ is drawn from the Gaussian distribution $\mathcal{N}(0, 1/m)$.

**Lemma H.1.** *Suppose $m, \eta_C, \eta_2$ satisfy the conditions in Theorem 5.1. Based on the Gaussian initialization of weight matrices, with probability at least $1 - \mathcal{O}(TkL) \cdot \exp(-\Omega(m\omega^{2/3}L))$ over the random initialization, for all $t \in [T], i \in [k]$, $\boldsymbol{\Theta}$ satisfying $\|\boldsymbol{\Theta} - \boldsymbol{\Theta}_0\|_2 \leq \omega$ with $\omega \leq \mathcal{O}(L^{-9/2}[\log m]^{-3})$, it holds uniformly that*

$$
|f_C(\boldsymbol{\nu}_{c,t}; \boldsymbol{\Theta})| \leq \mathcal{O}(1).
$$
$$
\|\nabla_{\boldsymbol{\Theta}} f_C(\boldsymbol{\nu}_{c,t}; \boldsymbol{\Theta})\|_2 \leq \mathcal{O}(\sqrt{L}).
$$
$$
\|\nabla_{\boldsymbol{\Theta}} \mathcal{L}_t(\boldsymbol{\Theta}_t)\|_2 \leq \mathcal{O}(\sqrt{L}).
$$

*where $\mathcal{L}_t(\boldsymbol{\Theta}) = (f_C(\boldsymbol{x}_t; \boldsymbol{\Theta}) - r_t)^2/2$ is the $L_2$ prediction loss in terms of the arm $\boldsymbol{x}_t$.*

*Proof.* (1) is a simply application of Cauchy–Schwarz inequality.

$$|f_C(\boldsymbol{x}_t; \boldsymbol{\Theta})| = |\boldsymbol{\Theta}_L(\prod_{l=1}^{L-1}\mathbf{D}_l\boldsymbol{\Theta}_l)\boldsymbol{\nu}_{c,t}| \leq \underbrace{\|\boldsymbol{\Theta}_L(\prod_{l=1}^{L-1}\mathbf{D}_l\boldsymbol{\Theta}_l)\|_2}_{I_1}\|\boldsymbol{\nu}_{c,t}\|_2 \leq \mathcal{O}(1)$$

where the inequality is based on the Lemma B.2 (Cao & Gu, 2019) and the fact that $I_1 \leq \mathcal{O}(1)$, and $\|\boldsymbol{\nu}_{c,t}\|_2 = 1$. For (2), it holds uniformly that

$$\|\nabla_{\boldsymbol{\Theta}}f_C(\boldsymbol{\nu}_{c,t}; \boldsymbol{\Theta})\|_2 = \|\mathrm{vec}(\nabla_{\boldsymbol{\Theta}_1}f)^\intercal, \ldots, \mathrm{vec}(\nabla_{\boldsymbol{\Theta}_L}f)^\intercal\|_2 \leq \mathcal{O}(\sqrt{L})$$

where $\|\nabla_{\boldsymbol{\Theta}_1}f\|_F \leq \mathcal{O}(1)$ is an application of Lemma B.3 (Cao & Gu, 2019) by removing $\sqrt{m}$.

For (3), we have $\|\nabla_{\boldsymbol{\Theta}}\mathcal{L}_t(\boldsymbol{\Theta}_t)\|_2 \leq |\mathcal{L}_t'| \cdot \|\nabla_{\boldsymbol{\Theta}}f_C(\boldsymbol{\nu}_{c,t}; \boldsymbol{\Theta})\|_2 \leq \mathcal{O}(\sqrt{L})$. □

**Lemma H.2.** *Suppose $m, \eta_C, \eta_2$ satisfy the conditions in Theorem 5.1. With probability at least $1 - \mathcal{O}(TkL) \cdot \exp(-\Omega(m\omega^{2/3}L))$, for all $t \in [T], i \in [k]$, $\boldsymbol{\Theta}, \boldsymbol{\Theta}'$ satisfying $\|\boldsymbol{\Theta} - \boldsymbol{\Theta}_0\|_2, \|\boldsymbol{\Theta}' - \boldsymbol{\Theta}_0\|_2 \leq \omega$ with $\omega \leq \mathcal{O}(L^{-9/2}[\log m]^{-3})$, it holds uniformly that*

$$|f_C(\boldsymbol{x}; \boldsymbol{\Theta}) - f_C(\boldsymbol{x}; \boldsymbol{\Theta}') - \langle\nabla_{\boldsymbol{\Theta}'}f_C(\boldsymbol{x}; \boldsymbol{\Theta}'), \boldsymbol{\Theta} - \boldsymbol{\Theta}'\rangle| \leq \mathcal{O}(w^{1/3}L^2\sqrt{\log m})\|\boldsymbol{\Theta} - \boldsymbol{\Theta}'\|_2.$$

*Proof.* Based on Lemma 4.1 (Cao & Gu, 2019), it holds uniformly that

$$|\sqrt{m}f_C(\boldsymbol{x}; \boldsymbol{\Theta}) - \sqrt{m}f_C(\boldsymbol{x}; \boldsymbol{\Theta}') - \langle\sqrt{m}\nabla_{\boldsymbol{\Theta}'}f_C(\boldsymbol{x}; \boldsymbol{\Theta}'), \boldsymbol{\Theta} - \boldsymbol{\Theta}'\rangle| \leq \mathcal{O}(w^{1/3}L^2\sqrt{m\log(m)})\|\boldsymbol{\Theta} - \boldsymbol{\Theta}'\|_2,$$

where $\sqrt{m}$ comes from the different scaling of neural network structure. Removing $\sqrt{m}$ completes the proof. □

**Lemma H.3.** *Suppose $m, \eta_C, \eta_2$ satisfy the conditions in Theorem 5.1. With probability at least $1 - \mathcal{O}(TkL) \cdot \exp(-\Omega(m\omega^{2/3}L))$, for all $t \in [T], i \in [k]$, $\boldsymbol{\Theta}, \boldsymbol{\Theta}'$ satisfying $\|\boldsymbol{\Theta} - \boldsymbol{\Theta}_0\|_2, \|\boldsymbol{\Theta}' - \boldsymbol{\Theta}_0\|_2 \leq \omega$ with $\omega \leq \mathcal{O}(L^{-9/2}[\log m]^{-3})$, it holds uniformly that*

$$|f_C(\boldsymbol{x}; \boldsymbol{\Theta}) - f_C(\boldsymbol{x}; \boldsymbol{\Theta}')| \leq \mathcal{O}(\omega\sqrt{L}) + \mathcal{O}(\omega^{4/3}L^2\sqrt{\log m}) \tag{10}$$

*Proof.* Based on the LHS of the inequality in Lemma H.3, we have

$$|f_C(\boldsymbol{x}; \boldsymbol{\Theta}) - f_C(\boldsymbol{x}; \boldsymbol{\Theta}')|$$
$$\leq |\langle\nabla_{\boldsymbol{\Theta}'}f_C(\boldsymbol{x}; \boldsymbol{\Theta}'), \boldsymbol{\Theta} - \boldsymbol{\Theta}'\rangle| + \mathcal{O}(\omega^{1/3}L^2\sqrt{\log m})\|\boldsymbol{\Theta} - \boldsymbol{\Theta}'\|_2$$
$$\leq \|\nabla_{\boldsymbol{\Theta}'}f_C(\boldsymbol{x}; \boldsymbol{\Theta}')\|_2 \cdot \|\boldsymbol{\Theta} - \boldsymbol{\Theta}'\|_2 + \mathcal{O}(\omega^{1/3}L^2\sqrt{\log m})\|\boldsymbol{\Theta} - \boldsymbol{\Theta}'\|_2$$
$$\leq \mathcal{O}(\sqrt{L})\|\boldsymbol{\Theta} - \boldsymbol{\Theta}'\|_2 + \mathcal{O}(\omega^{1/3}L^2\sqrt{\log m})\|\boldsymbol{\Theta} - \boldsymbol{\Theta}'\|_2$$

The proof is completed. □

**Lemma H.4** (Theorem 5 in (Allen-Zhu et al., 2019))**.** *Suppose $m, \eta_C, \eta_2$ satisfy the conditions in Theorem 5.1. With probability at least $1 - \mathcal{O}(TkL) \cdot \exp(-\Omega(m\omega^{2/3}L))$, for all $t \in [T], i \in [k]$, $\boldsymbol{\Theta}, \boldsymbol{\Theta}'$ satisfying $\|\boldsymbol{\Theta} - \boldsymbol{\Theta}_0\|_2, \|\boldsymbol{\Theta}' - \boldsymbol{\Theta}_0\|_2 \leq \omega$ with $\omega \leq \mathcal{O}(L^{-9/2}[\log m]^{-3})$, it holds uniformly that*

$$|\nabla_{\boldsymbol{\Theta}}f_C(\boldsymbol{x}; \boldsymbol{\Theta}) - \nabla_{\boldsymbol{\Theta}}f_C(\boldsymbol{x}; \boldsymbol{\Theta}')| \leq \mathcal{O}(\omega^{1/3}L^3\sqrt{\log m})\|\nabla_{\boldsymbol{\Theta}}f_C(\boldsymbol{x}; \boldsymbol{\Theta}_0)\|. \tag{11}$$

**Lemma H.5.** *Let $\mathcal{L}_t(\boldsymbol{\Theta}) = (f_C(\boldsymbol{x}_t; \boldsymbol{\Theta}) - r_t)^2/2$ being the $L_2$ prediction loss in terms of the arm $\boldsymbol{x}_t$. Suppose $m, \eta_C, \eta_2$ satisfy the conditions in Theorem 5.1. With probability at least $1 - \mathcal{O}(TkL^2)\exp[-\Omega(m\omega^{2/3}L)]$ over randomness of $\boldsymbol{\Theta}_1$, for all $t \in [T]$, and $\boldsymbol{\Theta}, \boldsymbol{\Theta}'$ satisfying $\|\boldsymbol{\Theta} - \boldsymbol{\Theta}_0\|_2 \leq \omega$ and $\|\boldsymbol{\Theta}' - \boldsymbol{\Theta}_0\|_2 \leq \omega$ with $\omega \leq \mathcal{O}(L^{-6}[\log m]^{-3/2})$, it holds uniformly that*

$$\mathcal{L}_t(\boldsymbol{\Theta}') \geq \mathcal{L}_t(\boldsymbol{\Theta}) + \langle\nabla_{\boldsymbol{\Theta}}\mathcal{L}_t(\boldsymbol{\Theta}), \boldsymbol{\Theta}' - \boldsymbol{\Theta}\rangle - \epsilon$$

*where $\epsilon = \mathcal{O}(\omega^{4/3}L^3\sqrt{\log m})$.*

*Proof.* Let $\mathcal{L}'_t$ be the derivative of $\mathcal{L}_t$ with respective to $f_C(\boldsymbol{x}_t; \boldsymbol{\Theta})$. Then, it holds that $|\mathcal{L}'_t| \leq \mathcal{O}(1)$ based on Lemma H.1. Then, by convexity of $\mathcal{L}_t$, we have

$$
\begin{aligned}
&\mathcal{L}_t(\boldsymbol{\Theta}') - \mathcal{L}_t(\boldsymbol{\Theta}) \\
&\overset{(a)}{\geq} \mathcal{L}'_t[f_C(\boldsymbol{x}_t; \boldsymbol{\Theta}') - f_C(\boldsymbol{x}_t; \boldsymbol{\Theta})] \\
&\overset{(b)}{\geq} \mathcal{L}'_t\langle \nabla f_C(\boldsymbol{x}_t; \boldsymbol{\Theta}), \boldsymbol{\Theta}' - \boldsymbol{\Theta} \rangle - |\mathcal{L}'_t| \cdot |f_C(\boldsymbol{x}_t; \boldsymbol{\Theta}') - f_C(\boldsymbol{x}_t; \boldsymbol{\Theta}) - \langle \nabla f_C(\boldsymbol{x}_t; \boldsymbol{\Theta}), \boldsymbol{\Theta}' - \boldsymbol{\Theta} \rangle| \\
&\geq \langle \nabla_{\boldsymbol{\Theta}} \mathcal{L}_t(\boldsymbol{\Theta}), \boldsymbol{\Theta}' - \boldsymbol{\Theta} \rangle - |\mathcal{L}'_t| \cdot |f_C(\boldsymbol{x}_t; \boldsymbol{\Theta}') - f_C(\boldsymbol{x}_t; \boldsymbol{\Theta}) - \langle \nabla f_C(\boldsymbol{x}_t; \boldsymbol{\Theta}), \boldsymbol{\Theta}' - \boldsymbol{\Theta} \rangle| \\
&\overset{(c)}{\geq} \langle \nabla_{\boldsymbol{\Theta}'} \mathcal{L}_t, \boldsymbol{\Theta}' - \boldsymbol{\Theta} \rangle - \mathcal{O}(\omega^{4/3} L^3 \sqrt{\log m}) \\
&\geq \langle \nabla_{\boldsymbol{\Theta}'} \mathcal{L}_t, \boldsymbol{\Theta}' - \boldsymbol{\Theta} \rangle - \epsilon
\end{aligned}
$$

where $(a)$ is due to the convexity of $\mathcal{L}_t$, $(b)$ is an application of triangle inequality, and $(c)$ is the application of Lemma H.2. The proof is completed. □

**Lemma H.6.** *Suppose $m, \eta_C, \eta_2$ satisfy the conditions in Theorem 5.1. With probability at least $1 - \mathcal{O}(TkL^2) \exp[-\Omega(m\omega^{2/3}L)]$ over randomness of $\boldsymbol{\Theta}_0$, for any $R > 0$, it holds uniformly that*

$$
\|\boldsymbol{\Theta}_t - \boldsymbol{\Theta}_0\|_2 \leq \mathcal{O}(R/m^{1/4}), t \in [T].
$$

*Proof.* Let $\omega \leq \mathcal{O}(R/m^{1/4})$. The proof follows a simple induction. Obviously, $\boldsymbol{\Theta}_0$ is in the sphere $B(\boldsymbol{\Theta}_0, \omega)$, where the center is $\boldsymbol{\Theta}_0$ and radius is $\omega$. Suppose that $\boldsymbol{\Theta}_1, \boldsymbol{\Theta}_2, \ldots, \boldsymbol{\Theta}_T \in \mathcal{B}(\boldsymbol{\Theta}_0^2, \omega)$. We have, for any $t \in [T]$,

$$
\begin{aligned}
\|\boldsymbol{\Theta}_T - \boldsymbol{\Theta}_0\|_2 &\leq \sum_{t=1}^{T} \|\boldsymbol{\Theta}_t - \boldsymbol{\Theta}_{t-1}\|_2 \leq \sum_{t=1}^{T} \eta \|\nabla \mathcal{L}_t(\Theta_t)\| \leq \sum_{t=1}^{T} \eta \sqrt{L} \\
&= \mathcal{O}(TR^2 \sqrt{L}/\sqrt{m}) \leq \mathcal{O}(R/m^{1/4})
\end{aligned}
$$

The proof is complete. □

**Lemma H.7** (Theorem 2 in (Ban et al., 2022)). *Let $\mathcal{L}_t(\boldsymbol{\Theta}) = (f_C(\boldsymbol{x}_t; \boldsymbol{\Theta}) - r_t)^2/2$. Suppose $m, \eta_C, \eta_2$ satisfy the conditions in Theorem 5.1. With probability at least $1 - \mathcal{O}(TkL^2) \exp[-\Omega(m\omega^{2/3}L)]$ over randomness of $\boldsymbol{\Theta}_1$, given any $R > 0$ it holds that*

$$
\sum_{t}^{T} \mathcal{L}_t(\boldsymbol{\Theta}_t) \leq \sum_{t}^{T} \mathcal{L}_t(\boldsymbol{\Theta}^*) + \mathcal{O}(1) + \frac{TLR^2}{\sqrt{m}} + \mathcal{O}(\frac{TR^{4/3}L^2\sqrt{\log m}}{m^{1/3}}). \tag{12}
$$

*where $\boldsymbol{\Theta}^* = \arg\inf_{\boldsymbol{\Theta} \in B(\boldsymbol{\Theta}_0, R)} \sum_{t}^{T} \mathcal{L}_t(\boldsymbol{\Theta})$.*

*Proof.* Let $\boldsymbol{\Theta}' \in B(\boldsymbol{\Theta}_1, R)$. In round $t$, based on Lemma H.6, for any $t \in [T]$, $\|\boldsymbol{\Theta}_t - \boldsymbol{\Theta}'\|_2 \leq \omega \leq O(R/m^{1/4})$. Then, based on H.5, it holds uniformly

$$
\mathcal{L}_t(\boldsymbol{\Theta}_t) - \mathcal{L}_t(\boldsymbol{\Theta}') \leq \langle \nabla \mathcal{L}_t(\boldsymbol{\Theta}_t), \boldsymbol{\Theta}_t - \boldsymbol{\Theta}' \rangle + \epsilon,
$$

where $\epsilon = O(\omega^{4/3} L^2 \sqrt{\log m})$.

Therefore, for all $t \in [T], \boldsymbol{\Theta}' \in B(\boldsymbol{\Theta}_1, R)$, it holds uniformly

$$
\begin{aligned}
\mathcal{L}_t(\boldsymbol{\Theta}_t) - \mathcal{L}_t(\boldsymbol{\Theta}') &\overset{(a)}{\leq} \frac{\langle \boldsymbol{\Theta}_t - \boldsymbol{\Theta}_{t+1}, \boldsymbol{\Theta}_t - \boldsymbol{\Theta}' \rangle}{\eta} + \epsilon \\
&\overset{(b)}{=} \frac{\|\boldsymbol{\Theta}_t - \boldsymbol{\Theta}'\|_2^2 + \|\boldsymbol{\Theta}_t - \boldsymbol{\Theta}_{t+1}\|_2^2 - \|\boldsymbol{\Theta}_{t+1} - \boldsymbol{\Theta}'\|_2^2}{2\eta} + \epsilon \\
&\overset{(c)}{\leq} \frac{\|\boldsymbol{\Theta}_t - \boldsymbol{\Theta}'\|_2^2 - \|\boldsymbol{\Theta}_{t+1} - \boldsymbol{\Theta}'\|_2^2}{2\eta} + O(L\eta) + \epsilon
\end{aligned}
$$

where $(a)$ is because of the definition of gradient descent, $(b)$ is due to the fact $2\langle A, B\rangle = \|A\|_F^2 + \|B\|_F^2 - \|A - B\|_F^2$, $(c)$ is by $\|\mathbf{\Theta}_t - \mathbf{\Theta}_{t+1}\|_2^2 = \|\eta\nabla_{\mathbf{\Theta}}\mathcal{L}_t(\mathbf{\Theta}_t)\|_2^2 \leq \mathcal{O}(\eta^2 L)$.

Then, for $T$ rounds, we have

$$\sum_{t=1}^{T}\mathcal{L}_t(\mathbf{\Theta}_t) - \sum_{t=1}^{T}\mathcal{L}_t(\mathbf{\Theta}') \overset{(1)}{\leq} \frac{\|\mathbf{\Theta}_1 - \mathbf{\Theta}'\|_2^2}{2\eta} + \sum_{t=2}^{T}\|\mathbf{\Theta}_t - \mathbf{\Theta}'\|_2^2(\frac{1}{2\eta} - \frac{1}{2\eta}) + \sum_{t=1}^{T}L\eta + T\epsilon$$

$$\leq \frac{\|\mathbf{\Theta}_1 - \mathbf{\Theta}'\|_2^2}{2\eta} + \sum_{t=1}^{T}L\eta + T\epsilon$$

$$\leq \mathcal{O}(\frac{R^2}{\sqrt{m}\eta}) + \sum_{t=1}^{T}L\eta + T\epsilon$$

$$\overset{(2)}{\leq} \mathcal{O}(1) + \frac{TLR^2}{\sqrt{m}} + \mathcal{O}(\frac{TR^{4/3}L^2\sqrt{\log m}}{m^{1/3}})$$

where (1) is by simply discarding the last term and (2) is by $\eta = \frac{R^2}{\sqrt{m}}$ and replacing $\epsilon$ with $\omega = \mathcal{O}(R/m^{1/4})$.

$\square$

## I Bounding Errors for the Arm-level Model

In this section, we would like to prove the bound for arm-level recommendations. Recall that after filtering the arm categories $\widetilde{\mathcal{C}}_t$ with the first-level, namely the category-level, recommendation, the arm-level model intends to recommend one arm $\boldsymbol{x}_t \in \bigcup_{c\in\widetilde{\mathcal{C}}_t}\mathcal{X}_{c,t}$ accordingly, where $\mathcal{X}_{c,t}$ refers to the collection of arms for each category $c \in \widetilde{\mathcal{C}}_t$. First, we would like to restate the definition of the NTK Gram matrix as well as the effective dimension again for readers' reference.

**Definition 4** (NTK Matrix (Jacot et al., 2018; Zhou et al., 2020)). *Let $\mathcal{N}$ denote the Gaussian distribution. With, define*

$$\mathbf{M}_{i,j}^0 = \mathbf{\Psi}_{i,j}^0 = \langle \boldsymbol{x}_i, \boldsymbol{x}_j\rangle, \quad \mathbf{N}_{i,j}^l = \begin{pmatrix} \mathbf{\Psi}_{i,i}^l & \mathbf{\Psi}_{i,j}^l \\ \mathbf{\Psi}_{j,i}^l & \mathbf{\Psi}_{j,j}^l \end{pmatrix}$$

$$\mathbf{\Psi}_{i,j}^l = 2\mathbb{E}_{a,b\sim\mathcal{N}(\mathbf{0},\mathbf{N}_{i,j}^{l-1})}[\phi(a)\phi(b)]$$

$$\mathbf{M}_{i,j}^l = 2\mathbf{M}_{i,j}^{l-1}\mathbb{E}_{a,b\sim\mathcal{N}(\mathbf{0},\mathbf{N}_{i,j}^{l-1})}[\phi'(a)\phi'(b)] + \mathbf{\Psi}_{i,j}^l.$$

*Then, given the chosen arms $\{\boldsymbol{x}_t\}_{t=1}^{T}$ and the optimal arms $\{\boldsymbol{x}_t^*\}_{t=1}^{T}$, the Neural Tangent Kernel Gram Matrix is defined accordingly as $\mathbf{M} = (\mathbf{M}^L + \mathbf{\Psi}^L)/2 \in \mathbb{R}^{3T\times 3T}$.*

Meanwhile, similar to existing neural bandit works (Zhou et al., 2020; Zhang et al., 2021), we have the effective dimension definition regarding the NTK Gram matrix

**Definition 5** (Effective Dimension (Zhou et al., 2020)). *Given the chosen arms $\{\boldsymbol{x}_t\}_{t=1}^{T}$ and the optimal arms $\{\boldsymbol{x}_t^*\}_{t=1}^{T}$ over the finite horizon $T$, the effective dimension $\widetilde{d}$ is defined as*

$$\widetilde{d} = \frac{\log\det(\mathbf{I} + \mathbf{M}/\lambda)}{\log(1 + TK/\lambda)}.$$

*which measures the vanishing speed of eigenvalues of Gram matrix $\mathbf{M}$.*

The effective dimension here essentially measures the vanishing speed of the Gram matrix eigenvalues.

Apart from the NTK defined above, we also define the gradient mapping w.r.t. the neural network parameters as following, where the gradient-based mapping will be close to the NTK-based mapping under the over-parameterized settings. By applying the Lemma J.1 on the arm-level model, when it satisfies the conditions in Theorem 5.1, we have the expected reward of an arm $\boldsymbol{x} \in \mathcal{X}_t$ being

$$\mathbb{E}[r|x] = h_A(x) = \langle g(x; [\mathbf{\Theta}_A]_0), \bar{\mathbf{\Theta}} - [\mathbf{\Theta}_A]_0\rangle$$

where there exist parameters $\bar{\boldsymbol{\Theta}}$ such that $\|\bar{\boldsymbol{\Theta}} - [\boldsymbol{\Theta}_A]_0\| \leq S/\sqrt{m}, S > 0$. Meanwhile, we also denote the following NTK regression parameters

$$\boldsymbol{\Sigma}_{t-1}^{(0)} = \lambda\mathbf{I} + \sum_{\tau\in[t-1]} g(\boldsymbol{x}_\tau; [\boldsymbol{\Theta}_A]_0) \cdot g(\boldsymbol{x}_\tau; [\boldsymbol{\Theta}_A]_0)^\mathsf{T}/m,$$

$$\boldsymbol{b}_{t-1}^{(0)} = \sum_{\tau\in[t-1]} g(\boldsymbol{x}_\tau; [\boldsymbol{\Theta}_A]_0) \cdot r_\tau/\sqrt{m},$$

$$\boldsymbol{\Sigma}_{t-1} = \lambda\mathbf{I} + \sum_{\tau\in[t-1]} g(\boldsymbol{x}_\tau; [\boldsymbol{\Theta}_A]_{\tau-1}) \cdot g(\boldsymbol{x}_\tau; [\boldsymbol{\Theta}_A]_{\tau-1})^\mathsf{T}/m,$$

$$\boldsymbol{b}_{t-1} = \sum_{\tau\in[t-1]} g(\boldsymbol{x}_\tau; [\boldsymbol{\Theta}_A]_{\tau-1}) \cdot r_\tau/\sqrt{m},$$

where $\{\boldsymbol{x}_\tau, r_\tau\}, \tau \in [t]$ respectively stands for the chosen arms as well as their rewards, and $g(\boldsymbol{x}; \boldsymbol{\Theta}_A) \in \mathbb{R}^{p_A}$ refer to the vectorized network gradients w.r.t. $\boldsymbol{\Theta}_A$ when taking $\boldsymbol{x}$ as the input. Then, with the above definitions, we proceed to bound the recommendation error of the arm-level model.

## I.1 Regret Bound for Arm-level Recommendation

Next, we would like to bound the arm-level recommendation error, which refers to $\mathcal{F}_A(\boldsymbol{x}_t) - h_C(\boldsymbol{\nu}_{c_t,t})$. This measures the difference between the arm-level model output and the arm category reward (i.e., the reward of the optimal arm in the filtered arm categories $\widetilde{x}_t^*$). Here, the estimation error of the arm-level recommendation, can be transformed to

$$\min\{\mathcal{F}_A(\boldsymbol{x}_t) - h_C(\boldsymbol{\nu}_{c_t,t}),\ 1\} = \min\{\mathcal{F}_A(\boldsymbol{x}_t) - h_A(\boldsymbol{x}_t) + h_A(\boldsymbol{x}_t) - h_C(\boldsymbol{\nu}_{c_t,t}),\ 1\}$$
$$\leq |\mathcal{F}_A(\boldsymbol{x}_t) - h_A(\boldsymbol{x}_t)| + |h_A(\boldsymbol{x}_t) - h_C(\boldsymbol{\nu}_{c_t,t})|$$
$$= \underbrace{|\mathcal{F}_A(\boldsymbol{x}_t) - h_A(\boldsymbol{x}_t)|}_{\text{Error for Reward Estimation}} + \underbrace{|h_A(\boldsymbol{x}_t) - h_A(\widetilde{\boldsymbol{x}}_t^*)|}_{\text{Reward Gap}}$$

where we recall that $\widetilde{\boldsymbol{x}}_t^* = \arg\max_{\boldsymbol{x}\in(\bigcup_{c\in\widetilde{\mathcal{C}}_t}\mathcal{X}_{c,t})}\left[h_A(\boldsymbol{x})\right]$ refers to the optimal arm within the filtered candidate arm pool. Here, the first term on the RHS refers to the estimation error of the arm-level model for the reward of chosen arm $\boldsymbol{x}_t$, while the second term refers to the reward gap between the chosen arm $\boldsymbol{x}_t$ and the optimal arm in the filtered candidate pool $\widetilde{\boldsymbol{x}}_t^*$.

### I.1.1 Term 1: Error for Reward Estimation

Here, we will have the following lemma for bounding the first term $|\mathcal{F}_A(\boldsymbol{x}_t) - h_A(\boldsymbol{x}_t)|$.

**Lemma I.1.** *Suppose the arm-level neural network $f_A(\cdot; [\boldsymbol{\Theta}_A]_{t-1})$ has been trained on chosen arms and their rewards $\{\boldsymbol{x}_\tau, r_\tau\}_{\tau\in[t-1]}$, with $J$-iteration GD and learning rate $\eta$. Meanwhile, $f_A(\cdot; [\boldsymbol{\Theta}_A]_{t-1})$ is a $L$-layer FC network with width $m$. Suppose we have $m, J, \eta$ satisfying the conditions in Theorem 5.1. Then, with the probability at least $1-\delta$, we will have*

$$|\mathcal{F}_A(\boldsymbol{x}_t) - h_A(\boldsymbol{x}_t)| \leq 2\gamma_{t-1}(\alpha) \cdot \|g(\boldsymbol{x}_t; [\boldsymbol{\Theta}_A]_{t-1})/\sqrt{m}\|_{\boldsymbol{\Sigma}_{t-1}^{-1}} + \mathcal{O}(m^{-1/6}\sqrt{\log(m)}t^{2/3}\lambda^{-2/3}L^3)$$
$$+ \mathcal{O}(Sm^{-1/6}\sqrt{\log(m)}t^{1/6}\lambda^{-1/6}L^{2/7})$$

*By definition in Lemma I.4 and Theorem 1, we have $\mathcal{F}_A(\boldsymbol{x}_t) = f_A(\boldsymbol{x}_t; [\boldsymbol{\Theta}_A]_{t-1}) + UCB_\alpha(\boldsymbol{x}_t)$, and the upper confidence bound $UCB_\alpha(\boldsymbol{x}_t) = \gamma_{t-1}(\alpha) \cdot \sqrt{g(\boldsymbol{x}_t; [\boldsymbol{\Theta}_A]_{t-1})^\mathsf{T}\boldsymbol{\Sigma}_{t-1}^{-1}g(\boldsymbol{x}_t; [\boldsymbol{\Theta}_A]_{t-1})}$, where*

$$\boldsymbol{\Sigma}_{t-1} = \lambda\mathbf{I} + \sum_{\tau\in[t]} g(\boldsymbol{x}_\tau; [\boldsymbol{\Theta}_A]_{\tau-1}) \cdot g(\boldsymbol{x}_\tau; [\boldsymbol{\Theta}_A]_{\tau-1})^\mathsf{T}$$

$$\gamma_{t-1}(\alpha) = \mathcal{O}\left(\alpha \cdot \sqrt{\log\frac{\det(\boldsymbol{\Sigma}_{t-1})}{\det(\Sigma_{\lambda\mathbf{I}})} - 2\log(\delta)} + \lambda^{1/2}S\right)$$

**Proof.** The proof of this lemma is following an analogous approach as in Lemma 5.2 of (Zhou et al., 2020). Recall that we have the combination of reward estimation and the UCB being $\mathcal{F}_A(\boldsymbol{x}) = f(\boldsymbol{x}; [\boldsymbol{\Theta}_A]_{t-1}) + \gamma_{t-1}(\alpha) \cdot \sqrt{g(\boldsymbol{x}; [\boldsymbol{\Theta}_A]_{t-1})^\intercal \boldsymbol{\Sigma}_{t-1}^{-1} g(\boldsymbol{x}; [\boldsymbol{\Theta}_A]_{t-1})}$. Meanwhile, consider an alternative being

$$\widetilde{\mathcal{F}}_A(\boldsymbol{x}) = \langle g(\boldsymbol{x}; [\boldsymbol{\Theta}_A]_{t-1}), [\boldsymbol{\Theta}_A]_{t-1} - [\boldsymbol{\Theta}_A]_0 \rangle + \gamma_{t-1}(\alpha) \cdot \sqrt{g(\boldsymbol{x}; [\boldsymbol{\Theta}_A]_{t-1})^\intercal \boldsymbol{\Sigma}_{t-1}^{-1} g(\boldsymbol{x}; [\boldsymbol{\Theta}_A]_{t-1})}$$
$$= \max_{\boldsymbol{\Theta}_A \in \mathcal{E}_{t-1}} \langle g(\boldsymbol{x}; [\boldsymbol{\Theta}_A]_{t-1}), \boldsymbol{\Theta}_A - [\boldsymbol{\Theta}_A]_0 \rangle$$

where we also have the output difference $|\mathcal{F}_A(\boldsymbol{x}) - \widetilde{\mathcal{F}}_A(\boldsymbol{x})| \le \mathcal{O}(m^{-1/6}\sqrt{\log(m)}t^{2/3}\lambda^{-2/3}L^3)$ based on Lemma J.8. Afterwards, with Lemma J.1 and Lemma I.4, we will also have $\boldsymbol{\Theta}_A \in \mathcal{E}_{t-1}$, and the objective can therefore be transformed into

$$\begin{aligned}
|\mathcal{F}_A(\boldsymbol{x}_t) - h_A(\boldsymbol{x}_t)| &\le |\widetilde{\mathcal{F}}_A(\boldsymbol{x}_t) - h_A(\boldsymbol{x}_t)| + \mathcal{O}(m^{-1/6}\sqrt{\log(m)}t^{2/3}\lambda^{-2/3}L^3) \\
&= \Big| \max_{\boldsymbol{\Theta}_A \in \mathcal{E}_{t-1}} \langle g(\boldsymbol{x}_t; [\boldsymbol{\Theta}_A]_{t-1}), \boldsymbol{\Theta}_A - [\boldsymbol{\Theta}_A]_0 \rangle - \langle g(\boldsymbol{x}_t; [\boldsymbol{\Theta}_A]_{t-1}), \bar{\boldsymbol{\Theta}}_A - [\boldsymbol{\Theta}_A]_0 \rangle \Big| \\
&\quad + \mathcal{O}(m^{-1/6}\sqrt{\log(m)}t^{2/3}\lambda^{-2/3}L^3) + \mathcal{O}(Sm^{-1/6}\sqrt{\log(m)}t^{1/6}\lambda^{-1/6}L^{2/7}) \\
&= \Big| \max_{\boldsymbol{\Theta}_A \in \mathcal{E}_{t-1}} \langle g(\boldsymbol{x}_t; [\boldsymbol{\Theta}_A]_{t-1}), \boldsymbol{\Theta}_A - [\boldsymbol{\Theta}_A]_{t-1} \rangle - \langle g(\boldsymbol{x}_t; [\boldsymbol{\Theta}_A]_{t-1}), \bar{\boldsymbol{\Theta}}_A - [\boldsymbol{\Theta}_A]_{t-1} \rangle \Big| \\
&\quad + \mathcal{O}(m^{-1/6}\sqrt{\log(m)}t^{2/3}\lambda^{-2/3}L^3) + \mathcal{O}(Sm^{-1/6}\sqrt{\log(m)}t^{1/6}\lambda^{-1/6}L^{2/7}) \\
&\le \|g(\boldsymbol{x}_t; [\boldsymbol{\Theta}_A]_{t-1})/\sqrt{m}\|_{\boldsymbol{\Sigma}_{t-1}^{-1}} \cdot \sqrt{m} \cdot \Big( \|\bar{\boldsymbol{\Theta}}_A - [\boldsymbol{\Theta}_A]_{t-1}\|_{\boldsymbol{\Sigma}_{t-1}} + \max_{\boldsymbol{\Theta}_A \in \mathcal{E}_{t-1}} \|\boldsymbol{\Theta}_A - [\boldsymbol{\Theta}_A]_{t-1}\|_{\boldsymbol{\Sigma}_{t-1}} \Big) \\
&\quad + \mathcal{O}(m^{-1/6}\sqrt{\log(m)}t^{2/3}\lambda^{-2/3}L^3) + \mathcal{O}(Sm^{-1/6}\sqrt{\log(m)}t^{1/6}\lambda^{-1/6}L^{2/7}) \\
&\le 2\gamma_{t-1}(\alpha) \cdot \|g(\boldsymbol{x}_t; [\boldsymbol{\Theta}_A]_{t-1})/\sqrt{m}\|_{\boldsymbol{\Sigma}_{t-1}^{-1}} + \mathcal{O}(m^{-1/6}\sqrt{\log(m)}t^{2/3}\lambda^{-2/3}L^3) \\
&\quad + \mathcal{O}(Sm^{-1/6}\sqrt{\log(m)}t^{1/6}\lambda^{-1/6}L^{2/7})
\end{aligned}$$

where the second inequality is by applying the Holder's inequality, and the last inequality is by applying Lemmas Lemma 5.2 in (Zhou et al., 2020), Lemma J.1, in terms of the confidence set $\mathcal{E}_{t-1}$.

$\square$

Apart from this lemma that bounds the UCB for the chosen arm $\boldsymbol{x}_t$, we can also have the following corollary for the UCB of the imaginary shadow model $\mathcal{F}_A^*(\cdot)$. Here, with the optimal arm being $\boldsymbol{x}_t^* = \arg\max_{\boldsymbol{x} \in \mathcal{X}_t} h_A(\boldsymbol{x})$, we can bound the UCB for reward estimation $|h_A(\boldsymbol{x}_t^*) - \mathcal{F}_A^*(\boldsymbol{x}_t^*)|$, where we have the shadow model $\mathcal{F}_A^*$ trained with past optimal arms and their rewards $\{\boldsymbol{x}_\tau, r_\tau\}_{\tau \in [t-1]}$. Comparable to the sequence of chosen arms $\{\boldsymbol{x}_\tau\}_{\tau \in [t-1]}$, The definition of $\gamma_{t-1}^*(\alpha)$

**Corollary I.2.** *Suppose the an imaginary arm-level neural network $f_A^*(\cdot; [\boldsymbol{\Theta}_A^*]_{t-1})$ has been trained on the optimal arms and their rewards $\{\boldsymbol{x}_\tau^*, r_\tau^*\}_{\tau \in [t-1]}$, with $J$-iteration GD and learning rate $\eta$. Similarly, $f_A^*(\cdot; [\boldsymbol{\Theta}_A^*]_{t-1})$ is a $L$-layer FC network with width $m$, and suppose we have $m, J, \eta$ satisfying the conditions in Theorem 5.1. Then, with the probability at least $1 - \delta$, we will have*

$$\begin{aligned}
|h_A(\boldsymbol{x}_t^*) - \mathcal{F}_A^*(\boldsymbol{x}_t^*)| &\le 2\gamma_{t-1}^*(\alpha) \cdot \|g(\boldsymbol{x}_t^*; [\boldsymbol{\Theta}_A^*]_{t-1})/\sqrt{m}\|_{(\boldsymbol{\Sigma}_{t-1}^*)^{-1}} + \mathcal{O}(m^{-1/6}\sqrt{\log(m)}t^{2/3}\lambda^{-2/3}L^3) \\
&\quad + \mathcal{O}(Sm^{-1/6}\sqrt{\log(m)}t^{1/6}\lambda^{-1/6}L^{2/7})
\end{aligned}$$

*where we have $\mathcal{F}_A^*(\boldsymbol{x}_t) = f_A^*(\boldsymbol{x}_t; [\boldsymbol{\Theta}_A^*]_{t-1}) + \mathsf{UCB}_\alpha^*(\boldsymbol{x}_t)$, and we also have $\mathsf{UCB}_\alpha^*(\boldsymbol{x}_t^*) = \gamma_{t-1}^*(\alpha) \cdot \sqrt{g(\boldsymbol{x}_t^*; [\boldsymbol{\Theta}_A^*]_{t-1})^\intercal (\boldsymbol{\Sigma}_{t-1}^*)^{-1} g(\boldsymbol{x}_t^*; [\boldsymbol{\Theta}_A^*]_{t-1})}$, as well as*

$$\boldsymbol{\Sigma}_{t-1}^* = \lambda\mathbf{I} + \sum_{\tau \in [t]} g(\boldsymbol{x}_\tau^*; [\boldsymbol{\Theta}_A^*]_{\tau-1}) \cdot g(\boldsymbol{x}_\tau^*; [\boldsymbol{\Theta}_A^*]_{\tau-1})^\intercal$$

$$\gamma_{t-1}^*(\alpha) = \mathcal{O}\left( \alpha \cdot \sqrt{\log\frac{\det(\boldsymbol{\Sigma}_{t-1}^*)}{\det(\lambda\mathbf{I})} - 2\log(\delta)} + \lambda^{1/2}S \right)$$

**Proof.** The proof of this corollary is similar to that of Lemma I.1. Analogously, consider an alternative of $\mathcal{F}_A^*(\cdot)$ being

$$\widetilde{\mathcal{F}}_A^*(\boldsymbol{x}) = \big\langle g(\boldsymbol{x}; [\boldsymbol{\Theta}_A^*]_{t-1}), [\boldsymbol{\Theta}_A^*]_{t-1} - [\boldsymbol{\Theta}_A^*]_0 \big\rangle + \gamma_{t-1}(\alpha) \cdot \sqrt{g(\boldsymbol{x}; [\boldsymbol{\Theta}_A^*]_{t-1})^\intercal (\boldsymbol{\Sigma}_{t-1}^*)^{-1} g(\boldsymbol{x}; [\boldsymbol{\Theta}_A^*]_{t-1})}$$

$$= \max_{\boldsymbol{\Theta}_A^* \in \mathcal{E}_{t-1}} \big\langle g(\boldsymbol{x}; [\boldsymbol{\Theta}_A^*]_{t-1}), \boldsymbol{\Theta}_A^* - [\boldsymbol{\Theta}_A^*]_0 \big\rangle$$

where we can also prove the output difference $|\mathcal{F}_A^*(\boldsymbol{x}) - \widetilde{\mathcal{F}}_A^*(\boldsymbol{x})| \leq \mathcal{O}(m^{-1/6}\sqrt{\log(m)}t^{2/3}\lambda^{-2/3}L^3)$ with a similar approach as in Lemma J.8. Afterwards, with Lemma J.1 and Lemma I.4, we will also have $\bar{\boldsymbol{\Theta}}_A \in \mathcal{E}_{t-1}^*$. We have $\mathcal{E}_{t-1}^* = \{\boldsymbol{\Theta}_A : \|\boldsymbol{\Theta}_A - [\boldsymbol{\Theta}_A^*]_{t-1}\|_{\boldsymbol{\Sigma}_{t-1}^*} \leq \gamma_{t-1}^*(\alpha)/\sqrt{m}\}$ being the confidence set centered around the parameters $[\boldsymbol{\Theta}_A^*]_{t-1}$ trained by optimal arms and their rewards. Then, the objective can therefore be transformed into

$$\big|\mathcal{F}_A^*(\boldsymbol{x}_t^*) - h_A(\boldsymbol{x}_t^*)\big| \leq \big|\widetilde{\mathcal{F}}_A^*(\boldsymbol{x}_t^*) - h_A(\boldsymbol{x}_t^*)\big| + \mathcal{O}(m^{-1/6}\sqrt{\log(m)}t^{2/3}\lambda^{-2/3}L^3)$$

$$\leq \|g(\boldsymbol{x}_t^*; [\boldsymbol{\Theta}_A]_{t-1})/\sqrt{m}\|_{\boldsymbol{\Sigma}_{t-1}^{-1}} \cdot \sqrt{m} \cdot \big(\|\bar{\boldsymbol{\Theta}}_A - [\boldsymbol{\Theta}_A^*]_{t-1}\|_{\boldsymbol{\Sigma}_{t-1}^*} + \max_{\boldsymbol{\Theta}_A \in \mathcal{E}_{t-1}^*} \|\boldsymbol{\Theta}_A - [\boldsymbol{\Theta}_A^*]_{t-1}\|_{\boldsymbol{\Sigma}_{t-1}^*}\big)$$

$$+ \mathcal{O}(m^{-1/6}\sqrt{\log(m)}t^{2/3}\lambda^{-2/3}L^3) + \mathcal{O}(Sm^{-1/6}\sqrt{\log(m)}t^{1/6}\lambda^{-1/6}L^{2/7})$$

$$\leq 2\gamma_{t-1}^*(\alpha) \cdot \|g(\boldsymbol{x}_t^*; [\boldsymbol{\Theta}_A^*]_{t-1})/\sqrt{m}\|_{(\boldsymbol{\Sigma}_{t-1}^*)^{-1}} + \mathcal{O}(m^{-1/6}\sqrt{\log(m)}t^{2/3}\lambda^{-2/3}L^3)$$

$$+ \mathcal{O}(Sm^{-1/6}\sqrt{\log(m)}t^{1/6}\lambda^{-1/6}L^{2/7}).$$

where the second inequality is by applying the Holder's inequality, and the last inequality is by applying Lemmas Lemma 5.2 in (Zhou et al., 2020), Lemma J.1, in terms of the confidence set of optimal arms $\mathcal{E}_{t-1}^*$. This is because for a given sequence of arms, which can be chosen arms $\{\boldsymbol{x}_t\}$ or the optimal arms $\{\boldsymbol{x}_t^*\}$, we will be able to approximate the true reward mapping function and derive the confidence set for it.

$\square$

### I.1.2 Term 2: Reward Gap

Next, we proceed to give the following lemma for bounding the second term, $\big|h_A(\boldsymbol{x}_t) - h_A(\widetilde{\boldsymbol{x}}_t^*)\big|$.

**Lemma I.3.** *Suppose the arm-level neural network $f_A(\cdot; [\boldsymbol{\Theta}_A]_{t-1})$ has been trained on chosen arms and their rewards $\{\boldsymbol{x}_\tau, r_\tau\}_{\tau \in [t-1]}$, with $J$-iteration GD and learning rate $\eta$. Meanwhile, $f_A(\cdot; [\boldsymbol{\Theta}_A]_{t-1})$ is a $L$-layer FC network with width $m$. Suppose we have $m, J, \eta$ satisfying the conditions in Theorem 5.1. Then, with the probability at least $1 - \delta$, we will have*

$$\big|h_A(\boldsymbol{x}_t) - h_A(\widetilde{\boldsymbol{x}}_t^*)\big| \leq 2\gamma_{t-1}(\alpha) \cdot \|g(\boldsymbol{x}_t; [\boldsymbol{\Theta}_A]_{t-1})/\sqrt{m}\|_{\boldsymbol{\Sigma}_{t-1}^{-1}} + \mathcal{O}(m^{-1/6}\sqrt{\log(m)}t^{2/3}\lambda^{-2/3}L^3)$$

$$+ \mathcal{O}(Sm^{-1/6}\sqrt{\log(m)}t^{1/6}\lambda^{-1/6}L^{2/7})$$

*By definitions in Lemma I.4, we denote $\mathcal{F}_A(\boldsymbol{x}_t) = f_A(\boldsymbol{x}_t; [\boldsymbol{\Theta}_A]_{t-1}) + UCB_\alpha(\boldsymbol{x}_t)$, and $UCB_\alpha(\boldsymbol{x}_t) = \gamma_{t-1}(\alpha) \cdot \sqrt{g(\boldsymbol{x}_t; [\boldsymbol{\Theta}_A]_{t-1})^\intercal \boldsymbol{\Sigma}_{t-1}^{-1} g(\boldsymbol{x}_t; [\boldsymbol{\Theta}_A]_{t-1})}$, where*

$$\gamma_{t-1}(\alpha) = \mathcal{O}\left(\alpha \cdot \sqrt{\log \frac{\det(\boldsymbol{\Sigma}_{t-1})}{\det(\lambda \mathbf{I})} - 2\log(\delta)} + \lambda^{1/2}S\right)$$

**Proof.** The proof of this lemma is extends the proof of Lemma I.1. Following the workflow in Lemma I.1, we continue to apply the alternative of the arm-selection mechanism as

$$\widetilde{\mathcal{F}}_A(\boldsymbol{x}) = \big\langle g(\boldsymbol{x}; [\boldsymbol{\Theta}_A]_{t-1}), [\boldsymbol{\Theta}_A]_{t-1} - [\boldsymbol{\Theta}_A]_0 \big\rangle + \gamma_{t-1}(\alpha) \cdot \sqrt{g(\boldsymbol{x}; [\boldsymbol{\Theta}_A]_{t-1})^\intercal \boldsymbol{\Sigma}_{t-1}^{-1} g(\boldsymbol{x}; [\boldsymbol{\Theta}_A]_{t-1})}$$

$$= \max_{\boldsymbol{\Theta}_A \in \mathcal{E}_{t-1}} \big\langle g(\boldsymbol{x}; [\boldsymbol{\Theta}_A]_{t-1}), \boldsymbol{\Theta}_A - [\boldsymbol{\Theta}_A]_0 \big\rangle$$

where we also have the output difference $|\mathcal{F}_A(\boldsymbol{x}) - \widetilde{\mathcal{F}}_A(\boldsymbol{x})| \leq \mathcal{O}(m^{-1/6}\sqrt{\log(m)}t^{2/3}\lambda^{-2/3}L^3)$ based on Lemma J.8. Next, with Lemma J.1, we can transform our objective to

$$
\begin{aligned}
\left|h_A(\boldsymbol{x}_t) - h_A(\widetilde{\boldsymbol{x}}_t^*)\right| &= h_A(\widetilde{\boldsymbol{x}}_t^*) - h_A(\boldsymbol{x}_t) \\
&\leq \left\langle g(\widetilde{\boldsymbol{x}}_t^*; [\boldsymbol{\Theta}_A]_0), \bar{\boldsymbol{\Theta}}_A - [\boldsymbol{\Theta}_A]_0 \right\rangle - \left\langle g(\boldsymbol{x}_t; [\boldsymbol{\Theta}_A]_0), \bar{\boldsymbol{\Theta}}_A - [\boldsymbol{\Theta}_A]_0 \right\rangle \\
&\leq \left\langle g(\widetilde{\boldsymbol{x}}_t^*; [\boldsymbol{\Theta}_A]_{t-1}), \bar{\boldsymbol{\Theta}}_A - [\boldsymbol{\Theta}_A]_0 \right\rangle - \left\langle g(\boldsymbol{x}_t; [\boldsymbol{\Theta}_A]_{t-1}), \bar{\boldsymbol{\Theta}}_A - [\boldsymbol{\Theta}_A]_0 \right\rangle + \mathcal{O}(Sm^{-1/6}\sqrt{\log(m)}t^{1/6}\lambda^{-1/6}L^{2/7}) \\
&\leq \max_{\boldsymbol{\Theta}_A \in \mathcal{E}_{t-1}} \left\langle g(\widetilde{\boldsymbol{x}}_t^*; [\boldsymbol{\Theta}_A]_{t-1}), \boldsymbol{\Theta}_A - [\boldsymbol{\Theta}_A]_0 \right\rangle - \left\langle g(\boldsymbol{x}_t; [\boldsymbol{\Theta}_A]_{t-1}), \bar{\boldsymbol{\Theta}}_A - [\boldsymbol{\Theta}_A]_0 \right\rangle + \mathcal{O}(Sm^{-1/6}\sqrt{\log(m)}t^{1/6}\lambda^{-1/6}L^{2/7}) \\
&= \widetilde{\mathcal{F}}_A(\widetilde{\boldsymbol{x}}_t^*) - \left\langle g(\boldsymbol{x}_t; [\boldsymbol{\Theta}_A]_{t-1}), \bar{\boldsymbol{\Theta}}_A - [\boldsymbol{\Theta}_A]_0 \right\rangle + \mathcal{O}(Sm^{-1/6}\sqrt{\log(m)}t^{1/6}\lambda^{-1/6}L^{2/7}),
\end{aligned}
$$

where the first inequality is due to Lemma J.1, second inequality is due to Lemma J.7, third inequality is due to the fact that $\bar{\boldsymbol{\Theta}}_A \in \mathcal{E}_{t-1}$, and the final equality is due to the definition of $\widetilde{\mathcal{F}}_A(\cdot)$.

Next, by utilizing the relationship between the pulling mechanisms, we will have

$$
\begin{aligned}
\widetilde{\mathcal{F}}_A(\widetilde{\boldsymbol{x}}_t^*) &- \left\langle g(\boldsymbol{x}_t; [\boldsymbol{\Theta}_A]_{t-1}), \bar{\boldsymbol{\Theta}}_A - [\boldsymbol{\Theta}_A]_0 \right\rangle + \mathcal{O}(Sm^{-1/6}\sqrt{\log(m)}t^{1/6}\lambda^{-1/6}L^{2/7}) \\
&\leq \mathcal{F}_A(\widetilde{\boldsymbol{x}}_t^*) - \left\langle g(\boldsymbol{x}_t; [\boldsymbol{\Theta}_A]_{t-1}), \bar{\boldsymbol{\Theta}}_A - [\boldsymbol{\Theta}_A]_0 \right\rangle + \mathcal{O}(Sm^{-1/6}\sqrt{\log(m)}t^{1/6}\lambda^{-1/6}L^{2/7}) \\
&\quad + \mathcal{O}(m^{-1/6}\sqrt{\log(m)}t^{2/3}\lambda^{-2/3}L^3) \\
&\leq \mathcal{F}_A(\boldsymbol{x}_t^*) - \left\langle g(\boldsymbol{x}_t; [\boldsymbol{\Theta}_A]_{t-1}), \bar{\boldsymbol{\Theta}}_A - [\boldsymbol{\Theta}_A]_0 \right\rangle + \mathcal{O}(Sm^{-1/6}\sqrt{\log(m)}t^{1/6}\lambda^{-1/6}L^{2/7}) \\
&\quad + \mathcal{O}(m^{-1/6}\sqrt{\log(m)}t^{2/3}\lambda^{-2/3}L^3)
\end{aligned}
$$

where the first inequality is due to to the difference between $\widetilde{\mathcal{F}}_A(\widetilde{\boldsymbol{x}}_t^*) - \mathcal{F}_A(\widetilde{\boldsymbol{x}}_t^*)$, and the second inequality is due to the arm pulling mechanism that $\mathcal{F}_A(\widetilde{\boldsymbol{x}}_t^*) \geq \mathcal{F}_A(\boldsymbol{x}_t^*)$. Afterwards, with Lemma J.1 and Lemma J.6, we will also have $\bar{\boldsymbol{\Theta}}_A \in \mathcal{E}_{t-1}$, and the objective can therefore be transformed into

$$
\begin{aligned}
\left|\mathcal{F}_A(\boldsymbol{x}_t) - h_A(\boldsymbol{x}_t)\right| &\leq \left|\widetilde{\mathcal{F}}_A(\boldsymbol{x}_t) - h_A(\boldsymbol{x}_t)\right| + \mathcal{O}(m^{-1/6}\sqrt{\log(m)}t^{2/3}\lambda^{-2/3}L^3) \\
&= \Big| \max_{\boldsymbol{\Theta}_A \in \mathcal{E}_{t-1}} \left\langle g(\boldsymbol{x}_t; [\boldsymbol{\Theta}_A]_{t-1}), \boldsymbol{\Theta}_A - [\boldsymbol{\Theta}_A]_0 \right\rangle - \left\langle g(\boldsymbol{x}_t; [\boldsymbol{\Theta}_A]_{t-1}), \bar{\boldsymbol{\Theta}}_A - [\boldsymbol{\Theta}_A]_0 \right\rangle \Big| \\
&\quad + \mathcal{O}(m^{-1/6}\sqrt{\log(m)}t^{2/3}\lambda^{-2/3}L^3) + \mathcal{O}(Sm^{-1/6}\sqrt{\log(m)}t^{1/6}\lambda^{-1/6}L^{2/7}) \\
&= \Big| \max_{\boldsymbol{\Theta}_A \in \mathcal{E}_{t-1}} \left\langle g(\boldsymbol{x}_t; [\boldsymbol{\Theta}_A]_{t-1}), \boldsymbol{\Theta}_A - [\boldsymbol{\Theta}_A]_{t-1} \right\rangle - \left\langle g(\boldsymbol{x}_t; [\boldsymbol{\Theta}_A]_{t-1}), \bar{\boldsymbol{\Theta}}_A - [\boldsymbol{\Theta}_A]_{t-1} \right\rangle \Big| \\
&\quad + \mathcal{O}(m^{-1/6}\sqrt{\log(m)}t^{2/3}\lambda^{-2/3}L^3) + \mathcal{O}(Sm^{-1/6}\sqrt{\log(m)}t^{1/6}\lambda^{-1/6}L^{2/7}) \\
&\leq \|g(\boldsymbol{x}_t; [\boldsymbol{\Theta}_A]_{t-1})/\sqrt{m}\|_{\boldsymbol{\Sigma}_{t-1}^{-1}} \cdot \sqrt{m} \cdot \left(\|\bar{\boldsymbol{\Theta}}_A - [\boldsymbol{\Theta}_A]_{t-1}\|_{\boldsymbol{\Sigma}_{t-1}} + \max_{\boldsymbol{\Theta}_A \in \mathcal{E}_{t-1}} \|\boldsymbol{\Theta}_A - [\boldsymbol{\Theta}_A]_{t-1}\|_{\boldsymbol{\Sigma}_{t-1}}\right) \\
&\quad + \mathcal{O}(m^{-1/6}\sqrt{\log(m)}t^{2/3}\lambda^{-2/3}L^3) + \mathcal{O}(Sm^{-1/6}\sqrt{\log(m)}t^{1/6}\lambda^{-1/6}L^{2/7}) \\
&\leq 2\gamma_{t-1}(\alpha) \cdot \|g(\boldsymbol{x}_t; [\boldsymbol{\Theta}_A]_{t-1})/\sqrt{m}\|_{\boldsymbol{\Sigma}_{t-1}^{-1}} + \mathcal{O}(m^{-1/6}\sqrt{\log(m)}t^{2/3}\lambda^{-2/3}L^3) \\
&\quad + \mathcal{O}(Sm^{-1/6}\sqrt{\log(m)}t^{1/6}\lambda^{-1/6}L^{2/7})
\end{aligned}
$$

where the second inequality is by applying the Holder's inequality, and the last inequality is by applying Lemmas Lemma 5.2 in (Zhou et al., 2020), Lemma J.1, in terms of the confidence set $\mathcal{E}_{t-1}$. Finally assembling all the components will finish the proof.

$\square$

## I.2 Upper Confidence Bound for the Arm-level Reward Estimation

Recall that in the pseudo-code of our proposed framework $H_2N$-Bandit (Algorithm 1), we utilize the UCB as the exploration strategy for the arm-level model. Here, UCB essentially models the uncertainty of the reward estimation, and measures how much the reward estimation can possibly deviates from the ground-truth expected arm reward. In particular, with the arm-level reward estimation network $f_A(\cdot; \boldsymbol{\Theta}_A)$, with a small probability $\delta > 0$, recall that we aim to have $\mathbb{P}\left(|f_A(\boldsymbol{x}; \boldsymbol{\Theta}_A) - h_A(\boldsymbol{x})| > \mathsf{UCB}_\alpha(\boldsymbol{x})\right) \leq \delta$, where $\mathsf{UCB}_\alpha(\cdot)$ refers

to the UCB associated with the tunable exploration coefficient $\alpha \geq 0$. Here, we formulate the following lemma to derive the exact form of $\mathsf{UCB}_\alpha(\cdot)$. The results in terms of the arm-level UCB will be used as the exploration scores for the arm-level recommendation (line 8-11, Algorithm 1).

**Lemma I.4.** *Suppose the arm-level neural network $f_A(\cdot; [\boldsymbol{\Theta}_A]_{t-1})$ has been trained on chosen arms and their rewards $\{\boldsymbol{x}_\tau, r_\tau\}_{\tau \in [t-1]}$, with $J$-iteration GD and learning rate $\eta$. Meanwhile, $f_A(\cdot; [\boldsymbol{\Theta}_A]_{t-1})$ is a $L$-layer FC network with width $m$. Suppose we have $m, J, \eta_A$ satisfying the conditions in Theorem 5.1. Then, given the candidate arm $\boldsymbol{x} \in \bigcup_{c \in \widetilde{\mathcal{C}}_t} \mathcal{X}_{c,t}$ in the filtered candidate pool, with the probability at least $1 - \delta$, we will have*

$$\left| f_A(\boldsymbol{x}; [\boldsymbol{\Theta}_A]_{t-1}) - h_A(\boldsymbol{x}) \right| \leq \gamma_{t-1}(\alpha) \cdot \| g(\boldsymbol{x}; [\boldsymbol{\Theta}_A]_{t-1})/\sqrt{m} \|_{\boldsymbol{\Sigma}_{t-1}^{-1}} + \mathcal{O}(Sm^{-1/6}\sqrt{\log(m)}t^{1/6}\lambda^{-1/6}L^{2/7})$$
$$+ \mathcal{O}(m^{-1/6}\sqrt{\log(m)}t^{2/3}\lambda^{-2/3}L^3)$$

*By definition in Lemma I.4 and Theorem 5.1, we denote $\mathcal{F}_A(\boldsymbol{x}_t) = f_A(\boldsymbol{x}_t; [\boldsymbol{\Theta}_A]_{t-1}) + \mathsf{UCB}_\alpha(\boldsymbol{x}_t)$, where the upper confidence bond $\mathsf{UCB}_\alpha(\boldsymbol{x}_t) = \gamma_{t-1}(\alpha) \cdot \sqrt{g(\boldsymbol{x}_t; [\boldsymbol{\Theta}_A]_{t-1})^\intercal \boldsymbol{\Sigma}_{t-1}^{-1} g(\boldsymbol{x}_t; [\boldsymbol{\Theta}_A]_{t-1})}$, with the terms*

$$\boldsymbol{\Sigma}_{t-1} = \lambda \mathbf{I} + \sum_{\tau \in [t]} g(\boldsymbol{x}_\tau; [\boldsymbol{\Theta}_A]_{\tau-1}) \cdot g(\boldsymbol{x}_\tau; [\boldsymbol{\Theta}_A]_{\tau-1})^\intercal$$

$$\gamma_{t-1}(\alpha) = \mathcal{O}\left( \alpha \cdot \sqrt{\log \frac{\det(\boldsymbol{\Sigma}_{t-1})}{\det(\lambda \mathbf{I})} - 2\log(\delta)} + \lambda^{1/2}S \right)$$

*with network gradient vector $g(\boldsymbol{x}; \boldsymbol{\Theta}_A) = vec(\nabla_{\boldsymbol{\Theta}} f(\boldsymbol{x}; \boldsymbol{\Theta}_A)) \in \mathbb{R}^{p_A}$.*

**Proof.** Applying the Lemma J.1, we can transform the objective to

$$\left| f_A(\boldsymbol{x}; \boldsymbol{\Theta}_A) - h_A(\boldsymbol{x}) \right| = \left| f_A(\boldsymbol{x}; \boldsymbol{\Theta}_A) - \langle g(\boldsymbol{x}; [\boldsymbol{\Theta}_A]_0), \bar{\boldsymbol{\Theta}}_A - [\boldsymbol{\Theta}_A]_0 \rangle \right|$$
$$\leq \left| f_A(\boldsymbol{x}; \boldsymbol{\Theta}_A) - \langle g(\boldsymbol{x}; [\boldsymbol{\Theta}_A]_{t-1}), \bar{\boldsymbol{\Theta}}_A - [\boldsymbol{\Theta}_A]_0 \rangle \right| + \mathcal{O}(Sm^{-1/6}\sqrt{\log(m)}t^{1/6}\lambda^{-1/6}L^{2/7})$$
$$\leq \left| f_A(\boldsymbol{x}; \boldsymbol{\Theta}_A) - \langle g(\boldsymbol{x}; [\boldsymbol{\Theta}_A]_{t-1}), [\boldsymbol{\Theta}_A]_{t-1} - [\boldsymbol{\Theta}_A]_0 \rangle \right|$$
$$+ \left| \langle g(\boldsymbol{x}; [\boldsymbol{\Theta}_A]_{t-1}), [\boldsymbol{\Theta}_A]_{t-1} - [\boldsymbol{\Theta}_A]_0 \rangle - \langle g(\boldsymbol{x}; [\boldsymbol{\Theta}_A]_{t-1}), \bar{\boldsymbol{\Theta}}_A - [\boldsymbol{\Theta}_A]_0 \rangle \right|$$
$$+ \mathcal{O}(Sm^{-1/6}\sqrt{\log(m)}t^{1/6}\lambda^{-1/6}L^{2/7})$$

where the first equality is to Lemma J.1, the first inequality is due to Lemma J.7 and Lemma J.1, and the last inequality is because of the triangular inequality. Then, we proceed to separately bound the first and second term on the RHS. For the first term, we will have

$$\left| f_A(\boldsymbol{x}; [\boldsymbol{\Theta}_A]_{t-1}) - \langle g(\boldsymbol{x}; [\boldsymbol{\Theta}_A]_{t-1}), [\boldsymbol{\Theta}_A]_{t-1} - [\boldsymbol{\Theta}_A]_0 \rangle \right|$$
$$= \left| f_A(\boldsymbol{x}; [\boldsymbol{\Theta}_A]_{t-1}) - f_A(\boldsymbol{x}; [\boldsymbol{\Theta}_A]_{t-1}) - \langle g(\boldsymbol{x}; [\boldsymbol{\Theta}_A]_{t-1}), [\boldsymbol{\Theta}_A]_{t-1} - [\boldsymbol{\Theta}_A]_0 \rangle \right|$$
$$\leq \mathcal{O}(m^{-1/6}\sqrt{\log(m)}t^{2/3}\lambda^{-2/3}L^3)$$

where the first equality is due to the fact that $f_A(\boldsymbol{x}; [\boldsymbol{\Theta}_A]_0) = 0$ based on our parameter initialization approach, and the inequality is by applying Lemma J.8 and Lemma J.6. Then, for the second term, we will have

$$\left| \langle g(\boldsymbol{x}; [\boldsymbol{\Theta}_A]_{t-1}), [\boldsymbol{\Theta}_A]_{t-1} - [\boldsymbol{\Theta}_A]_0 \rangle - \langle g(\boldsymbol{x}; [\boldsymbol{\Theta}_A]_{t-1}), \bar{\boldsymbol{\Theta}}_A - [\boldsymbol{\Theta}_A]_0 \rangle \right|$$
$$= \left| \langle g(\boldsymbol{x}; [\boldsymbol{\Theta}_A]_{t-1}), \bar{\boldsymbol{\Theta}}_A - [\boldsymbol{\Theta}_A]_{t-1} \rangle \right|$$
$$\leq \| g(\boldsymbol{x}; [\boldsymbol{\Theta}_A]_{t-1})/\sqrt{m} \|_{\boldsymbol{\Sigma}_{t-1}^{-1}} \cdot \sqrt{m} \cdot \| \bar{\boldsymbol{\Theta}}_A - [\boldsymbol{\Theta}_A]_{t-1} \|_{\boldsymbol{\Sigma}_{t-1}}$$
$$\leq \gamma_{t-1}(\alpha) \cdot \| g(\boldsymbol{x}; [\boldsymbol{\Theta}_A]_{t-1})/\sqrt{m} \|_{\boldsymbol{\Sigma}_{t-1}^{-1}}$$

where the first inequality is by applying the Holder's inequality, and the last inequality is by applying Lemmas Lemma 5.2 in (Zhou et al., 2020), Lemma J.1, in terms of the confidence set $\mathcal{E}_{t-1}$ and the fact that $\bar{\boldsymbol{\Theta}}_A \in \mathcal{E}_{t-1}$. Finally, assembling all the components will finish the proof.

$\square$

# J   Lemmas For Arm-level Neural Networks

Similar to the category-level models, we also apply the FC network for the arm-level prediction model. But, since we are applying different initialization approaches for category-level model and arm-level model respectively, the neural network behavior will diverge. Analogously, given the input feature vector $\boldsymbol{x} \in \mathbb{R}^d$, we denote the $L$-layer FC neural network with network width $m$ by

$$f_A(\boldsymbol{x}; \boldsymbol{\Theta}) = \boldsymbol{\Theta}_L (\prod_{l=1}^{L-1} \mathbf{D}_l \boldsymbol{\Theta}_l) \cdot \boldsymbol{x}, \tag{13}$$

where with $\phi$ being the ReLU activation, we define the intermediate hidden representations $\boldsymbol{h}_l, l \in \{0, \ldots, L-1\}$ as

$$\boldsymbol{h}_0 = \boldsymbol{x}, \quad \boldsymbol{h}_l = \phi(\boldsymbol{\Theta}_l \boldsymbol{h}_{l-1}), l \in [L-1].$$

and we also have the binary diagonal matrix functioning as the ReLU activation being

$$\mathbf{D}_l = \mathrm{diag}(\mathbb{I}\{(\boldsymbol{\Theta}_l \boldsymbol{h}_{l-1})_1\}, \ldots, \mathbb{I}\{(\boldsymbol{\Theta}_l \boldsymbol{h}_{l-1})_m\}), l \in [L-1].$$

where $\mathbb{I}(\cdot)$ is the indicator function. Afterwards, the corresponding gradients will naturally become

$$\nabla_{\boldsymbol{\Theta}_l} f_A(\boldsymbol{x}; \boldsymbol{\Theta}) = \begin{cases} [\boldsymbol{h}_{l-1} \boldsymbol{\Theta}_L (\prod_{\tau=l+1}^{L-1} \mathbf{D}_\tau \boldsymbol{\Theta}_\tau)]^\intercal, l \in [L-1] \\ \boldsymbol{h}_{L-1}^\intercal, l = L. \end{cases} \tag{14}$$

For the arm-level model $f_A$, for $l \in [L-1]$, each entry of $\boldsymbol{\Theta}_l$ is drawn from the Gaussian distribution $\mathcal{N}(0, 2/m)$; Each entry of $\boldsymbol{\Theta}_L$ is drawn from the Gaussian distribution $\mathcal{N}(0, 1/m)$.

## J.1   Theoretical Results with NTK Gram Matrices

To begin with, we will first introduce some results, in order to link the NTK matrices (Def. 1 and Def. 2) with the reward mapping function $h(\cdot)$ and the gradient covariance matrix $\boldsymbol{\Sigma}$ (Algorithm 1).

**Lemma J.1.** *There exist a positive constant $C > 0$ such that with probability at least $1 - \delta$, if network width $m$ satisfies the condition in Theorem 5.1, for any $\boldsymbol{x} \in \mathcal{A}_T$, there exists a set of parameters $\bar{\boldsymbol{\Theta}}$ such that*

$$h(\boldsymbol{x}) = \langle g(\boldsymbol{x}; \boldsymbol{\Theta}_0), \bar{\boldsymbol{\Theta}} - \boldsymbol{\Theta}_0 \rangle \tag{15}$$

*where parameters $\bar{\boldsymbol{\Theta}}$ satisfy $\|\bar{\boldsymbol{\Theta}} - \boldsymbol{\Theta}_0\| \leq S/\sqrt{m}$, along with the NTK norm $S \geq \sqrt{2 \check{\boldsymbol{h}}^\intercal \check{\mathbf{M}}^{-1} \check{\boldsymbol{h}}}$, where the expected reward vector $\check{\boldsymbol{h}} = [h(\boldsymbol{x})]_{\boldsymbol{x} \in \check{\mathcal{A}}_T} \in \mathbb{R}^{|\check{\mathcal{A}}_T|}$.*

**Proof.** The proof of this lemma is inspired by that of Lemma 5.1 in (Zhou et al., 2020). However, we build our proof upon the non-duplicate arms $\check{\mathcal{A}}_T$ and the corresponding NTK Gram matrix $\check{\mathbf{M}}$ (Def. 2), instead of imposing the full-rank assumption on the conventional NTK matrix $\mathbf{M}$ (Def. 1). Here, we recall that the matrix $\check{\mathbf{M}}$ is naturally positive definite ($\check{\lambda}_0 = \lambda_{\mathsf{min}}(\check{\mathbf{M}}) > 0$), as it is the NTK Gram matrix built upon a set of distinct arms.

Then, consider the gradient matrix with no-duplicate arms $\check{\mathbf{G}} = [g(\boldsymbol{x}; \boldsymbol{\Theta}_0)]_{\boldsymbol{x} \in \check{\mathcal{A}}_T} / \sqrt{m} \in \mathbb{R}^{p \times |\check{\mathcal{A}}_T|}$, where $p$ represents the total number of parameters in the neural network. As a result, by applying conclusion from Lemma J.3 and due to the fact that $|\check{\mathcal{A}}_T| \leq 3T$, with the network width $m \geq \Omega(L^6 \log(TL/\delta)/\epsilon)$, $\forall \epsilon > 0$ and the probability at least $1 - \delta$, we will have

$$\|\check{\mathbf{G}}^\intercal \check{\mathbf{G}} - \check{\mathbf{M}}\|_F \leq |\check{\mathcal{A}}_T| \cdot \epsilon.$$

By setting $\epsilon = \frac{\check{\lambda}_0}{2|\check{\mathcal{A}}_T|}$, we will have

$$\check{\mathbf{G}}^\intercal \check{\mathbf{G}} \succeq \check{\mathbf{M}} - \|\check{\mathbf{G}}^\intercal \check{\mathbf{G}} - \check{\mathbf{M}}\|_F \mathbf{I} \succeq \check{\mathbf{M}} - \check{\lambda}_0/2 \mathbf{I} \succeq \check{\mathbf{M}}/2 \succ \mathbf{0}.$$

where the last two inequalities are due to the fact that $\breve{\mathbf{M}} \succeq \breve{\lambda}_0 \mathbf{I} \succ \mathbf{0}$. Analogous to Lemma 5.1 in (Zhou et al., 2020), we consider the singular value decomposition of $\breve{\mathbf{G}}$ being $\breve{\mathbf{G}} = \breve{\mathbf{P}}\breve{\mathbf{A}}\breve{\mathbf{Q}}^\mathsf{T}$, where we naturally have $\breve{\mathbf{A}} \succ 0$ since $\breve{\mathbf{M}}$ is positive definite. Then, with the expected reward vector $\breve{\boldsymbol{h}}$ (Def. 2), we have

$$\breve{\boldsymbol{h}} = (\breve{\mathbf{Q}}\breve{\mathbf{A}}\breve{\mathbf{P}}^\mathsf{T}) \cdot (\breve{\mathbf{P}}\breve{\mathbf{A}}^{-1}\breve{\mathbf{Q}}^\mathsf{T}) \cdot \breve{\boldsymbol{h}} = \sqrt{m} \cdot \breve{\mathbf{G}}^\mathsf{T}(\bar{\boldsymbol{\Theta}} - \boldsymbol{\Theta}_0)$$

by considering there exists a set of parameters $\bar{\boldsymbol{\Theta}} = \boldsymbol{\Theta}_0 + (\breve{\mathbf{P}}\breve{\mathbf{A}}^{-1}\breve{\mathbf{Q}}^\mathsf{T}) \cdot \breve{\boldsymbol{h}}/\sqrt{m}$.

Therefore, since $\breve{\boldsymbol{h}} = \sqrt{m} \cdot \breve{\mathbf{G}}^\mathsf{T}(\bar{\boldsymbol{\Theta}} - \boldsymbol{\Theta}_0)$, we will have $\forall \boldsymbol{x} \in \breve{\mathcal{A}}_T$,

$$h(\boldsymbol{x}) = \langle g(\boldsymbol{x}; \boldsymbol{\Theta}_0), \bar{\boldsymbol{\Theta}} - \boldsymbol{\Theta}_0 \rangle.$$

Meanwhile, since $\breve{\mathbf{G}}^\mathsf{T}\breve{\mathbf{G}} \succeq \breve{\mathbf{M}}/2$, we will have the Euclidean distance

$$m \cdot \|\bar{\boldsymbol{\Theta}} - \boldsymbol{\Theta}_0\|_2^2 = \boldsymbol{h}^\mathsf{T}\breve{\mathbf{Q}}\breve{\mathbf{A}}^{-1}\breve{\mathbf{P}}^\mathsf{T} \cdot \breve{\mathbf{P}}\breve{\mathbf{A}}^{-1}\breve{\mathbf{Q}}^\mathsf{T}\boldsymbol{h} = \boldsymbol{h}^\mathsf{T}(\breve{\mathbf{G}}^\mathsf{T}\breve{\mathbf{G}})^{-1}\boldsymbol{h} \leq 2\boldsymbol{h}^\mathsf{T}\breve{\mathbf{M}}^{-1}\boldsymbol{h}.$$

Finally, since the above results holds $\forall \boldsymbol{x} \in \breve{\mathcal{A}}_T$, due to the fact that $\breve{\mathcal{A}}_T$ contains all the *unique arms* of the collection $\mathcal{A}_T$, we will directly have the above results regarding parameters $\bar{\boldsymbol{\Theta}}$ feasible $\forall \boldsymbol{x} \in \mathcal{A}_T$. This completes the proof.

$\square$

**Lemma J.2.** *Suppose $m$ satisfies the conditions in Theorem 5.1. Suppose the gradient matrix with randomly initialized parameters is $\boldsymbol{\Sigma}^{(0)} = \lambda \mathbf{I} + \sum_{\boldsymbol{x} \in \mathcal{A}} g(\boldsymbol{x}; \boldsymbol{\Theta}_0) \cdot g(\boldsymbol{x}; \boldsymbol{\Theta}_0)^\mathsf{T}/m$, upon an arbitrary subset $\mathcal{A} \subseteq \mathcal{A}_T$ of arm collection $\mathcal{A}_T$. With probability at least $1 - \delta$ over the initialization, the result holds:*

$$\log\left(\frac{\det \boldsymbol{\Sigma}^{(0)}}{\det \lambda \mathbf{I}}\right) \leq \widetilde{d}\log(1 + TK/\lambda) + 1.$$

**Proof.** First, recall that we have $\mathcal{A}_T$ as the arm collection of: (i) the chosen arms $\{\boldsymbol{x}_t\}_{t=1}^T$; (ii) the optimal arms $\{\boldsymbol{x}_t^*\}_{t=1}^T$; (iii) and the imaginary ones $\{\widetilde{\boldsymbol{x}}_t\}_{t=1}^T$ chosen by the corruption-free model. This makes its cardinality $|\mathcal{A}_T| = 3T$. Thus, for the left hand side, we have

$$\log\frac{\det(\boldsymbol{\Sigma}^{(0)})}{\det(\lambda \mathbf{I})} \leq \log\det(\lambda \mathbf{I} + \sum_{\boldsymbol{x} \in \mathcal{A}_T} g(\boldsymbol{x}; \boldsymbol{\Theta}_0)g(\boldsymbol{x}; \boldsymbol{\Theta}_0)^\mathsf{T}/m) = \det(\lambda \mathbf{I} + \mathbf{G}_0\mathbf{G}_0^\mathsf{T}),$$

where we define gradient matrix $\mathbf{G}_0 = \left[g(\boldsymbol{x}; \boldsymbol{\Theta}_0)/\sqrt{m}\right]_{\boldsymbol{x} \in \mathcal{A}_T} \in \mathbb{R}^{p \times (3T)}$. Here, based on Lemma J.3, we can bound the distance between the Gradient matrix product $\mathbf{G}_0^\mathsf{T}\mathbf{G}_0$ and the NTK matrix $\mathbf{M}$ (Def. 1), as

$$\|\mathbf{G}_0^\mathsf{T}\mathbf{G}_0 - \mathbf{M}\| \leq 3T \cdot \frac{1}{3T \cdot \mathcal{O}(\sqrt{T}/\lambda)} = \frac{1}{\mathcal{O}(\sqrt{T}/\lambda)},$$

by setting $\epsilon = \frac{1}{3T \cdot \mathcal{O}(\sqrt{T}/\lambda)}$. The above results will hold, as long as we have the network width $m \geq \Omega((TL)^6 \log(TL/\delta)/\lambda^4)$, matching the conditions in Theorem 5.1. As a result, we can have

$$\log\det(\mathbf{I} + \mathbf{G}_0^\mathsf{T}\mathbf{G}_0/\lambda)$$
$$= \log\det(\mathbf{I} + \mathbf{H}/\lambda + (\mathbf{G}_0^\mathsf{T}\mathbf{G}_0 - \mathbf{H})/\lambda)$$
$$\leq \log\det(\mathbf{I} + \mathbf{H}/\lambda) + \langle(\mathbf{I} + \mathbf{H}/\lambda)^{-1}, (\mathbf{G}_0^\mathsf{T}\mathbf{G}_0 - \mathbf{H})/\lambda\rangle$$
$$\leq \log\det(\mathbf{I} + \mathbf{H}/\lambda) + \|(\mathbf{I} + \mathbf{H}/\lambda)^{-1}\|_F\|\mathbf{G}_0^\mathsf{T}\mathbf{G}_0 - \mathbf{H}\|_F/\lambda$$
$$\leq \log\det(\mathbf{I} + \mathbf{H}/\lambda) + \mathcal{O}(\sqrt{T}/\lambda) \cdot \|\mathbf{G}_0^\mathsf{T}\mathbf{G}_0 - \mathbf{H}\|_F$$
$$\leq \log\det(\mathbf{I} + \mathbf{H}/\lambda) + 1$$
$$= \widetilde{d}\log(1 + TK/\lambda) + 1.$$

The first inequality is because the concavity of $\log\det(\cdot)$ function; The third inequality is due to $\|(\mathbf{I} + \mathbf{H}\lambda)^{-1}\|_F \leq \|\mathbf{I}^{-1}\|_F \leq \sqrt{T}$; The fourth inequality is by applying the above distance upper bound $\|\mathbf{G}_0^\mathsf{T}\mathbf{G}_0 -$

$\mathbf{M}\| \leq \frac{1}{\mathcal{O}(\sqrt{T}/\lambda)}$. The last inequality is because of the choice the $m$; The last equality is because of the Definition of $\tilde{d}$. The proof is completed.

$\square$

**Lemma J.3.** *With the randomly initialized network parameters $\mathbf{\Theta}_0 \in \mathbb{R}^p$ and a collection of arms $\mathcal{A} \subset \mathbb{R}^d$, define the gradient matrix $\mathbf{G}_{\mathcal{A}} = [g(\boldsymbol{x}; \mathbf{\Theta}_0)]_{\boldsymbol{x} \in \mathcal{A}} \in \mathbb{R}^{p \times |\mathcal{A}|}$. Following the recursive procedure in Def. 1, construct the NTK Gram matrix $\mathbf{M}_{\mathcal{A}}$ based on arms $\mathcal{A}$. Then, with the probability at least $1 - \delta$, we will have*

$$\|\mathbf{G}_{\mathcal{A}}^{\intercal} \mathbf{G}_{\mathcal{A}} - \mathbf{M}_{\mathcal{A}}\|_F \leq |\mathcal{A}| \cdot \epsilon,$$

*with the network width $m \geq \Omega(L^6 \log(|\mathcal{A}|L/\delta)/\epsilon^4)$.*

**Proof.** The proof of this lemma is analogous to the proof of Lemma B.1 in (Zhou et al., 2020). Based on Theorem 3.1 from (Arora et al., 2019), we have that for any two arms $\boldsymbol{x}, \boldsymbol{x}' \in \mathcal{A}$, if the network width $m \geq \Omega(L^6 \log(L/\delta)/\epsilon^4)$, we will have $|\langle g(\boldsymbol{x}; \mathbf{\Theta}_0), g(\boldsymbol{x}'; \mathbf{\Theta}_0)\rangle/m - \mathbf{M}_{\mathcal{A}}[\boldsymbol{x}, \boldsymbol{x}']| \leq \epsilon$, where $\mathbf{M}_{\mathcal{A}}[\boldsymbol{x}, \boldsymbol{x}']$ represents the element in NTK Gram matrix $\mathbf{M}_{\mathcal{A}}$ that corresponds to arms $\boldsymbol{x}$ and $\boldsymbol{x}'$. Next, taking the union bound over all the arms in $\mathcal{A}$, we will have

$$\|\mathbf{G}_{\mathcal{A}}^{\intercal} \mathbf{G}_{\mathcal{A}} - \mathbf{M}_{\mathcal{A}}\|_F = \sqrt{\sum_{\boldsymbol{x} \in \mathcal{A}} \sum_{\boldsymbol{x}' \in \mathcal{A}} |\langle g(\boldsymbol{x}; \mathbf{\Theta}_0), g(\boldsymbol{x}'; \mathbf{\Theta}_0)\rangle/m - \mathbf{M}_{\mathcal{A}}[\boldsymbol{x}, \boldsymbol{x}']|^2} \leq |\mathcal{A}| \cdot \epsilon,$$

as long as the network width $m \geq \Omega(L^6 \log(|\mathcal{A}|L/\delta)/\epsilon^4)$.

$\square$

## J.2 Additional Theoretical Results under Over-parameterization

**Lemma J.4.** *There exist a positive constant $C > 0$ such that with probability at least $1 - \delta$, if $m \geq CT^4L^6 \log(T^2L/\delta)/\lambda^4$ for any $\boldsymbol{x} \in \bigcup_{\tau \in [t]} \mathcal{X}_\tau$, there exists a set of parameters $\bar{\mathbf{\Theta}}$ such that with the neural parameters $\mathbf{\Theta}_{t-1}$ trained on $\{\boldsymbol{x}_\tau\}_{\tau=1}^{t-1}$, we have*

$$|\langle g(\boldsymbol{x}; \mathbf{\Theta}_0), \bar{\mathbf{\Theta}} - \mathbf{\Theta}_0\rangle - \langle g(\boldsymbol{x}; \mathbf{\Theta}_{t-1}), \bar{\mathbf{\Theta}} - \mathbf{\Theta}_0\rangle| \leq \mathcal{O}(Sm^{-1/6}\sqrt{\log(m)}t^{1/6}\lambda^{-1/6}L^{2/7})$$

*where parameters $\bar{\mathbf{\Theta}}$ satisfy $\|\bar{\mathbf{\Theta}} - \mathbf{\Theta}_0\| \leq S/\sqrt{m}, S > 0$ as shown in Lemma J.1.*

**Proof.** This lemma is based on Lemma J.1. Here, our objective can be reformed into

$$\begin{aligned}|\langle g(\boldsymbol{x}; \mathbf{\Theta}_0), \bar{\mathbf{\Theta}} - \mathbf{\Theta}_0\rangle - \langle g(\boldsymbol{x}; \mathbf{\Theta}_{t-1}), \bar{\mathbf{\Theta}} - \mathbf{\Theta}_0\rangle| &= \|\bar{\mathbf{\Theta}} - \mathbf{\Theta}_0\|_2 \cdot \left(g(\boldsymbol{x}; \mathbf{\Theta}_0) - g(\boldsymbol{x}; \mathbf{\Theta}_{t-1})\right) \\ &\leq S/\sqrt{m} \cdot \left(g(\boldsymbol{x}; \mathbf{\Theta}_0) - g(\boldsymbol{x}; \mathbf{\Theta}_{t-1})\right) \\ &\leq \mathcal{O}(Sm^{-1/6}\sqrt{\log(m)}t^{1/6}\lambda^{-1/6}L^{2/7}),\end{aligned}$$

where the first inequality is due to Lemma J.1, and the second inequality is due to Lemmas J.5, J.6, and Lemma J.7.

$\square$

**Lemma J.5** (Lemma B.3 in (Cao & Gu, 2019) )**.** *There exist constants $\{C_i\}_{i=1}^2$ such that for any $\delta > 0$, if we have*

$$\omega \leq C_1 L^{-6}(\log m)^{-3/2},$$

*then, with probability at least $1 - \delta$, for any $\|\mathbf{\Theta} - \mathbf{\Theta}_0\| \leq \omega$ and $\boldsymbol{x}_t \in \{\boldsymbol{x}_t\}_{t=1}^T$ we have $\|g(\boldsymbol{x}_t; \mathbf{\Theta})\|_2 \leq C_2\sqrt{mL}$.*

**Lemma J.6** (Lemma B.2 in (Zhou et al., 2020) )**.** *For the $L$-layer full-connected network $f$ trained with $J$-iteration GD, there exist constants $\{C_i\}_{i=1}^5 \geq 0$ such that for $\delta > 0$, if for all $t \in [T]$, $\eta, m$ satisfy*

$$\begin{aligned}2\sqrt{t/(m\lambda)} &\geq C_1 m^{-3/2} L^{-3/2}[\log(TL^2/\delta)]^{3/2}, \\ 2\sqrt{t/(m\lambda)} &\leq C_2 \min\{L^{-6}[\log m]^{-3/2}, (m(\lambda\eta)^2 L^{-6} t^{-1}(\log m)^{-1})^{3/8}\}, \\ \eta &\leq C_3(m\lambda + tmL)^{-1}, \\ m^{1/6} &\geq C_4\sqrt{\log m}L^{7/2}t^{7/6}\lambda^{-7/6}(1 + \sqrt{t/\lambda}),\end{aligned}$$

*then, with probability at least $1 - \delta$, we have*

$$\|\mathbf{\Theta}_t - \mathbf{\Theta}_0\| \leq 2\sqrt{t/(m\lambda)}$$
$$\|\mathbf{\Theta}_t - \mathbf{\Theta}_0 - \mathbf{\Theta}'\| \leq (1 - \eta m\lambda)^{J/2}\sqrt{t/(m\lambda)} + C_5 m^{-2/3}\sqrt{\log m}L^{7/2}t^{5/3}\lambda^{-5/3}(1 + \sqrt{t/\lambda}).$$

**Lemma J.7** (Theorem 5 in (Allen-Zhu et al., 2019)). *With probability at least $1 - \delta$, there exist constants $C_1, C_2$ such that if $\omega \leq C_1 L^{-9/2}\log^{-3} m$, for $\|\mathbf{\Theta}_t - \mathbf{\Theta}_0\|_2 \leq \omega$, we have*

$$\|g(\boldsymbol{x}_t; \mathbf{\Theta}_t) - g(\boldsymbol{x}_t; \mathbf{\Theta}_0)\|_2 \leq C_2\sqrt{\log m}\omega^{1/3}L^3\|g(\boldsymbol{x}_t; \mathbf{\Theta}_0)\|_2.$$

**Lemma J.8** ( Lemma 4.1 in (Cao & Gu, 2019) ). *There exist constants $\{\bar{C}_{i=1}^3\} \geq 0$ such that for any $\delta \geq 0$, if $\tau$ satisfies that*

$$\tau \leq \bar{C}_2 L^{-6}[\log m]^{-3/2},$$

*then with probability at least $1 - \delta$, for all $\mathbf{\Theta}^1, \mathbf{\Theta}^2$ satisfying $\|\mathbf{\Theta}^1 - \mathbf{\Theta}_0\| \leq \tau, \|\mathbf{\Theta}^2 - \mathbf{\Theta}_0\| \leq \tau$ and for any $\boldsymbol{x}t \in \{\boldsymbol{x}t\}_{t=1}^T$, we have*

$$|f(\boldsymbol{x}; \mathbf{\Theta}^1) - f(\boldsymbol{x}; \mathbf{\Theta}^2) - \langle(g(\boldsymbol{x}; \mathbf{\Theta}^2), \mathbf{\Theta}^1 - \mathbf{\Theta}^2)\rangle| \leq \bar{C}_3 \tau^{4/3} L^3 \sqrt{m\log m}.$$

**Lemma J.9** ( Lemma 11 in (Abbasi-Yadkori et al., 2011), Lemma B.7 in (Zhou et al., 2020)). *Suppose a sequence of arms $\{\boldsymbol{x}'_\tau\}_{\tau \in [t]}$, with an arbitrary arm $\boldsymbol{x}'_\tau \in \mathcal{X}_\tau$ from each time step $\tau \in [t]$. The gradient matrix is denoted by $\mathbf{\Sigma}_t = \lambda\mathbf{I} + \sum_{\tau \in [t]} g(\boldsymbol{x}'_\tau; \mathbf{\Theta}_{\tau-1}) \cdot g(\boldsymbol{x}'_\tau; \mathbf{\Theta}_{\tau-1})^\mathsf{T}/m$. Then, we can have*

$$\sum_{\tau \in [t]} \min\{\|g(\boldsymbol{x}_\tau; \mathbf{\Theta}_{\tau-1}/\sqrt{m})\|^2_{\mathbf{\Sigma}_{\tau-1}^{-1}}, 1\} \leq 2\log\frac{\det(\mathbf{\Sigma}_t)}{\det(\lambda\mathbf{I})}.$$

**Lemma J.10.** *Suppose $m$ satisfies the conditions in Theorem 5.1. Suppose the gradient matrix can be represented by $\mathbf{\Sigma} = \lambda\mathbf{I} + \sum_{t \in [T]} g(\boldsymbol{x}_{c,t}^{(i)}; \mathbf{\Theta}_{t-1}) \cdot g(\boldsymbol{x}_{c,t}^{(i)}; \mathbf{\Theta}_{t-1})^\mathsf{T}/m$, with the arm $\boldsymbol{x}_{c,t}^{(i)} \in \{\boldsymbol{x}_t, \boldsymbol{x}_t^*\}$ from each time step $t$. With probability at least $1 - \delta$ over the initialization, the result holds:*

$$\|\mathbf{\Sigma}\|_2 \leq \lambda + \mathcal{O}(TL),$$
$$\|\mathbf{\Sigma} - \mathbf{\Sigma}^{(0)}\|_F \leq \mathcal{O}(m^{-1/6}\sqrt{\log(m)}L^4 t^{7/6}\lambda^{-1/6})$$
$$\left|\log\frac{\det(\mathbf{\Sigma})}{\det(\lambda\mathbf{I})} - \log\frac{\det(\mathbf{\Sigma}^{(0)})}{\det(\lambda\mathbf{I})}\right| \leq \mathcal{O}(m^{-1/6}\sqrt{\log(m)}L^4 t^{5/6}\lambda^{-1/6})$$

*where the matrix with randomly initialized parameters is $\mathbf{\Sigma}^{(0)} = \lambda\mathbf{I} + \sum_{t \in [T]} g(\boldsymbol{x}_{c,t}^{(i)}; \mathbf{\Theta}_0) \cdot g(\boldsymbol{x}_{c,t}^{(i)}; \mathbf{\Theta}_0)^\mathsf{T}/m$.*

*Proof.* Based on the Lemma J.5, for any $t \in [T]$, $\|g(\boldsymbol{x}_{c,t}^{(i)}; \mathbf{\Theta}_0)\|_2 \leq \mathcal{O}(\sqrt{mL})$. Then, for the first item:

$$\|\mathbf{\Sigma}^{(0)}\|_2 = \|\lambda\mathbf{I} + \sum_{t=1}^T g(\boldsymbol{x}_{c,t}^{(i)}; \mathbf{\Theta}_0)g(\boldsymbol{x}_{c,t}^{(i)}; \mathbf{\Theta}_0)^\mathsf{T}/m\|_2$$

$$\leq \|\lambda\mathbf{I}\|_2 + \|\sum_{t=1}^T g(\boldsymbol{x}_{c,t}^{(i)}; \mathbf{\Theta}_0)g(\boldsymbol{x}_{c,t}^{(i)}; \mathbf{\Theta}_0)^\mathsf{T}/m\|_2$$

$$\leq \lambda + \sum_{t=1}^T \|g(\boldsymbol{x}_{c,t}^{(i)}; \mathbf{\Theta}_0)/\sqrt{m}\|_2^2 \leq \lambda + \mathcal{O}(TL).$$

Then, the second and third inequalities in this lemma are the direct application of Lemma B.3 of (Zhou et al., 2020). The proof is completed.

$\square$

