# OpenReview forum: "Bi-level Hierarchical Neural Contextual Bandits for Online Recommendation"
_TMLR — Accepted by TMLR_

### Review · Reviewer_rT1b · 2025-09-12

**Summary Of Contributions:**

The paper introduces $\mathrm{H}^2\mathrm{N}$-Bandit, a bi-level hierarchical neural framework for stochastic contextual bandits in online recommendation systems. It exploits arm category information (e.g., genres in MovieLens) to first select $B$ promising categories via neural exploitation/exploration nets, then recommends arms within the filtered pool using UCB. Pseudo-rewards from arm UCB train category models, enabling efficient exploration without true labels. Contributions: (1) Bi-level neural formulation for arbitrary rewards; (2) First stochastic hierarchical neural bandit regret bound $\tilde{O}(\sqrt{T \tilde{d}})$ and category selection guarantee (Theorems 5.1-5.2); (3) Experiments on 3 datasets showing superior performance and runtime against 9 baselines; (4) Extension to three-level hierarchies. Overall, the authors have provided a very well written paper, with extensive theoretical grounding, and comprehensive experimental investigation. Solid work.

**Additional Comments:**

The $\mathrm{H}^2\mathrm{N}$-Bandit framework shines on computational efficiency, but could sample efficiency be improved? Could latent bandits ideas (e.g., Hong et al. 2020) be integrated to infer hidden states within categories, potentially helping with sample efficiency by modeling latent user preferences?

**Audience:**

Yes

**Audience Explanation:**

TMLR's RL and bandit theory audience would greatly value this work. Also potentially appealing to recsys communities.

**Claims And Evidence:**

Yes

**Claims Explanation:**

- **Theoretical Support:** Claims of $\tilde{O}(\sqrt{T \tilde{d}})$ regret (Sec. 5) are rigorously derived via NTK/martingale, handling arbitrary rewards without priors (novel vs. linear hierarchies like Hong 2022); category guarantee (Theorem 5.2) under mild $\zeta$-margin; Comprehensive proofs in appendix
- **Empirical Validation:** Experiments (Sec. 6) convincingly show lower runtime vs. baselines on 3 datasets (Figs. 1-3, Tables 1-2) while competitive or outperforming in regret; further experiments confirm bi-level value (e.g., no category exp. worsens performance).
- **Clarity:** The whole paper: formulation, theoretical analysis and empriical results are clear and precise

**Requested Changes:**

None

---

> ### Author Response · Authors · 2025-10-08
> **Thank you for the thoughtful review! (1/2)**
>
> We sincerely thank the reviewer for the valuable comments and suggestions, which have been instrumental in enhancing the quality of our paper. We will try to address your feedback in the following Q\&A format. Please also refer to our **updated manuscript for added experiments**.
> *Thank you again for your thoughtful review!*
>
>
>
>
>
> ----------------------
>
>
>
> > Q1: Could latent bandits ideas (e.g., Hong et al. 2020) be integrated to infer hidden states within categories, potentially helping with sample efficiency by modeling latent user preferences?
>
>
> Thank you for this great and insightful question.
> It is possible to integrate latent bandit ideas to further improve sample efficiency, and we will briefly offer some high-level, initial insights and discussions below. However, since this is a non-trivial topic that can constitute a distinct line of research, *it is beyond the scope of this paper and we propose it as a promising future direction of this work*.
>
>
>
>
> **Connecting to latent bandits and an integration.**
> (Hong et al., 2022) leverage hierarchical/latent structure to share information across related sub-trees, which can improve sample efficiency when the latent structure is informative. While we deliberately avoid parametric priors on reward in our base method, the core intuition: pooling evidence via latent structure to generalize faster, can be compatible with our framework. Below is an intuitive variant that models latent and within-category structure while preserving our computational budget.
>
>
>
> - **Latent knowledge within a category.**
>   For each category at round $t$, we can posit a small latent vector that represents fine-grained user preferences or item subtype(s) composition inside that category (e.g., "sporty" vs. "formal" within shoes). We estimate this latent vector from the current items in the category using a light encoder (a small network that summarizes the set via pooling or attention). This step is generally affordable: the encoding itself does not require expensive per-arm gradient computations.
>
> - **Latent-aware category representation.**
>   Instead of a single category embedding, we build a summary that reflects how the category looks on average under the latent view, and how diverse it appears. Concretely, we feed simple item-feature statistics (e.g., mean/standard deviation as in this work) together with the latent vector into our existing exploitation $f_C^{(1)}$ and exploration $f_C^{(2)}$ networks, with no external interface changes.
>
> - **Information-gain bonus on the latent.**
>   We can include an additional exploration bonus that favors categories where a pull is expected to teach us the most about the latent vector (i.e., reduce uncertainty about the latent facets). Intuitively, if feedback from a category is likely to shrink uncertainty in its hidden structure, that category receives a higher score. This focuses exploration where learning is fastest, not just where predicted reward is high or noisy.
>
> - **Arm-level reuse without extra enumeration.**
>   After ranking categories with the latent-aware score, we keep our existing arm-level UCB step. Optionally, we can condition arm uncertainty on a sampled latent to tighten bonuses in well-understood category facets and loosen them where we are unsure. This modification does not require any additional arm enumeration beyond our current procedure and preserves a key advantage of our framework: avoiding computationally expensive per-arm gradient exploration.
>
> - **Lightweight training via existing signals.**
>   The encoder is trained with the same supervision signals we already compute (our pseudo-rewards). A simple objective encourages the latent-conditioned summary to agree with these signals, plus a light regularizer to keep the latent well-behaved. In practice, this adds only a small, constant overhead per category.
>
>
>
> **Sample efficiency.**
> (i) Within a category, uncertainty can decompose into arm noise and category latent facets. The new bonus focuses pulls where epistemic uncertainty shrinks fastest, accelerating generalization. (ii) Evidence transfers across arms that share a latent facet, reducing the number of pulls needed to down-weight unpromising facets or to lock onto promising ones, which is expected to improve the category-level recommendation efficiency. (iii) All computations happen at the category level before arm enumeration; the category-level cost generally remains $O(|\mathcal{C}_t|)$, and the arm-level cost is largely unchanged because it still operates on the filtered set $\tilde{X}_t$.

---

> ### Author Response · Authors · 2025-10-08
> **Thank you for the thoughtful review! (2/2)**
>
> **Theoretical outlook.**
> Our existing analysis is also expected to yield a $\tilde{O}(\sqrt{T})$-style regret with margin-based tightening, once the optimal category is consistently selected.
> On the other hand, the latent variant suggested can potentially help replace the effective dimension by a different latent effective dimension $\tilde{d}_{\text{latent}} \le \tilde{d}$ once the latent encoder concentrates, plus an additive term controlled by cumulative information gain over the latent.
> A formal proof would extend our confidence-ellipsoid arguments with a posterior-concentration layer for the encoder, as well as an information-gain-related term.
> Therefore, such a research topic is beyond the scope of this paper and is a promising direction for future work.

---

### Review · Reviewer_gBv8 · 2025-09-13

**Summary Of Contributions:**

The paper proposes a bi-level hierarchical neural contextual bandit (“H2N-Bandit”). It first scores and selects a small set of promising categories, then chooses an arm only within those categories. Category “reward” is defined as the class’s best arm, aligning the category decision with the final arm decision. A UCB-style uncertainty term at the arm level and a pseudo-reward for categories enable end-to-end training. The authors provide regret bounds and show competitive regret with notable speedups on standard recommendation datasets.

**Additional Comments:**

**Strengths.**

* Clear motivation for large-K settings; the category→arm decomposition is natural and practically useful.
* Objective alignment (class reward = within-class max) is thoughtful and well-argued.
* Theory mirrors the two-stage design and helps justify the approach; empirical results indicate meaningful runtime gains.

**Weaknesses.**

* The pseudo-reward (max of prediction + UCB within a chosen class) risks optimism feedback and biasing the category model, while unchosen classes receive little corrective signal.
* Category representations (mean/std of contexts) may miss rare, high-value arms; stronger set encoders or quantile/top-k features could help.
* Some assumptions (over-parameterization, class-margin conditions) are strong; the dependence of network width/complexity on K needs clearer discussion.
* Limited ablations on (i) category features, (ii) removing the exploration head, and (iii) robustness to noisy/overlapping categories and very large K.

**Questions for authors.**

1. How sensitive is performance to B, and do you have curves showing when the optimal class consistently enters the shortlist?
2. What numerical safeguards are used for the arm-level UCB (e.g., covariance updates, regularization)?
3. Does injecting conservative pseudo-labels for a few non-selected classes reduce optimism bias?

**Suggestions.**

* Add ablations on category encoding and the exploration head; test robustness to perturbed/merged categories.

**Audience:**

Yes

**Audience Explanation:**

Researchers in online learning and bandits will likely be interested in this paper.

**Broader Impact Concerns:**

No specific broader impact concerns identified; the work is an algorithmic refinement with a risk profile similar to existing contextual bandit methods.

**Claims And Evidence:**

Yes

**Claims Explanation:**

The claims made in the submission are supported by accurate, convincing and clear evidence.

**Requested Changes:**

See additional comments

---

> ### Author Response · Authors · 2025-10-08
> **Thank you for the thoughtful review! (1/5)**
>
> We sincerely thank the reviewer for the valuable comments and suggestions, which have been instrumental in enhancing the quality of our paper. We will try to address your feedback in the following Q\&A format. Please also refer to our **updated manuscript for added experiments**.
> *Thank you again for your thoughtful review!*
>
>
>
>
> -----------------------------------
>
>
> > Q1: Some assumptions (over-parameterization, class-margin conditions) are strong? The dependence of network width/complexity on K?
>
>
>
>
> **Over-parameterization.**
> We would like to mention that our analysis follows the standard NTK/over-parameterized regime that is common and indispensable in neural bandit theoretical analysis (e.g., [1,2,3,4,5]), since the neural needs to be powerful enough to be able to approximate an arbitrary reward function.
> Here, while our regret bound (Thm. 5.1) is proved for $L$-layer FC networks with width $m$ sufficiently large, the proof techniques can be readily extended to other over-parameterized architectures such as CNNs and ResNets (cited in Sec. 5.2). The requirement is analytical (e.g., to control confidence ellipsoids), not a design constraint of the algorithm.
> Meanwhile, we also remove the common “arm separateness” assumption by working with NTK Gram matrices over non-duplicate arms (Remark 2), which makes our theoretical analysis for the neural hierarchical bandit setting more general.
>
>
> **Margin/separability at the category level.**
> We would like to mention that *we do not assume strong separateness globally*. The additional $\zeta$-margin is a mild assumption that appears only in our category-selection guarantee (Thm. 5.2), where it is used to distinguish the optimal category from the second-best ones, while achieving a potentially tighter regret bound (Discussion below Thm. 5.2). Specifically, with the optimal category $c\_{t}^* = \arg\max\_{c\in \mathcal{C}\_{t}} h\_{C}(\mathbf{\nu}\_{c, t})$ and second-optimal category $c\_{t}^\circ$, we suppose there exists a constant gap $\zeta > 0$ such that $| \mathbb{E}[r\_{c_{t}^\*, t} | \mathcal{X}\_{c_{t}^\*, t}] - \mathbb{E}[r\_{c_{t}^\circ, t} | \mathcal{X}\_{c_{t}^\circ, t}] | \geq \zeta$.
>
> This condition is mild because it allows for multiple optimal or second-optimal categories to have identical expected rewards. It only requires that not all arm categories share the exact same expected reward, ensuring there is *some* discernible gap between the best group of categories and the next-best group. Without such a nonzero gap, no method can certify optimal-category inclusion. Crucially, our main regret bound (Thm. 5.1) does not rely on this $\zeta$-margin at all, demonstrating the generality of our primary theoretical result.
>
>
>
> **Network width scaling with $K$.**
> The width $m$ in Thm. 5.1 scales as
> $
> m \ge \Omega \big(\mathrm{poly}(T,L,R,\bar\lambda^{-1}_0,S^{-1})\cdot K \xi_R \log(1/\delta)\big),
> $
> i.e., polynomial in problem/optimization parameters and *linear* in $K$ through the exploration-related factor $\xi_R$. Meanwhile, the regret depends on the effective dimension $\tilde d$ and on $\log(1+TK/\lambda)$:
> $
> R(T)\le O \Big(\sqrt{T \tilde{d} \log(1{+}TK/\lambda)}\big(\alpha\sqrt{\tilde{d} \log(1{+}TK/\lambda)}-2\log\delta + \sqrt{\lambda} S\big)\Big)+\tilde O(\sqrt{T}),
> $
> so $K$ enters *logarithmically* in the leading factor.
> Meanwhile, our NTK Gram matrices only involve at most $3T$ arms (chosen, optimal, and category-best), which keeps $\tilde d$ compact compared to definitions based on all $TK$ candidate arms.
>
>
>
> In this context, (i) the NTK/over-parameterization assumption is standard and used for analysis. (ii) The $\zeta$-margin is a mild, local identifiability condition used *only* for the “optimal-category is selected” theorem; the general regret result holds without it. (iii) The dependence on $K$ is linear in the (sufficient) width condition but only logarithmic in the regret via $\log(1{+}TK/\lambda)$, while our Gram-matrix construction controls complexity by restricting to $\le 3T$ arms.
>
>
>
>
>
>
>
>
>
>
>
>
>
>
>
>
>
> -----------------------------------
>
>
> > Q2: Curves showing when the optimal class consistently enters the shortlist?
>
>
>
>
> We follow the reviewer’s comments by incorporating additional results illustrating the category-level recommendation outcomes, reporting both the cumulative count of overlapping categories and the cumulative any-overlap incidence across time steps.
> The results in Fig. 6 of the revised manuscript demonstrate the effectiveness of category-level selection, where we observe an increasing success and category coverage rates, as the number of time steps grows.

---

> ### Author Response · Authors · 2025-10-08
> **Thank you for the thoughtful review! (2/5)**
>
> -----------------------------------
>
>
>
> > Q3: Does injecting conservative pseudo-labels for a few non-selected classes improve performance?
>
>
>
> We thank the reviewer for your comments.
> Following the reviewer's suggestion, we conducted an additional experiment with conservative category labels. In this setup, at each time step, we first randomly sample a portion of the un-selected categories. Then, for these specific categories, we train the category model using a small, conservative pseudo-label instead of our standard pseudo-reward.
>
>
>
>
> As shown in Fig. 7 of the revised manuscript, injecting conservative pseudo-labels impacts performance differently across datasets. For Amazon, the resulting regret curves are consistently above the baseline (indicating worse performance), and this gap tends to increase over time. One reason is that these biased labels can potentially dilute reward signals and hinder useful exploration by forcing the model to consider randomly chosen, non-selected categories. For Yelp, the conservative labels initially lead to slightly better results. However, this performance gain eventually disappears, leading to sub-optimal performance in the long run. We also observe that Yelp is less sensitive than Amazon to changes in the conservative label's magnitude and the sampling portion.
>
>
>
>
> -----------------------------------
>
>
> > Q4: Test robustness to perturbed categories.
>
>
>
>
>
> Following the reviewer's suggestion, we conducted an additional experiment to test the robustness of the proposed bi-level architecture. Here, we inject zero-mean Gaussian perturbations into the category representation: for each round $t$ and category $c$, we add random noise drawn from $\mathcal{N}(0,\sigma^2 I)$ to the normalized arm contexts before aggregating them to form the category feature.
> We vary the noise scale $\sigma \ge 0$ (with $\sigma = 0$ recovering the original setup) and evaluate the resulting category selection and regret, thereby testing the category module while keeping the arm module and supervision signals clean.
>
>
>
>
>
> Shown in Fig. 8 of the revised manuscript, perturbing only the category representation indicates that our bi-level architecture is stable under mild noise, but the performance can vary as the noise level increases.
> We observe the effect is dataset-dependent: Yelp shows only small performance difference at a large perturbation magnitude $\sigma=0.2$, whereas Amazon can exhibit a more noticeable rise in cumulative regret given a strong perturbation $\sigma=0.2$.
> In this context, these trends suggest the impacts of such injected noise can be application-specific.
> Developing dedicated denoised aggregation mechanisms to handle such category-injected noise is a promising future direction of our work, to further strengthen the stability under adversarial settings.

---

> ### Author Response · Authors · 2025-10-08
> **Thank you for the thoughtful review! (3/5)**
>
> -----------------------------------
>
>
> > Q5: Category encoding with Quantile/top-k features?
>
>
> Here, we explore enriching our category representation with two alternative formulations.
> We will concatenate each of these two formulations with the current category representation, increasing the dimension of the new representation to $3d$.
>
>
> **Quantile Category Feature.**
> To help capture rare but potentially high-value information within a category, we augment the category representation with a per-dimension quantile of the arm contexts. Let $\mathcal{X}\_{c,t}=\\{ \mathbf{x}\_1,\ldots,\mathbf{x}\_{n_c} \\}\subset\mathbb{R}^d$ denote the set of normalized arm contexts available for category $c$ at round $t$. In addition to the mean and standard deviation vectors,
> $
> \boldsymbol{\mu}\_{c,t}=\frac{1}{n_c} \sum\_{i=1}^{n_c} \mathbf{x}\_i,~~
> \boldsymbol{\sigma}\_{c,t}=\sqrt{\frac{1}{n\_c}\sum_{i=1}^{n\_c}(\mathbf{x}\_i - \boldsymbol{\mu}\_{c,t})\^{\odot 2}},
> $
> we compute a coordinate-wise $q$-quantile $\mathbf{q}\_{c,t} \in \mathbb{R}^d$ with $q \in (0,1)$, where $[\mathbf{q}\_{c,t}]\_j$ is the $q$-quantile of $\\{ [\mathbf{x}\_i]\_j \\}\_{i=1}\^{n_c}$.
>
> The resulting feature is the concatenation $\nu\^{\text{quant}}\_{c,t} = \operatorname{norm} \big([\boldsymbol{\mu}\_{c,t};\boldsymbol{\sigma}\_{c,t};\mathbf{q}\_{c,t}]\big)$, where $\operatorname{norm}(\cdot)$ denotes $L_2$ normalization.
> Choosing a high quantile (we set $q=0.7$) helps emphasize the upper tail of each coordinate distribution, making the representation more sensitive to rare but strong signals.
>
>
>
>
> **Top-$k$ Category Feature.**
> We can also augment the representation with a pooled statistic over the top-$k$ arms according to a simple, permutation-invariant score. Given a scoring function $s(\cdot)$ (we apply coordinate variance), select the indices of the $k$ highest-scoring arms, $I_k = \operatorname{arg top-k}\_{i\in[n_c]} s(\mathbf{x}_i)$, and compute the top-$k$ mean
> $ \bar{\mathbf{x}}\^{(k)}\_{c,t} = $ $\frac{1}{k} \sum\_{i \in I_k} \mathbf{x}_i. $
>
> Concatenating this vector with the base statistics yields $\nu\^{\text{topk}}\_{c,t} = \operatorname{norm} \big([\boldsymbol{\mu}\_{c,t} ; \boldsymbol{\sigma}\_{c,t} ; \bar{\mathbf{x}}\^{(k)}\_{c,t}] \big)$. This heuristic highlights can rare information-rich arms that might be washed out by the global mean, and we set $k=5$.
>
>
>
>
>
>
>
>
>
>
>
> Shown in Fig. 9 of the revised manuscript, augmenting the category representation with tail-sensitive signals (quantile or top-$k$) can yield mixed effects: modest gains on Yelp and slight performance degradation on Amazon. These results suggest that while emphasizing rare, high-magnitude arms can help when the tail carries actionable signal, it can also introduce variance or complexity when the tail is noisy. Given the dataset-dependent benefits, we retain our basic mean+std representation as our default and view quantile or top-$k$ representation augmentation as optional and data-dependent enhancements that the practitioner can choose, based on their specific application scenarios.

---

> ### Author Response · Authors · 2025-10-08
> **Thank you for the thoughtful review! (4/5)**
>
> -----------------------------------
>
>
>
> > Q6: What numerical safeguards are used for the arm-level UCB (e.g., covariance updates, regularization)?
>
>
> Thank you for the insightful question. We use several built-in stabilizers in the arm-level UCB to ensure numerically-stable updates and theoretical analysis:
>
> 1. **Ridge regularization in the covariance.**
>   The gradient covariance is updated with an explicit safeguard originated from the our model training (Section 4.2) and kernelized ridge-regression-based analysis:
>   $\Sigma\_{t-1} = \lambda I + \sum\_{\tau\in [t]} g(x\_{\tau};[\Theta_A]\_{\tau-1}) g(x\_{\tau};[\Theta_A]\_{\tau-1})\^\top,$
>   and the UCB uses
>   $
>   UCB\_{\alpha}(x_t) = \gamma\_{t-1}(\alpha)  \sqrt{ g(x_t;[\Theta_A]\_{t-1})^{ \top} \Sigma\_{t-1}^{-1} g(x\_t;[\Theta_A]\_{t-1}) }
>   $.
>   In this context, the $\lambda I$ guarantees invertibility of the matrix $\Sigma$ and improves the computation stability.
>
>
> 2. **Spectral bounds on the covariance trajectory.**
>   For our theoretical analysis, we leverage uniform bounds on $\\|\Sigma\\|_2$, on $\\|\Sigma-\Sigma^{(0)}\\|_F$ under over-parameterization settings, and on the log-determinant increments, which collectively ensure that the inverse and the UCB radius remain numerically stable over time:
>   $ \\|\Sigma\\|_2 \le \lambda + O(TL), \big\|\log \tfrac{ \det(\Sigma) }{ \det(\lambda I) } - \log \tfrac{ \det(\Sigma^{(0)}) }{ \det(\lambda I) } \big\| \le O(\cdots).
>   $
>   (please see Lemma J.10 for explicit rates.)
>
>
> 3. **Positive definiteness via NTK/Gram structure.**
>   Our analysis ties the reward map to a linearization in the NTK feature space and works with NTK Gram matrices over *non-duplicate* arms, which are naturally positive definite. This guarantees that the induced covariance forms are well-conditioned under our width assumptions.
>   The formulation of dual NTK Gram matrices also enable us to get rid of the arm separateness assumptions (Remark 2).
>
> 4. **Bounded gradients from initialization and smoothness lemmas.**
>   With Gaussian initialization and standard over-parameterized conditions, the network outputs and gradients remain bounded during training with over-parameterization, preventing scaling of $g(x;\Theta)$ that would destabilize $\Sigma_{t-1}$ and its inverse. (please see Lemma J.7 / J.8 and related bounds used in deriving the UCB.)
>
>
>
>
> In this context, the ridge term $\lambda I$, log-determinant control of cumulative leverage, spectral bounds on $\Sigma_t$, and positive-definite guarantees from the NTK/Gram construction provide the numerical safeguards for the arm-level UCB used in our algorithm and experiments.
>
>
>
>
>
>
>
>
>
>
>
>
>
>
> -----------------------------------
>
>
>
>
> > Q7: Removing the exploration head?
>
> In our framework, the category level uses two heads $f\^{(1)}\_{C}$ (exploitation) and $f\^{(2)}\_{C}$ (exploration) for selection before arm enumeration, while the arm level applies a lightweight UCB-style bonus controlled by $\alpha$; ablations correspond exactly to disabling $f\^{(2)}\_{C}$ or setting $\alpha=0$.
>
>
> **Category level.**
> Disabling the category exploration head $f\^{(2)}\_{C}$ (“No Cat. Exp.”) consistently hurts performance. In Table 2, regret increases from the best setting with exploration (Yelp: $3077{\pm}41$ at $\alpha \in [0.1,0.3]$; Amazon: $1623{\pm}42$ at $\alpha=0.1$) to notably worse values when $f\^{(2)}\_{C}$ is removed (Yelp: $3345{\pm}23$; Amazon: $1761{\pm}27$). This confirms that the adaptive category-level exploration term is essential for filtering and for maintaining strong downstream arm selection.
>
>
> **Arm level.**
> Removing the arm-level exploration (by setting the exploration coefficient $\alpha=0$ for the UCB bonus) also degrades performance relative to modest exploration: on Yelp, regret rises from $3077{\pm}41$ (at $\alpha \in [0.1,0.3]$) to $3212{\pm}22$ at $\alpha=0$; on Amazon, from $1623{\pm}42$ ($\alpha=0.1$) to $1809{\pm}36$ at $\alpha=0$. Overall, a small but nonzero arm-level exploration is beneficial and complements the category-level exploration.

---

> ### Author Response · Authors · 2025-10-08
> **Thank you for the thoughtful review! (5/5)**
>
> -----------------------------------
>
>
>
>
>
> > Q8: How sensitive is performance to $B$ (number of selected categories per round)?
>
>
>
> Our parameter study shows that the effect of $B$ is task–dependent. On Yelp (20 categories), increasing $B$ generally improves performance by raising the chance that the optimal category is included each round; a moderate setting such as $B=4$ offers a strong trade–off between regret and runtime (Table 1). On Amazon (10 categories), larger $B$ does not always help and can degrade performance by weakening the filtering effect (also in Table 1). Practically, we recommend tuning $B$ from smaller values upward to balance recommendation quality and efficiency.
>
>
>
> In this context, the theoretical analysis also reflects this trade–off: with $B=|\widetilde{\mathcal{C}}_t|$ categories retained at round $t$, the condition for including the optimal category scales with $B$ via an allowable total estimation error of $\le B\cdot \zeta$. Thus, increasing $B$ relaxes the per–category precision needed to keep the optimal category (helpful when categories are many/diverse), but overly large $B$ reduces the sharpness of the filter and can invite unnecessary arm–level exploration in practice.
>
>
>
>
>
>
> **References**
>
> [1] Yikun Ban, Jingrui He, and Curtiss B Cook. Multi-facet contextual bandits: A neural network
> perspective. In Proceedings of the 27th ACM SIGKDD Conference on Knowledge Discovery & Data Mining, pages 35–45, 2021.
>
> [2] Zhongxiang Dai, Yao Shu, Arun Verma, Flint Xiaofeng Fan, Bryan Kian Hsiang Low, and Patrick Jaillet. Federated neural bandits. In The Eleventh International Conference on Learning Representations, 2023.
>
> [3] Pan Xu, Zheng Wen, Handong Zhao, and Quanquan Gu. Neural contextual bandits with deep representation and shallow exploration. arXiv preprint arXiv:2012.01780, 2020.
>
> [4] Weitong Zhang, Dongruo Zhou, Lihong Li, and Quanquan Gu. Neural thompson sampling. In
> International Conference on Learning Representations, 2021.
>
> [5] Dongruo Zhou, Lihong Li, and Quanquan Gu. Neural contextual bandits with ucb-based exploration. In International Conference on Machine Learning, pages 11492–11502. PMLR, 2020.

---

> > ### Comment · Reviewer_gBv8 · 2025-10-30
> >
> > Thank you for the detailed response. It addresses my concerns, and I have no further questions.

---

### Review · Reviewer_5nwy · 2025-09-27

**Summary Of Contributions:**

This paper presents a novel bi-level hierarchical neural contextual bandit algorithm H_2N-Bandit for online recommendation system. The proposed framework aims to mitigate the huge computational cost incurred in traditional neural contextual bandits, where the computational cost scale as O(K) with K being the number of arms. The hierarchical approach adopted consists of two levels: a category-level recommendation at the outer level where a set of arm categories are first recommended, and an arm-level recommendation at the inner level, where the optimal arm is selected from the pre-filtered set of arm categories. Theoretical guarantees are provided for overparameterized neural models, which scale similar to the conventional neural bandits, but at far reduced computational time. Extensive experiments have been provided.

The paper is very clearly written, and is easy to read and understand. However, some points need further clarification, which I list below.

**Audience:**

Yes

**Audience Explanation:**

Yes, using neural network models for highly non-linear reward function has been of interest for quite some time, with several works. The critical challenge here is the computational complexity associated with the online learning. This paper presents a hierarchical model for contextual neural bandit learning that can address the computational challenges, with similar regret guarantees.

**Broader Impact Concerns:**

No concerns

**Claims And Evidence:**

Yes

**Claims Explanation:**

Rigourous theoretical proofs are provided, along with extensive experiments on MovieLens, Yelp and Amazon experiments.

**Requested Changes:**

The regret bound in Theorem 5.1 is shown to scale similarly as in conventional neural bandits. It is good to explain how the effective dimension is affected when operating with almost 3T arms, at least experimentally. Further more, it is not clear what  denotes in the theorem and how it scales with T.
The algorithm now leverages three overparameterized neural networks that need to be trained. In particular, the network f_C^2(\cdot) takes in input gradients from the first network f_C^1, whereby the input dimension scales with the number of parameters. How do these contribute to the computational complexity, in comparison to the conventional neuralUCB?
How is exploration and exploitation trade-off implemented at the category level recommendation?

---

> ### Author Response · Authors · 2025-10-08
> **Thank you for the thoughtful review! (1/2)**
>
> We sincerely thank the reviewer for the valuable comments and suggestions, which have been instrumental in enhancing the quality of our paper. We will try to address your feedback in the following Q\&A format. Please also refer to our **updated manuscript for added experiments**.
> *Thank you again for your thoughtful review!*
>
>
> ----------------------
>
>
>
>
>
> > Q1: How the effective dimension is affected when operating with almost 3T arms? How it scales with $T$?
>
>
> Thank you for the insightful question.
> Here, let $M_{TK}$ be the kernel Gram matrix from all $TK$ arms and $M_{3T}$ the one from the $3T$-arm subset. After re-indexing, $M_{3T}$ is a principal sub-matrix of $M_{TK}$, so their eigenvalues interlace and satisfy $\mu_i(M_{3T})\le \mu_i(M_{TK})$ for $i\le 3T$. Since $g(u)=\log(1+u/\lambda)$ is increasing and concave on $[0,\infty)$, we have
> $\log\det(I+M_{3T}/\lambda)=\sum_{i=1}^{3T} g(\mu_i(M_{3T}))\le \sum_{i=1}^{3T} g(\mu_i(M_{TK}))\le \sum_{i=1}^{TK} g(\mu_i(M_{TK}))=\log\det(I+M_{TK}/\lambda)$.
> Using the same positive denominator $\log(1+TK/\lambda)$ for both definitions then gives $\tilde d(M_{3T},\lambda)\le \tilde d(M_{TK},\lambda)$, with equality only if the omitted arms add no new kernel directions.
>
>
>
>
>
>
>
>
> Meanwhile, we agree that it is valuable to visualize the effective dimension discrepancy to demonstrate the growth rate difference between $TK$ arms and $3T$ arms in Fig. 4 of the revised manuscript. Consistent with our theory, the empirical curves show $\tilde d_{3T}(T,\lambda)\le \tilde d_{TK}(T,\lambda)$ for all $T$, with the $3T$ curve growing more slowly and typically saturating earlier, indicating fewer novel kernel directions are added per round. This gap quantifies the redundancy present when all $K$ candidates are included each round and highlights that restricting to three representative arms controls capacity without increasing the effective dimension, which in turn tightens the constants in regret bounds and stabilizes learning.
>
>
>
>
>
> ----------------------
>
>
>
>
> > Q2: The network $f_C^2(\cdot)$ takes in input gradients from the first network $f_C^1$, whereby the input dimension scales with the number of parameters. How do these contribute to the computational complexity?
>
>
>
> Thank you for the insightful question.
> As discussed in paragraph right above Subsection 4.1.2, to enhance efficiency for practical applications and experiments, we adopt the common practice from neural adaptive exploration [1, 2] of applying pooling to the gradients $\nabla f_{C}^{(1)}$, before their use as input to the exploration network $f_{C}^{(2)}$. This pooling mechanism allows for control over the input dimension, balancing computational complexity against the informational granularity of the gradient signal. We can therefore evaluate the resulting performance and runtime trade-offs based on various pooling sizes.
>
> Across both datasets (Fig. 5 of the revised manuscript), varying the pooling size, namely the dimensionality of the gradient features passed from $\nabla f_C^{(1)}$ to $f_C^{(2)}$, has no significant impact on runtime, while regret remains close to the reference.
> This indicates that, within the tested range, the exploration network input dimension is not a runtime bottleneck and the added input dimensions do not translate into substantial computational overhead, **as the main computational bottleneck is still at the arm-level gradient-based exploration**.
> Still maintaining a computational advantage over conventional Neural-UCB, one can choose a pooling size to capture sufficient gradient granularity for exploration without particularly worrying about meaningful increases in running time.

---

> ### Author Response · Authors · 2025-10-08
> **Thank you for the thoughtful review! (2/2)**
>
> ----------------------
>
>
>
>
>
>
>
> > Q3: How is exploration and exploitation trade-off implemented at the category level recommendation?
>
>
> The exploration-exploitation trade-off at the category level is a core component of our proposed framework, implemented through a dual neural network architecture inspired by neural adaptive exploration methods [1, 2]. This design allows for a more flexible, adaptive exploration strategy compared to traditional UCB methods.
>
> The mechanism consists of two separately trained neural networks:
>
> (1) **An Exploitation Network ($f_{C}^{(1)}$):** This network's primary role is exploitation. It is trained to directly predict the expected reward of a given category from its representation $\mathbf{\nu}\_{c, t}$. The optimization process for this network uses the category representations as input and their corresponding (pseudo) rewards $\widetilde{r}\_{c, t}$ as labels.
>
> (2) **An Exploration Network ($f_{C}^{(2)}$):** This second network is responsible for principled exploration. It is trained to predict the estimation uncertainty, or "potential gain," of the exploitation network. Critically, it takes the *gradient* of the exploitation network, $\nabla f_{C}^{(1)}(\mathbf{\nu}\_{c, t})$, as its input. The label used to train this network is the residual, i.e., the difference between the category reward and the exploitation network's estimate (calculated via the derived pseudo-reward due to the absence knowledge of the true category reward). By learning to predict this residual from the gradient, $f_{C}^{(2)}$ can therefore model the uncertainty of $f_{C}^{(1)}$.
>
>
> The final decision is based on an *overall score* for each category, which is the sum of the outputs from both networks, as shown in Line 6 of Algorithm 1:
> $$
> \mathcal{F}\_{C}(\mathbf{\nu}\_{c,t}) = f_{C}^{(1)}(\mathbf{\nu}\_{c,t}) + f_{C}^{(2)} \big(\nabla f_{C}^{(1)}(\boldsymbol{\nu}_{c,t})\big).
> $$
> *The first term is exploitation (estimated reward); the second term is exploration (learned uncertainty).*
>
>
> A key advantage of this adaptive approach is that the exploration score from $f_{C}^{(2)}$ can be *negative*. This allows the model to perform "downward" exploration to correct for instances where the exploitation network $f_{C}^{(1)}$ overestimates a category's reward (Section 6.2), and its effectiveness is also evaluated via an ablation study in Table 2.
>
> In contrast, traditional UCB-based methods can only provide a non-negative exploration bonus, which can unintentionally amplify such overestimation errors. Our method can thus adaptively apply a positive bonus to explore uncertain but promising categories or a negative correction to temper over-optimistic estimates, leading to an efficient category-level exploration strategy.
>
>
>
> **References**
>
> [1] Yikun Ban, Yuchen Yan, Arindam Banerjee, and Jingrui He. Ee-net: Exploitation-exploration neural networks in contextual bandits. In International Conference on Learning Representations, 2022.
>
> [2] Yunzhe Qi, Yikun Ban, and Jingrui He. Graph neural bandits. In Proceedings of the 29th ACM SIGKDD Conference on Knowledge Discovery and Data Mining, KDD ’23, pp. 1920–1931, New York, NY, USA, 2023. Association for Computing Machinery. ISBN 9798400701030. Doi: 10.1145/3580305.3599371. URL https://doi.org/10.1145/3580305.3599371.

---

### Comment · Reviewer_5nwy · 2025-10-30
**Thank you for the response**

I appreciate and thank the authors for careful response to all my questions. I am happy with the response, and have no further questions.

---

### Decision · Action_Editor_ZxJh · 2025-12-10

**Recommendation:** Accept as is

**Audience:**

Yes

**Audience Explanation:**

This paper is on multi-armed bandits and neural networks, two central topics in machine learning.

**Claims And Evidence:**

Yes

**Claims Explanation:**

The claims made are accurate. I checked the proofs at a high level and the results look believable.